# Least-to-Most Prompting Enables Complex Reasoning in Large Language Models

**Denny Zhou**[*]  **Nathanael Schärli**   **Le Hou**   **Jason Wei**   **Nathan Scales**   **Xuezhi Wang**
**Dale Schuurmans**   **Claire Cui**   **Olivier Bousquet**   **Quoc Le**   **Ed Chi**

Google Research, Brain Team

## Abstract

Chain-of-thought prompting has demonstrated remarkable performance on various natural language reasoning tasks. However, it tends to perform poorly on tasks which requires solving problems harder than the exemplars shown in the prompts. To overcome this challenge of easy-to-hard generalization, we propose a novel prompting strategy, *least-to-most prompting*. The key idea in this strategy is to break down a complex problem into a series of simpler subproblems and then solve them in sequence. Solving each subproblem is facilitated by the answers to previously solved subproblems. Our experimental results on tasks related to symbolic manipulation, compositional generalization, and math reasoning reveal that least-to-most prompting is capable of generalizing to more difficult problems than those seen in the prompts. A notable finding is that when the GPT-3 `code-davinci-002` model is used with least-to-most prompting, it can solve the compositional generalization benchmark SCAN in any split (including length split) with an accuracy of at least 99% using just 14 exemplars, compared to only 16% accuracy with chain-of-thought prompting. This is particularly noteworthy because neural-symbolic models in the literature that specialize in solving SCAN are trained on the entire training set containing over 15,000 examples. We have included prompts for all the tasks in the Appendix.

## 1 Introduction

Despite the great success of deep learning in the past decade, there still remain huge differences between human intelligence and machine learning: (1) Given a new task, humans usually can learn to accomplish it from only a few demonstration examples, while machine learning requires a large amount of labeled data for model training; (2) Humans can clearly explain the underlying rationale for their predictions or decisions, while machine learning is essentially a black box; (3) Humans can solve problems more difficult than any they have seen before, while for machine learning, examples in training and testing are typically at the same level of difficulty.

The recently proposed chain-of-thought prompting approach (Wei et al., 2022; Chowdhery et al., 2022) has taken a significant step for narrowing the gap between human intelligence and machine intelligence. It combines the idea of natural language rationales (Ling et al., 2017; Cobbe et al., 2021) with few-shot prompting (Brown et al., 2020). When further integrated with self-consistency decoding (Wang et al., 2022b) rather than using the typical greedy decoding, few-shot chain-of-thought prompting largely outperforms the state-of-the-art results in the literature on many challenging natural language processing tasks obtained from specially designed neural models trained with hundreds of times more annotated examples, while being fully interpretable.

However, chain-of-thought prompting has a key limitation—it often performs poorly on tasks that require generalization of solving problems harder than the demonstration examples, such as compositional generalization (Lake & Baroni, 2018; Keysers et al., 2020). To tackle such easy-to-hard generalization issues, we propose *least-to-most prompting*. It consists of two stages: first decomposing a complex problem into a list of easier subproblems, and then sequentially solving these

---
[*]Corresponding to: dennyzhou@google.com

subproblems, whereby solving a given subproblem is facilitated by the answers to previously solved subproblems. Both stages are implemented by few-shot prompting, so that there is no training or finetuning in either stage. An example usage of least-to-most prompting is illustrated in Figure 1.

The term least-to-most prompting is borrowed from educational psychology (Libby et al., 2008), where it is used to denote the technique of using a progressive sequence of prompts to help a student to learn a new skill. Here we apply this technique for teaching humans to teach language models. Empirical results on symbolic manipulation, compositional generalization, and math reasoning show that least-to-most prompting can indeed generalize to problems harder than those demonstrated.

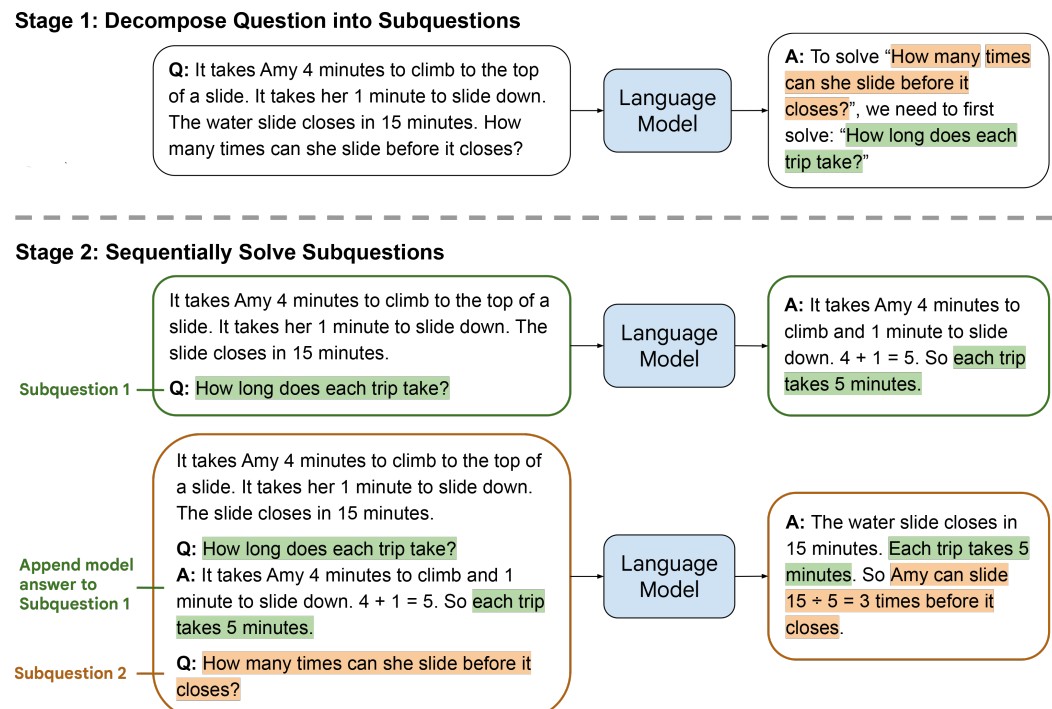

Figure 1: Least-to-most prompting solving a math word problem in two stages: (1) query the language model to decompose the problem into subproblems; (2) query the language model to sequentially solve the subproblems. The answer to the second subproblem is built on the answer to the first subproblem. The demonstration examples for each stage's prompt are omitted in this illustration.

## 2 LEAST-TO-MOST PROMPTING

Least-to-most prompting teaches language models how to solve a complex problem by decomposing it to a series of simpler subproblems. It consists of two sequential stages:

1. **Decomposition**. The prompt in this stage contains constant examples that demonstrate the decomposition, followed by the specific question to be decomposed.

2. **Subproblem solving**. The prompt in this stage consists of three parts: (1) constant examples demonstrating how subproblems are solved; (2) a potentially empty list of previously answered subquestions and generated solutions, and (3) the question to be answered next.

In the example shown in Figure 1, the language model is first asked to decompose the original problem into subproblems. The prompt that is passed to the model consists of examples that illustrate how to decompose complex problems (which are not shown in the figure), followed by the specific problem to be decomposed (as shown in the figure). The language model figures out that the original problem can be solved via solving an intermediate problem "How long does each trip take?".

In the next phase, we ask the language model to sequentially solve the subproblems from the problem decomposition stage. The original problem is appended as the final subproblem. The solving starts from passing to the language model a prompt that consists of examples that illustrate how problems are solved (not shown in the figure), followed by the first subproblem "How long does each trip take?". We then take the answer generated by the language model ("... each trip takes 5 minutes.") and construct the next prompt by appending the generated answer to the previous prompt, followed by the next subproblem, which happens to be the original problem in this example. The new prompt is then passed back to the language model, which returns the final answer.

Least-to-most prompting can be combined with other prompting techniques like chain-of-thought (Wei et al., 2022) and self-consistency (Wang et al., 2022b), but does not need to be. Also, for some tasks, the two stages in least-to-most prompting can be merged to form a single-pass prompt.

## 3 RESULTS

We present least-to-most prompting results for symbolic manipulation, compositional generalization, and math reasoning tasks, and compare it with chain-of-thought prompting.

### 3.1 SYMBOLIC MANIPULATION

We take the last-letter-concatenation task (Wei et al., 2022). In this task, each input is a list of words, and the corresponding output is the concatenation of the last letters of the words in the list. For example, "thinking, machine" outputs "ge", since the last letter of "thinking" is "g" and the last letter of "machine" is "e". Chain-of-thought prompting does a perfect job when the testing lists have the same length as the lists in the prompt exemplars. However, it performs poorly when the testing lists are much longer than the lists in the prompt exemplars. We show that least-to-most prompting overcomes this limitation and significantly outperforms chain-of-thought prompting on length generalization.

---

Q: "think, machine, learning"
A: "think", "think, machine", "think, machine, learning"

---

Table 1: Least-to-most prompt context (decomposition) for the last-letter-concatenation task. It can decompose arbitrary long lists into sequential sublists with an accuracy of 100%.

---

Q: "think, machine"
A: The last letter of "think" is "k". The last letter of "machine" is "e". Concatenating "k", "e" leads to "ke". So, "think, machine" outputs "ke".

Q: "think, machine, learning"
A: "think, machine" outputs "ke". The last letter of "learning" is "g". Concatenating "ke", "g" leads to "keg". So, "think, machine, learning" outputs "keg".

---

Table 2: Least-to-most prompt context (solution) for the last-letter-concatenation task. The two exemplars in this prompt actually demonstrate a base case and a recursive step.

**Least-to-most prompting**. The least-to-most prompt contexts for the last-letter-concatenation task are shown in Tables 1 and 2. The exemplar in Table 1 demonstrates how to decompose a list into a sequence of sublists. The exemplar in Table 2 demonstrates how to map an input to the desired output. Given a new list, we first append it to the exemplar in Table 1 to construct the decomposition prompt, which is sent to the language model to obtain the list's decomposition. Then, we construct for each sublist $S$ a solution prompt, which consists of the exemplars in Table 2, followed by the previous sublist/response pairs (if any), followed by $S$. We sequentially issue these prompts to the language model and use the last response as the final solution.

It is worth a closer look at the exemplars in Table 2. Essentially, they teach language models how to build answers to new problems using the answers to previously solved problems: (1) the list in the

second exemplar ("think, machine, learning") is an extension of the list in the first exemplar ("think, machine") rather than an entirely independent one; (2) the response to "think, machine, learning" is built on the output of "think, machine" by starting with a sentence saying that "think, machine" outputs "ke". The two exemplars together illustrate a base case and a recursive step.

**Chain-of-thought prompting**. The chain-of-thought prompt context for the last-letter-concatenation task is listed in Table 3. It uses the same lists as the least-to-most prompt in Table 2. The only difference is that, in the chain-of-thought prompt, the response to the second list ("think, machine, learning") is built from scratch, instead of using the output of the first list ("think, machine").

---

Q: "think, machine"
A: The last letter of "think" is "k". The last letter of "machine" is "e". Concatenating "k", "e" leads to "ke". So, "think, machine" outputs "ke".

Q: "think, machine, learning"
A: The last letter of "think" is "k". The last letter of "machine" is "e". The last letter of "learning" is "g". Concatenating "k", "e", "g" leads to "keg". So, "think, machine, learning" outputs "keg".

---

Table 3: Chain-of-thought prompt context for the last-letter-concatenation task. Unlike the least-to-most prompt in Table 2, the exemplars in the chain-of-thought prompt are independent of each other.

We compare least-to-most prompting (Table 1 & 2) with chain-of-thought prompting (Table 3) and the standard few-shot prompting. The prompt for the standard few-shot prompting is constructed by removing the intermediate explanations in the chain-of-thought prompt. That is, it just consists of these two exemplars: (1) "think, machine" outputs "ke"; and (2) "think, machine, learning" outputs "keg". We do not consider a training or finetuning baseline because a machine learning model based on two examples would generalize very poorly.

**Results**. We randomly sample words in Wiktionary[1] to construct testing lists with lengths varying from 4 to 12. For each given length, 500 lists are constructed. The accuracies of different methods with `code-davinci-002` in GPT-3 are shown in Table 4. Standard prompting completely fails all test cases with an accuracy of 0. Chain-of-thought prompting significantly boosts the performance over standard prompting, but it still falls well behind least-to-most prompting, particularly when the lists are long. Moreover, the performance of chain-of-thought prompting drops much faster than least-to-most prompting as the length increases.

|  | $L = 4$ | $L = 6$ | $L = 8$ | $L = 10$ | $L = 12$ |
|---|---|---|---|---|---|
| Standard prompting | 0.0 | 0.0 | 0.0 | 0.0 | 0.0 |
| Chain-of-Thought | 84.2 | 69.2 | 50.2 | 39.8 | 31.8 |
| Least-to-Most | **94.0** | **88.4** | **83.0** | **76.4** | **74.0** |

Table 4: Accuracies of different prompting methods on the last-letter-concatenation task. The length of testing lists increases from 4 to 12.

In Appendices 7.2 and 7.3, we present additional experiments with different chain-of-thought prompts and different language models. Note that in contrast to least-to-most prompting, the exemplars in a chain-of-thought prompt can be independent of each other. For the last-letter concatenation task, this means that we do not need to present exemplars that are sublists of other exemplars. In fact, a chain-of-thought prompt with independent lists tends to outperform one with dependent lists, as the former conveys more information. Furthermore, we can enhance chain-of-thought prompting by incorporating additional exemplars. This seems to be fair, as the least-to-most prompt contains more words due to its extra decomposition. As shown in Table 13 (Appendix 7.3), for lists with length 12, chain-of-thought prompting achieves an accuracy of 37.4% with 4 independent exemplars (Appendix 7.2.2), and 38.4% with 8 independent exemplars (Appendix 7.2.3). Although there

---

[1] https://en.wiktionary.org/wiki/Wiktionary:Frequency_lists/PG/2006/04/1-10000

have been notable advancements compared to an accuracy of 31.8% by the original prompt in Table 3, chain-of-thought prompting still lags behind least-to-most prompting, which boasts an accuracy of 74.0%.

**Error analysis**. While least-to-most prompting significantly outperforms chain-of-thought prompting, it is still far from achieving 100% accuracy for long lists. In Appendix 7.4, we present a detailed error analysis. We find that only very few of them are due to incorrect last letters, while most of them are concatenation errors (dropping or adding a letter). For example, given the list "gratified, contract, fortitude, blew", the model drops the last letter in the concatenation of "dte" and "w", and thus predicts the outcome to be "dte" instead of "dtew". In another example "hollow, supplies, function, gorgeous", the model somehow duplicates the last letter "s" in the concatenation of "wsn" and "s", and thus the prediction becomes "wsnss" instead of "wsns".

## 3.2 COMPOSITIONAL GENERALIZATION

SCAN (Lake & Baroni, 2018) is probably the most popular benchmark for evaluating compositional generalization. It requires mapping natural language commands to action sequences (Table 5). Sequence-to-sequence models perform poorly under length split where the action sequences in the training set (about 80% of the full set with over 20,000 examples) are shorter than the action sequences in the testing set. Many specialized neural-symbolic models have been proposed to solve SCAN (Chen et al., 2020; Liu et al., 2020; Nye et al., 2020; Shaw et al., 2021; Kim, 2021). We show that large language models with least-to-most prompting can solve SCAN using only a few demonstration examples. No training or finetuning is needed.

| Command | Action Sequence |
|---------|-----------------|
| "look thrice after jump" | JUMP LOOK LOOK LOOK |
| "run left and walk" | TURN_LEFT RUN WALK |
| "look opposite right" | TURN_RIGHT TURN_RIGHT LOOK |

Table 5: Example commands in SCAN and their corresponding action sequences. An agent successfully executes a natural language command by performing its corresponding action sequence.

**Least-to-most prompting.** Like the last-letter-concatenation task in Section 3.1, least-to-most prompting for SCAN is based on two kinds of prompts: (1) a command decomposition prompt containing 8 exemplars to demonstrate how to decompose a long command to a list of short commands (see Table 6 for some of these exemplars); and (2) a command mapping prompt containing 14 exemplars to demonstrate how to map natural language commands to action sequences (see Table 7 for some of these exemplars). The full prompt contexts for both decomposition and mapping are listed in Appendix 8. The exemplars used in the command-mapping prompt are intended to fully cover the semantics of the SCAN commands.

---

Q: "look opposite right thrice after walk"
A: "look opposite right thrice" can be solved by: "look opposite right", "look opposite right thrice". "walk" can be solved by "walk". So, "look opposite right thrice after walk" can be solved by: "look opposite right", "look opposite right thrice", "walk".

Q: "look around right thrice and walk"
A: "look around right thrice" can be solved by: "look right", "look around right", "look around right thrice". "walk" can be solved by "walk". So, "look around right thrice and walk" can be solved by: "look right", "look around right", "look around right thrice", "walk".

---

Table 6: Two exemplars taken from the least-to-most prompt for decomposing a long command to a sequence of short commands.

We use Python notation to make our prompts in least-to-most prompting and the baselines (stand few-shot prompting and chain-of-thought prompting) concise and meet the input size limit of language models (usually up to 2048 tokens). For example, in our prompt design, we map "look twice"

to "LOOK" * 2 rather than "LOOK LOOK". We may have to point out that these Python expressions are just intermediate representations. After obtaining responses from language models, we run a postprocessing script to expand the Python expressions to generate the final results. However, it should not be surprising that we can also further teach language models to correctly process expressions. In Appendix 8.4, we show that with several demonstration exemplars, language models are able to expand Python expressions with a near-perfect accuracy of 99.7%.

---

Q: "jump left"
A: The output of "jump left" concatenates: the output of "turn left", the output of "jump". "turn left" outputs "TURN_LEFT". "jump" outputs "JUMP". So concatenating the output of "turn left" and the output of "jump" leads to "TURN_LEFT" + "JUMP". So the output of "jump left" is "TURN_LEFT" + "JUMP".

Q: "run and look twice"
A: The output of "run and look twice" concatenates: the output of "run", the output of "look twice". "run" outputs "RUN". "look twice" outputs "LOOK" * 2. So concatenating the output of "run" and the output of "look twice" leads to "RUN" + "LOOK" * 2. So the output of "run and look twice" is "RUN" + "LOOK" * 2.

---

Table 7: Two exemplars taken from the least-to-most / chain-of-thought prompt for mapping commands to action sequences. Python expressions are used as intermediate representations.

**Chain-of-thought prompting**. The chain-of-thought prompt for SCAN uses the same command-mapping context as least-to-most prompting (see Table 7) but it does not use command decomposition, which is exclusively used for least-to-most prompting.

**Results**. We compare least-to-most prompting with chain-of-thought prompting and standard few-shot prompting. The exemplars for standard few-shot prompting are derived from the chain-of-thought prompt by removing the intermediate explanations. The accuracies of different prompting methods with different language models are presented in Table 8. Example outputs can be found in Appendix 8.3. Using code-davinci-002, least-to-most prompting achieves an accuracy of 99.7% under length split. We also test least-to-most prompting on all other splits and even the full SCAN dataset. We find that its solving rate remains the same. In addition, it may be interesting to note that code-davinci-002 consistently outperforms text-davinci-002, regardless of the prompting method.

| Method | Standard prompting | Chain-of-Thought | Least-to-Most |
|---|---|---|---|
| code-davinci-002 | 16.7 | 16.2 | **99.7** |
| text-davinci-002 | 6.0 | 0.0 | **76.0** |
| code-davinci-001 | 0.4 | 0.0 | **60.7** |

Table 8: Accuracies (%) of different prompting methods on the test set of SCAN under length split. The results of text-davinci-002 are based on a random subset of 100 commands.

**Error analysis**. In the test set of the length split, there are 13 failures in total from least-to-most prompting: 6 of them incorrectly interpret "twice" and "thrice" following "around", and the rest incorrectly interpret "after" as "and". Let us show a failed example for each category. In the example "walk opposite right twice after run around right thrice", code-davinci-002 correctly translates the expression "run around right" to ("TURN_RIGHT" + "RUN") * 4. Then it makes a mistake when applying "thrice" to this expression and produces ("TURN_RIGHT" + "RUN") * 9 instead of ("TURN_RIGHT" + "RUN") * 4 * 3 or ("TURN_RIGHT" + "RUN") * 12. In the example "run opposite left thrice after run around left twice", code-davinci-002 produces the correct translations for both sub-expressions that are connected by "after" but it combines them as if they were connected by "and". This means that the model produces ("TURN_LEFT" * 2 + "RUN") * 3 + ("TURN_LEFT" + "RUN") * 4 * 2 instead of ("TURN_LEFT" + "RUN") * 4 * 2 + ("TURN_LEFT" * 2 + "RUN") * 3. A detailed error analysis can be found in Appendix 8.2.

## 3.3 MATH REASONING

In this section, we apply least-to-most prompting to solve math word problems in GSM8K (Cobbe et al., 2021) and DROP (Dua et al., 2019). We are particularly interested at seeing if large language models combined with least-to-most prompting can solve problems more difficult than those seen in the prompts. Here, we simply measure the difficulty by the number of solving steps.

---

Q: Elsa has 5 apples. Anna has 2 more apples than Elsa. How many apples do they have together?
A: Let's break down this problem: 1. How many apples does Anna have? 2. How many apples do they have together?

1. Anna has 2 more apples than Elsa. So Anna has 2 + 5 = 7 apples.
2. Elsa and Anna have 5 + 7 = 12 apples together.

The answer is: 12.

---

Table 9: The least-to-most prompt for solving GSM8K. The demonstration problem is solved in only two steps, but the prompt can handle problems that require multiple steps to be solved.

---

Q: Elsa has 5 apples. Anna has 2 more apples than Elsa. How many apples do they have together?
A: Anna has 2 more apples than Elsa. So Anna has 2 + 5 = 7 apples. So Elsa and Anna have 5 + 7 = 12 apples together.

The answer is: 12.

---

Table 10: The chain-of-thought prompt for solving GSM8K. It is derived from the least-to-most prompt in Table 9 by removing the decomposition part.

The prompt that we design to solve GSM8K is shown in Table 9. The demonstration exemplar consists of two parts. The first part (starting from "Let's break down this problem ...") shows how the original problem can be decomposed into simpler subproblems, and the the second part shows how the subproblems are solved in sequence. Note that this prompt combines decomposition and subproblem solving into a single pass. One may instead design two different prompts respectively for decomposition and subproblem solving, as the least-to-most prompts in the previous sections, to further improve performance. Here, we focus on investigating how this simple least-to-most prompt generalizes from a simple 2-step problem to more complex multi-step problems.

We also construct a chain-of-thought prompt (Table 10) as our baseline. It is derived from the least-to-most prompt (Table 9) by removing the decomposition part. The results are shown in Table 11. Overall, least-to-most prompting only slightly improves chain-of-thought prompting: from 60.97% to 62.39%. However, least-to-most prompting essentially improves chain-of-thought prompting in solving problems which need at least 5 steps to be solved: from 39.07% to 45.23% (Table 12). We find that almost every problem in GSM8K that least-to-most prompting fails to solve can be eventually solved by using a manually crafted decomposition. This should not be surprising. For our humans, as long as we know how to decompose a complex problem into simpler subproblems, we actually have solved it. For the DROP benchmark, least-to-most prompting outperforms chain-of-thought prompting by a large margin (Table 11). That is probably because most problems in DROP can be trivially decomposed.

| Method | Non-football (DROP) | Football (DROP) | GSM8K |
|---|---|---|---|
| Zero-Shot | 43.86 | 51.77 | 16.38 |
| Standard prompting | 58.78 | 62.73 | 17.06 |
| Chain-of-Thought | 74.77 | 59.56 | 60.87 |
| Least-to-Most | **82.45** | **73.42** | **62.39** |

Table 11: Accuracies (%) of different prompting methods on GSM8K and DROP (only the subset containing numerical problems). The base language model is `code-davinci-002`.

| Accuracy by Steps (GSM8K) | All | 2 Steps | 3 Steps | 4 steps | $\geq$ 5 steps |
|---|---|---|---|---|---|
| Least-to-Most | **62.39** | 74.53 | **68.91** | **59.73** | **45.23** |
| Chain-of-Thought | 60.87 | **76.68** | 67.29 | 59.39 | 39.07 |

Table 12: Accuracies (%) of least-to-most prompting and chain-of-thought prompting, broken down by the number of reasoning steps required in the expected solution.

## 4 RELATED WORK

**Compositional generalization**. SCAN (Lake & Baroni, 2018) is a widely used benchmark to evaluate compositional generalization. Among all of its splits, the most challenging is the length split, which requires a model to generalize to test sequences longer than training ones. Prior work with good performance on SCAN mostly proposed neural-symbolic architectures (Chen et al., 2020; Liu et al., 2020) and grammar induction techniques (Nye et al., 2020; Shaw et al., 2021; Kim, 2021). Chen et al. (2020) proposed the neural-symbolic stack machine, which contains a neural network as the controller to generate an execution trace for a given input, and a symbolic stack machine to execute the trace and produce the output. The execution trace consists of domain-specific primitives for sequence manipulation, which allows the machine to break down the input sentence into different components, translate them separately, and compose them together. Liu et al. (2020) proposed a framework that cooperatively learns two neural modules, a composer and a solver, to jointly learn the input structure and the symbolic grammar rules. Both Nye et al. (2020) and Shaw et al. (2021) inferred the symbolic grammar rules of SCAN, while Kim (2021) proposed to learn a latent neural grammar. While approaches with symbolic components are able to achieve 100% accuracy on SCAN (Chen et al., 2020; Liu et al., 2020; Nye et al., 2020; Shaw et al., 2021), they require complicated model training and grammar inference algorithms to search in a large grammar space. Another line of work on SCAN designs data augmentation schemes (Andreas, 2020; Akyürek et al., 2021; Lake, 2019). Both Andreas (2020) and Akyürek et al. (2021) construct synthetic training samples by recombining fragments occurring in different training samples, and Akyürek et al. (2021) further designs a sampling scheme that encourages the recombination model to produce rare samples. On the other hand, Lake (2019) proposed a meta training algorithm, which requires a meta-grammar space to construct training data, and the format of sampled grammars is similar to the SCAN grammar. While these data augmentation techniques improve the performance on several compositional generalization benchmarks, they fail to solve the length split of SCAN. Other prior works propose neural network architectures to improve compositional generalization, where they encourage the model to learn the word and span mapping (Russin et al., 2019; Li et al., 2019), the alignment of input and output as span trees (Herzig & Berant, 2021), and the permutation equivariance of input and output words (Gordon et al., 2020). Still, these end-to-end neural networks without symbolic components do not generalize to longer test inputs. Unlike the existing work, we demonstrate that without model architectures and symbolic components specially designed to improve compositional generalization, least-to-most prompting achieves 99.7% accuracy on any split (including length split) with only a handful of demonstration examples, and it does not require any training or finetuning.

**Easy-to-hard generalization**. In addition to compositional generalization, there are many other tasks where the test cases require more reasoning steps to solve than the training examples, for example, the last-letter-concatenation task where the test lists are longer than the demonstration examples. Dong et al. (2019) propose Neural Logic Machines (NLMs) for both inductive learning and logic reasoning. NLMs trained on small-scale tasks (such as small size block worlds) can perfectly generalize to large-scale tasks (such as larger size block worlds). Schwarzschild et al. (2021) show that recurrent networks trained to solve simple problems with few recurrent steps (such as small size mazes or chess puzzles) can solve more complex problems (such as larger size mazes or chess puzzles) by performing additional recurrences during inference. In our method, we achieve easy-to-hard generalization by decomposing a complex problem into a series of easier problems.

**Task decomposition**. Perez et al. (2020) decompose a multi-hop question into a number of independent single-hop subquestions, which are answered by an off-the-shelf question answering (QA) model. Then those answers are aggregated to form the final answer. Both question decomposition and answer aggregation are implemented by trained models. Wang et al. (2022a) conducts multi-hop QA by modeling prompts as continuous virtual tokens and progressively eliciting relevant knowl-

edge from language models via iterative prompting. Unlike these methods, our approach does not involve any training or finetuning. Moreover, the subquestions generated in least-to-most prompting are usually dependent and have to be sequentially solved in a specific order so that answers to some subquestions can be used as building blocks to solve other subquestions. Yang et al. (2022) translate natural language questions to SQL queries by decomposing a question into a sequence of slot-filling natural language prompts corresponding to SQL clauses via a rule-based system. Wu et al. (2022) propose chaining large language model steps such that the output of one step becomes the input for the next and develop an interactive system for users to construct and modify chains. Least-to-most prompting chains the processes of problem decomposition and subproblem solving.

## 5 LIMITATIONS

Decomposition prompts typically don't generalize well across different domains. For instance, a prompt that demonstrates decomposing math word problems (as seen in Table 9) isn't effective for teaching large language models to break down common sense reasoning problems, such as "Did Aristotle use a laptop?" (Geva et al., 2021). A new prompt must be designed to demonstrate decomposition for these types of problems in order to achieve optimal performance.

Generalizing decomposition can even be difficult within the same domain. We've observed that nearly all problems in GSM8K can be accurately solved if the large language models are provided with the correct decomposition of those challenging problems. This finding isn't surprising and aligns with our experiences in solving math problems. Whenever we successfully break down a math problem into simpler subproblems we can solve, we've essentially solved the original problem. Exceptional results are achieved on the last-letter-concatenation task and the SCAN benchmark because decomposition in these tasks is relatively straightforward.

## 6 CONCLUSION AND DISCUSSION

We introduced least-to-most prompting to enable language models to solve problems that are harder than those in the prompt. This approach entails a two-fold process: a top-down decomposition of the problem and a bottom-up resolution generation. Our empirical findings, which encompass symbolic manipulation, compositional generalization, and mathematical reasoning, reveal that least-to-most prompting significantly surpasses standard prompting and chain-of-thought prompting.

In general, prompting might not be the optimal method for teaching reasoning skills to large language models. Prompting can be viewed as a unidirectional communication form in which we instruct a language model without considering its feedback. A natural progression would be to evolve prompting into fully bidirectional conversations, enabling immediate feedback to language models, thereby facilitating more efficient and effective learning. The least-to-most prompting technique represents a stride towards instructing language models through such bidirectional interactions.

## ACKNOWLEDGEMENT

We sincerely thank Xinyun Chen, Xinying Song, Jeff Dean, Zoubin Ghahramani, Fernando Pereira, Jacob Devlin, and Pete Shaw for sharing their valuable knowledge and advice during our discussions. Their expertise greatly improved the quality of our work. Additionally, we are grateful to the anonymous reviewers for their careful review and helpful suggestions, which helped shape our manuscript into its final form.

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

# Appendix

## Table of Contents

# 7 LAST-LETTER-CONCATENATION

## 7.1 PROMPT CONTEXT FOR DECOMPOSING A WORD LIST INTO SUBPROBLEMS

In Section 3.1 we mentioned that language model prompting can be used to decompose a word list such as "think, machine, learning, reasoning" into a sequence of subproblems "think, machine", "think, machine, learning", and "think, machine, learning, reasoning".

The following prompt context achieves 100% accuracy on this task when using the text-davinci-002 model. Note that it achieves perfect accuracy on lists up to size 12 (which is the maximum that we tested for our experiment) even though it only contains one exemplar each for lists of sizes 2 and 3.

Q: "machine, learning"
A: creating sequential sublists of the list "machine, learning":
"machine"
"machine, learning"

Q: "machine, learning, artificial"
A: creating sequential sublists of the list "machine, learning, artificial":
"machine"
"machine, learning"
"machine, learning, artificial"

## 7.2 PROMPT CONTEXTS WITH MORE AND DIFFERENT EXAMPLES

The last-letter-concatenation experiments presented in Section 3.1 are based on prompt contexts that consists of 2 demonstration examples. To make sure that the accuracy gain achieved by least-to-most prompting is not caused by the slight increase in example length when compared to chain-of-thought, we also performed experiments with more context examples so that we can compare least-to-most vs. chain-of-thought for different prompt sizes. Also, we perform experiments where we use for chain-of-thought prompting the same prompt examples that we use for least-to-most prompting (unlike the situation in Table **??** where we use different examples). All these prompts are shown below, and we present and discuss the corresponding accuracies in Section 7.3.

### 7.2.1 STANDARD PROMPTING, 4-SHOT

Q: "think, machine"
A: "ke"

Q: "learning, reasoning, generalization"
A: "ggn"

Q: "artificial, intelligence"
A: "le"

Q: "transformer, language, vision"
A: "ren"

### 7.2.2 CHAIN-OF-THOUGHT PROMPTING, 4-SHOT

Q: "think, machine"
A: The last letter of "think" is "k". The last letter of "machine" is "e". Concatenating "k", "e" leads to "ke". So, "think, machine" outputs "ke".

Q: "learning, reasoning, generalization"
A: The last letter of "learning" is "g". The last letter of "reasoning" is "g". The last letter of "generalization" is "n". Concatenating "g", "g", "n" leads to "ggn". So, "learning, reasoning, generalization" outputs "ggn".

Q: "artificial, intelligence"
A: The last letter of "artificial" is "l". The last letter of "intelligence" is "e". Concatenating "l", "e" leads to "le". So, "artificial, intelligence" outputs "le".

Q: "transformer, language, vision"
A: The last letter of "transformer" is "r". The last letter of "language" is "e". The last letter of "vision" is "n". Concatenating "r", "e", "n" leads to "ren". So, "transformer, language, vision" outputs "ren".

### 7.2.3 CHAIN-OF-THOUGHT PROMPTING, 8-SHOT

Q: "think, machine"
A: The last letter of "think" is "k". The last letter of "machine" is "e". Concatenating "k", "e" leads to "ke". So, "think, machine" outputs "ke".

Q: "learning, reasoning, generalization"
A: The last letter of "learning" is "g". The last letter of "reasoning" is "g". The last letter of "generalization" is "n". Concatenating "g", "g", "n" leads to "ggn". So, "learning, reasoning, generalization" outputs "ggn".

Q: "artificial, intelligence"
A: The last letter of "artificial" is "l". The last letter of "intelligence" is "e". Concatenating "l", "e" leads to "le". So, "artificial, intelligence" outputs "le".

Q: "transformer, language, vision"
A: The last letter of "transformer" is "r". The last letter of "language" is "e". The last letter of "vision" is "n". Concatenating "r", "e", "n" leads to "ren". So, "transformer, language, vision" outputs "ren".

Q: "school, teacher"
A: The last letter of "school" is "l". The last letter of "teacher" is "r". Concatenating "l", "r" leads to "lr". So, "school, teacher" outputs "lr".

Q: "art, science, department"
A: The last letter of "art" is "t". The last letter of "science" is "e". The last letter of "department" is "t". Concatenating "t", "e", "t" leads to "tet". So, "art, science, department" outputs "tet".

Q: "pen, laptop"
A: The last letter of "pen" is "n". The last letter of "laptop" is "p". Concatenating: "n", "p" leads to "np". So, "pen, laptop" outputs "np".

Q: "book, campus, university"
A: The last letter of "book" is "k". The last letter of "campus" is "s". The last letter of "university" is "y". Concatenating: "k", "s", "y" leads to "ksy". So, "book, campus, university" outputs "ksy".

### 7.2.4 CHAIN-OF-THOUGHT PROMPTING, 2-SHOT, SAME EXAMPLES AS FOR LEAST-TO-MOST

Q: "think, machine"
A: The last letter of "think" is "k". The last letter of "machine" is "e". Concatenating "k", "e" leads to "ke". So, "think, machine" outputs "ke".

Q: "think, machine, learning" A: The last letter of "think" is "k". The last letter of "machine" is "e". The last letter of "learning" is "g". Concatenating "k", "e", "g" leads to "keg". So, "think, machine, learning" outputs "keg".

### 7.2.5 LEAST-TO-MOST PROMPTING, 4-SHOT

Q: "think, machine"
A: The last letter of "think" is "k". The last letter of "machine" is "e". Concatenating "k", "e" leads to

"ke". So, "think, machine" outputs "ke".

Q: "think, machine, learning"
A: "think, machine" outputs "ke". The last letter of "learning" is "g". Concatenating "ke", "g" leads to "keg". So, "think, machine, learning" outputs "keg".

Q: "transformer, language"
A: The last letter of "transformer" is "r". The last letter of "language" is "e". Concatenating: "r", "e" leads to "re". So, "transformer, language" outputs "re".

Q: "transformer, language, vision"
A: "transformer, language" outputs "re". The last letter of "vision" is "n". Concatenating: "re", "n" leads to "ren". So, "transformer, language, vision" outputs "ren".

## 7.3 DATA GENERATION AND ADDITIONAL RESULTS

**Data generation.** The last-letter-concatenation dataset is based on a list of the 10k most common English words (including proper nouns) used in books that are part of project Gutenberg, as collected in Wiktionary[2]. After eliminating profane words, we ended up with a list of 9694 words (all lowercase). For each of the desired list sizes 2, 4, 6, 8, 10, 12, we then generated 500 examples, each of which consists of a random sequence of these words (input) and the corresponding sequence of last letters (output). We will release the full dataset upon publication of this paper. Below are 10 random examples of list size 6:

- IN: "narrative, celebrate, neighbouring, indebted, stove, calling" OUT: "eegdeg"
- IN: "barley, silk, thankful, kiss, logs, silent" OUT: "yklsst"
- IN: "knitting, conveyance, receives, represent, cow, shut" OUT: "gestwt"
- IN: "olive, dark, limitation, airy, pocket, wondered" OUT: "eknytd"
- IN: "apprehensive, exclamation, perspiration, trusting, destiny, tactics" OUT: "enngys"
- IN: "qualified, envoy, disciple, exert, witnesses, plane" OUT: "dyetse"
- IN: "decidedly, dome, france, chris, knowing, peaceful" OUT: "yeesgl"
- IN: "deceit, refinement, tips, cord, princes, discovery" OUT: "ttsdsy"
- IN: "drops, paste, defective, bohemia, requested, convenient" OUT: "seeadt"
- IN: "diverse, christopher, homely, agreeable, fright, suspended" OUT: "eryetd"

**Complete results.** Table 13 summarizes all the experiments we performed for the last-letter-concatenation task. In addition to the experiments where prompt contexts contain 2 demonstration examples presented in Section 3.1, this includes experiments where the prompts contain 4 and 8 demonstration examples (see above).

While more prompt examples have no effect for standard prompting (the accuracy remains at 0), they increase the accuracy across the board for chain-of-thought and least-to-most prompting. However, least-to-most prompting consistently outperforms chain-of-thought prompting. In fact, even if we compare 2-shot least-to-most (prompt size 123 GPT3 tokens) to 8-shot chain-of-thought (prompt size 573 GPT3 tokens), the accuracy for least-to-most prompting is much higher than for chain-of-thought prompting. The difference is especially pronounced for long sequences (e.g., for L = 12, we have least-to-most at 74.0% vs. chain-of-thought at 38.4%). This shows that least-to-most prompting is much more data-efficient than chain-of-thought prompting for this problem.

Comparing the first two rows for chain-of-thought prompting shows that chain-of-thought achieves higher accuracy if we use two independent examples (see prompt in Table **??**) instead of the two dependent examples that we use for least-to-most prompting. This demonstrates that the accuracy advantage of least-to-most prompting over chain-of-thought prompting remains even if the use the same examples for both of them.

---

[2]https://en.wiktionary.org/wiki/Wiktionary:Frequency_lists/PG/2006/04/1-10000

| Prompting method | # Examples | Model | L = 4 | L = 6 | L = 8 | L = 10 | L = 12 |
|---|---|---|---|---|---|---|---|
| Standard | Any | Any | 0.0 | 0.0 | 0.0 | 0.0 | 0.0 |
| Chain-of-Thought | 2 | code-002 | 89.4 | 75.0 | 51.8 | 39.8 | 33.6 |
| | 2 (L2M) | code-002 | 84.2 | 69.2 | 50.2 | 39.8 | 31.8 |
| | 4 | code-002 | 88.6 | 77.0 | 53.4 | 44.0 | 37.4 |
| | 8 | code-002 | 91.0 | 79.8 | 56.8 | 46.8 | 38.4 |
| | 4 | text-002* | 87.0 | 64.0 | 46.0 | 25.0 | 14.0 |
| | 4 | code-001 | 13.0 | 1.8 | 0.0 | 0.0 | 0.0 |
| Least-to-Most | 2 | code-002 | 94.0 | 88.4 | 83.0 | 76.4 | 74.0 |
| | 4 | code-002 | **96.0** | **92.0** | **84.6** | **80.2** | **76.6** |
| | 4 | text-002* | 94.0 | 90.0 | 84.0 | 72.0 | 66.0 |
| | 4 | code-001 | 19.6 | 8.4 | 4.0 | 1.0 | 0.1 |

Table 13: Accuracy of different prompting methods, prompt sizes, and GPT3 models on the last-letter-concatenation task with the length of lists increasing from 4 to 12. We use `code-002` to denote the model `code-davinci-002`, `text-002` to denote the model `text-davinci-002`, and `code-001` to denote the model `code-davinci-001`. The results in the second row for chain-of-thought prompting correspond to the experiment where we use for chain-of-thought the same prompt examples that we use for least-to-most. The results of `text-davinci-002` are based on a subset of 100 random examples (rather than the full set of 500 exammples).

The table also contains the results from running against two additional GPT-3 models: `text-davinci-002` and `codex-davinci-001`. While `text-davinci-002` shows similar accuracy to `code-davinci-002` on small list sizes, the accuracy drops off much faster when moving to larger list sizes, both for chain-of-thought prompting as well as for least-to-most prompting. This indicates that the `code-davinci-002` model has an advantage when it comes to dealing with iteration and recursion.

The `code-davinci-001` model performs much worse than `code-davinci-002` across all dimensions. Even for the shortest list size (L = 4), the accuracy for least-to-most prompting is only 19.6% compared to 96% for `code-davinci-002`. This indicates that there is a large potential for improvement when using the exact same configuration with new model generations.

## 7.4 ERROR ANALYSIS: LEAST-TO-MOST PROMPTING

| Error type | 2 examples | | 4 examples | |
|---|---|---|---|---|
| | L = 4 | L = 12 | L = 4 | L = 12 |
| Concatenation error | 13 | 19 | 21 | 20 |
| - Dropping a letter | 8 | 12 | 15 | 15 |
| - Adding a letter | 4 | 7 | 4 | 3 |
| - Wrong order | 1 | 0 | 2 | 2 |
| Wrong template | 7 | 1 | 0 | 0 |
| Incorrect last letter | 2 | 1 | 1 | 2 |
| Copy error | 0 | 0 | 1 | 0 |

Table 14: Least-to-most prompting error analysis of 20 random failures of the `code-davinci-002` model on list lengths 4 and 12 for prompt contexts consisting of 2 and 4 examples. Note that for some examples, the model made more than one type of error (e.g., dropping and adding a letter during concatenation).

For least-to-most prompting, we analyzed 20 random failures of the `code-davinci-002` model on list lengths 4 and 12 for prompt contexts consisting of 2 and 4 examples. The results are shown in Table 14. Concatenation errors may either be due to dropping a letter, adding a letter or outputting the letters in the wrong order. Wrong template means that the model used the extension template instead of the base template to concatenate the last letter of the first two words of the list. Incorrect last letter means that the model got the last letter of a word wrong, and copy error means that the error was due to making a mistake when copying an intermediate result.

We observe that for the prompt consisting of 2 examples, the fraction of concatenation errors increases as we go from length 4 to length 12 while the fraction of wrong template errors go down. This makes sense because the number of concatenations grows with the length of the list, while the number of times the model needs to use the base template stays constant. Note that the template errors disappear when we move to the double prompt, which means that adding two more examples helps the model recognize which template to use. As a consequence, the double prompt has a similar distribution of errors for both list lengths.

**Examples of concatenation errors.** In the example "gratified, contract, fortitude, blew", the model drops the last letter in the concatenation of "dte" and "w", which means that it predicts the last letter sequence to be "dte" instead of "dtew".

In the example "hollow, supplies, function, gorgeous", the model duplicates the last letter "s" in the concatenation of "wsn" and "s", which means that it predicts the last letter sequence "wsnss" instead of "wsns".

In the example "madly, vengeance, cowardice, monk", the model drops the last letter "k" in the concatenation of "yee" and "k" and instead adds the letter "g". Consequently, the model predicts "yeeg" instead of "yeek".

In the example "slender, lash, throng, scheme", the model breaks the order of the letters "h" and "g" in the concatenation of "rh" and "g", which means that it predicts the last letter sequence "rghe" instead of "rhge".

**Example of incorrect last letter.** In the example "modification, introducing, schools, lunch", the model determines the last letter of the word "modification" to be "g". Consequently, the predicted last letter sequence is "ggsh" instead of "ngsh".

**Example of wrong template application.** In the example "upper, unexpectedly, specifically, connection", the model uses the extension template to determine the output of the first two words "upper, unexpectedly". I.e., it produces:

- "upper" outputs "er". The last letter of "unexpectedly" is "y". Concatenating "er", "y" leads to "ery". So, "upper, unexpectedly" outputs "ery".

when it should have produced:

- The last letter of "upper" is "r". The last letter of "unexpectedly" is "y". Concatenating "r", "y" leads to "ry".

As a consequence, the model predicts the sequence "eryyn" instead of "ryyn".

**Example of copy error.** In the example "displeased, hawk, healing, anchor", the model correctly computes the final concatenation "dkgr" but then makes a mistake when copying the result to the final output and drops the final "r", which leads to the prediction "dkg" instead of "dkgr":

- "displeased, hawk, healing" outputs "dkg". The last letter of "anchor" is "r". Concatenating: "dkg", "r" leads to "dkgr". So, "displeased, hawk, healing, anchor" outputs "dkg".

### 7.5 EXAMPLE OUTPUTS FROM CODE-DAVINCI-002

In this section, we present 5 examples (one example for lengths 4, 6, 8, 10, 12) of successes and failures of the `code-davinci-002` model for standard prompting, chain-of-thought prompting, and least-to-most prompting. They are all based on the prompt contexts containing 4 demonstration examples.

#### 7.5.1 STANDARD PROMPTING: FAILURE

**"witnesses, arkansas, remaining, privileged"**

**Golden:**
"ssgd"

**Solution:**
Q: "witnesses, arkansas, remaining, privileged"
A: "n"

---

**"clan, noun, roy, nicholas, relish, liking"**

**Golden:**
"nnyshg"

**Solution:**
Q: "clan, noun, roy, nicholas, relish, liking"
A: "nicholas"

---

**"franklin, temporarily, high, origin, rapidly, diversion, firmament, promptly"**

**Golden:**
"nyhnynty"

**Solution:**
Q: "franklin, temporarily, high, origin, rapidly, diversion, firmament, promptly"
A: "n"

---

**"conquered, owed, enclosure, indeed, shattered, beads, increased, provided, weak, grind"**

**Golden:**
"ddeddsddkd"

**Solution:**
Q: "conquered, owed, enclosure, indeed, shattered, beads, increased, provided, weak, grind"
A: "d"

---

**"hope, alison, mirrors, benefits, altered, curve, keen, portuguese, distance, interfaces, proof, seamus"**

**Golden:**
"enssdeneesfs"

**Solution:**
Q: "hope, alison, mirrors, benefits, altered, curve, keen, portuguese, distance, interfaces, proof, seamus"
A: "hope"

### 7.5.2 CHAIN-OF-THOUGHT PROMPTING: SUCCESS

**"geese, rogers, benefactor, version"**

**Golden:**
"esrn"

**Solution:**
Q: "geese, rogers, benefactor, version"
A: The last letter of "geese" is "e". The last letter of "rogers" is "s". The last letter of "benefactor" is "r". The last letter of "version" is "n". Concatenating "e", "s", "r", "n" leads to "esrn". So, "geese, rogers,

benefactor, version" outputs "esrn".

---

**"amply, bestow, maryland, edited, blowing, prosperous"**

**Golden:**
"ywddgs"

**Solution:**
Q: "amply, bestow, maryland, edited, blowing, prosperous"
A: The last letter of "amply" is "y". The last letter of "bestow" is "w". The last letter of "maryland" is "d". The last letter of "edited" is "d". The last letter of "blowing" is "g". The last letter of "prosperous" is "s". Concatenating "y", "w", "d", "d", "g", "s" leads to "ywddgs". So, "amply, bestow, maryland, edited, blowing, prosperous" outputs "ywddgs".

---

**"needless, courts, reflected, curly, glowing, asp, chronicles, anthony"**

**Golden:**
"ssdygpsy"

**Solution:**
Q: "needless, courts, reflected, curly, glowing, asp, chronicles, anthony"
A: The last letter of "needless" is "s". The last letter of "courts" is "s". The last letter of "reflected" is "d". The last letter of "curly" is "y". The last letter of "glowing" is "g". The last letter of "asp" is "p". The last letter of "chronicles" is "s". The last letter of "anthony" is "y". Concatenating "s", "s", "d", "y", "g", "p", "s", "y" leads to "ssdygpsy". So, "needless, courts, reflected, curly, glowing, asp, chronicles, anthony" outputs "ssdygpsy".

---

**"nigel, alaska, rent, hospital, newspaper, medical, nation, dave, associated, rumour"**

**Golden:**
"latlrlnedr"

**Solution:**
Q: "nigel, alaska, rent, hospital, newspaper, medical, nation, dave, associated, rumour"
A: The last letter of "nigel" is "l". The last letter of "alaska" is "a". The last letter of "rent" is "t". The last letter of "hospital" is "l". The last letter of "newspaper" is "r". The last letter of "medical" is "l". The last letter of "nation" is "n". The last letter of "dave" is "e". The last letter of "associated" is "d". The last letter of "rumour" is "r". Concatenating "l", "a", "t", "l", "r", "l", "n", "e", "d", "r" leads to "latlrlnedr". So, "nigel, alaska, rent, hospital, newspaper, medical, nation, dave, associated, rumour" outputs "latlrlnedr".

---

**"brightness, preparatory, intense, accumulated, reluctantly, nets, inquiries, grip, inquisition, alley, innocent, herbert"**

**Golden:**
"syedysspnytt"

**Solution:**
Q: "brightness, preparatory, intense, accumulated, reluctantly, nets, inquiries, grip, inquisition, alley, innocent, herbert"
A: The last letter of "brightness" is "s". The last letter of "preparatory" is "y". The last letter of "intense" is "e". The last letter of "accumulated" is "d". The last letter of "reluctantly" is "y". The last letter of "nets" is "s". The last letter of "inquiries" is "s". The last letter of "grip" is "p". The last letter of "inquisition" is "n". The last letter of "alley" is "y". The last letter of "innocent" is "t". The last letter of

"herbert" is "t". Concatenating "s", "y", "e", "d", "y", "s", "s", "p", "n", "y", "t", "t" leads to "syedysspnytt". So, "brightness, preparatory, intense, accumulated, reluctantly, nets, inquiries, grip, inquisition, alley, innocent, herbert" outputs "syedysspnytt".

### 7.5.3 CHAIN-OF-THOUGHT PROMPTING: FAILURE

**"pattern, vote, afterwards, solemn"**

**Golden:**
"nesn"

**Solution:**
Q: "pattern, vote, afterwards, solemn"
A: The last letter of "pattern" is "n". The last letter of "vote" is "e". The last letter of "afterwards" is "s". The last letter of "solemn" is "n". Concatenating "n", "e", "s", "n" leads to "ness". So, "pattern, vote, afterwards, solemn" outputs "ness".

---

**"disappointment, troublesome, respectfully, monstrous, worry, politics"**

**Golden:**
"teysys"

**Solution:**
Q: "disappointment, troublesome, respectfully, monstrous, worry, politics"
A: The last letter of "disappointment" is "t". The last letter of "troublesome" is "e". The last letter of "respectfully" is "y". The last letter of "monstrous" is "s". The last letter of "worry" is "y". The last letter of "politics" is "s". Concatenating "t", "e", "y", "s", "y", "s" leads to "teyssy". So, "disappointment, troublesome, respectfully, monstrous, worry, politics" outputs "teyssy".

---

**"drawing, changes, purified, basic, others, send, chimney, vice"**

**Golden:**
"gsdcsdye"

**Solution:**
Q: "drawing, changes, purified, basic, others, send, chimney, vice"
A: The last letter of "drawing" is "g". The last letter of "changes" is "s". The last letter of "purified" is "d". The last letter of "basic" is "c". The last letter of "others" is "s". The last letter of "send" is "d". The last letter of "chimney" is "y". The last letter of "vice" is "e". Concatenating "g", "s", "d", "c", "s", "d", "y", "e" leads to "gsdcsdye". So, "drawing, changes, purified, basic, others, send, chimney, vice" outputs "gsdcsyde".

---

**"sensitive, pray, wreath, betwixt, golden, regions, loyalty, interval, hunting, ranch"**

**Golden:**
"eyhtnsylgh"

**Solution:**
Q: "sensitive, pray, wreath, betwixt, golden, regions, loyalty, interval, hunting, ranch"
A: The last letter of "sensitive" is "e". The last letter of "pray" is "y". The last letter of "wreath" is "h". The last letter of "betwixt" is "t". The last letter of "golden" is "n". The last letter of "regions" is "s". The last letter of "loyalty" is "y". The last letter of "interval" is "l". The last letter of "hunting" is "g". The last letter of "ranch" is "h". Concatenating "e", "y", "h", "t", "n", "s", "y", "l", "g", "h" leads to "eyhtsynslgh". So, "sensitive, pray, wreath, betwixt, golden, regions, loyalty, interval, hunting, ranch" outputs "eyhtsynslgh".

**"freezing, arduous, pushing, vitality, hue, generosity, white, motto, matthew, preferred, hounds, breast"**

**Golden:**
"gsgyeyeowdst"

**Solution:**
Q: "freezing, arduous, pushing, vitality, hue, generosity, white, motto, matthew, preferred, hounds, breast"
A: The last letter of "freezing" is "g". The last letter of "arduous" is "s". The last letter of "pushing" is "g". The last letter of "vitality" is "y". The last letter of "hue" is "e". The last letter of "generosity" is "y". The last letter of "white" is "e". The last letter of "motto" is "o". The last letter of "matthew" is "w". The last letter of "preferred" is "d". The last letter of "hounds" is "s". The last letter of "breast" is "t". Concatenating "g", "s", "g", "y", "e", "y", "e", "o", "w", "d", "s", "t" leads to "gsgyeyowdst". So, "freezing, arduous, pushing, vitality, hue, generosity, white, motto, matthew, preferred, hounds, breast" outputs "gsgyeyowdst".

### 7.5.4 Least-to-most prompting: Success

**"worm, jackson, widow, car"**

**Golden:**
"mnwr"

**Solution:**
Q: "worm, jackson"
A: The last letter of "worm" is "m". The last letter of "jackson" is "n". Concatenating: "m", "n" leads to "mn". So, "worm, jackson" outputs "mn".

Q: "worm, jackson, widow"
A: "worm, jackson" outputs "mn". The last letter of "widow" is "w". Concatenating: "mn", "w" leads to "mnw". So, "worm, jackson, widow" outputs "mnw".

Q: "worm, jackson, widow, car"
A: "worm, jackson, widow" outputs "mnw". The last letter of "car" is "r". Concatenating: "mnw", "r" leads to "mnwr". So, "worm, jackson, widow, car" outputs "mnwr".

**"twinkle, blushed, pale, fast, irony, tearing"**

**Golden:**
"edetyg"

**Solution:**
Q: "twinkle, blushed"
A: The last letter of "twinkle" is "e". The last letter of "blushed" is "d". Concatenating: "e", "d" leads to "ed". So, "twinkle, blushed" outputs "ed".

Q: "twinkle, blushed, pale"
A: "twinkle, blushed" outputs "ed". The last letter of "pale" is "e". Concatenating: "ed", "e" leads to "ede". So, "twinkle, blushed, pale" outputs "ede".

Q: "twinkle, blushed, pale, fast"
A: "twinkle, blushed, pale" outputs "ede". The last letter of "fast" is "t". Concatenating: "ede", "t" leads to "edet". So, "twinkle, blushed, pale, fast" outputs "edet".

Q: "twinkle, blushed, pale, fast, irony"

A: "twinkle, blushed, pale, fast" outputs "edet". The last letter of "irony" is "y". Concatenating: "edet", "y" leads to "edety". So, "twinkle, blushed, pale, fast, irony" outputs "edety".

Q: "twinkle, blushed, pale, fast, irony, tearing"
A: "twinkle, blushed, pale, fast, irony" outputs "edety". The last letter of "tearing" is "g". Concatenating: "edety", "g" leads to "edetyg". So, "twinkle, blushed, pale, fast, irony, tearing" outputs "edetyg".

---

**"colonists, empty, fowl, intend, diligent, reluctantly, seed, gentry"**

**Golden:**
"syldtydy"

**Solution:**
Q: "colonists, empty"
A: The last letter of "colonists" is "s". The last letter of "empty" is "y". Concatenating: "s", "y" leads to "sy". So, "colonists, empty" outputs "sy".

Q: "colonists, empty, fowl"
A: "colonists, empty" outputs "sy". The last letter of "fowl" is "l". Concatenating: "sy", "l" leads to "syl". So, "colonists, empty, fowl" outputs "syl".

Q: "colonists, empty, fowl, intend"
A: "colonists, empty, fowl" outputs "syl". The last letter of "intend" is "d". Concatenating: "syl", "d" leads to "syld". So, "colonists, empty, fowl, intend" outputs "syld".

Q: "colonists, empty, fowl, intend, diligent"
A: "colonists, empty, fowl, intend" outputs "syld". The last letter of "diligent" is "t". Concatenating: "syld", "t" leads to "syldt". So, "colonists, empty, fowl, intend, diligent" outputs "syldt".

Q: "colonists, empty, fowl, intend, diligent, reluctantly"
A: "colonists, empty, fowl, intend, diligent" outputs "syldt". The last letter of "reluctantly" is "y". Concatenating: "syldt", "y" leads to "syldty". So, "colonists, empty, fowl, intend, diligent, reluctantly" outputs "syldty".

Q: "colonists, empty, fowl, intend, diligent, reluctantly, seed"
A: "colonists, empty, fowl, intend, diligent, reluctantly" outputs "syldty". The last letter of "seed" is "d". Concatenating: "syldty", "d" leads to "syldtyd". So, "colonists, empty, fowl, intend, diligent, reluctantly, seed" outputs "syldtyd".

Q: "colonists, empty, fowl, intend, diligent, reluctantly, seed, gentry"
A: "colonists, empty, fowl, intend, diligent, reluctantly, seed" outputs "syldtyd". The last letter of "gentry" is "y". Concatenating: "syldtyd", "y" leads to "syldtydy". So, "colonists, empty, fowl, intend, diligent, reluctantly, seed, gentry" outputs "syldtydy".

---

**"leo, observations, power, thou, dart, emergency, notorious, indulging, consume, provoke"**

**Golden:**
"osrutysgee"

**Solution:**
Q: "leo, observations"
A: The last letter of "leo" is "o". The last letter of "observations" is "s". Concatenating: "o", "s" leads to "os". So, "leo, observations" outputs "os".

Q: "leo, observations, power"
A: "leo, observations" outputs "os". The last letter of "power" is "r". Concatenating: "os", "r" leads to "osr". So, "leo, observations, power" outputs "osr".

Q: "leo, observations, power, thou"
A: "leo, observations, power" outputs "osr". The last letter of "thou" is "u". Concatenating: "osr", "u"

leads to "osru". So, "leo, observations, power, thou" outputs "osru".

Q: "leo, observations, power, thou, dart"
A: "leo, observations, power, thou" outputs "osru". The last letter of "dart" is "t". Concatenating: "osru", "t" leads to "osrut". So, "leo, observations, power, thou, dart" outputs "osrut".

Q: "leo, observations, power, thou, dart, emergency"
A: "leo, observations, power, thou, dart" outputs "osrut". The last letter of "emergency" is "y". Concatenating: "osrut", "y" leads to "osruty". So, "leo, observations, power, thou, dart, emergency" outputs "osruty".

Q: "leo, observations, power, thou, dart, emergency, notorious"
A: "leo, observations, power, thou, dart, emergency" outputs "osruty". The last letter of "notorious" is "s". Concatenating: "osruty", "s" leads to "osrutys". So, "leo, observations, power, thou, dart, emergency, notorious" outputs "osrutys".

Q: "leo, observations, power, thou, dart, emergency, notorious, indulging"
A: "leo, observations, power, thou, dart, emergency, notorious" outputs "osrutys". The last letter of "indulging" is "g". Concatenating: "osrutys", "g" leads to "osrutysg". So, "leo, observations, power, thou, dart, emergency, notorious, indulging" outputs "osrutysg".

Q: "leo, observations, power, thou, dart, emergency, notorious, indulging, consume"
A: "leo, observations, power, thou, dart, emergency, notorious, indulging" outputs "osrutysg". The last letter of "consume" is "e". Concatenating: "osrutysg", "e" leads to "osrutysge". So, "leo, observations, power, thou, dart, emergency, notorious, indulging, consume" outputs "osrutysge".

Q: "leo, observations, power, thou, dart, emergency, notorious, indulging, consume, provoke"
A: "leo, observations, power, thou, dart, emergency, notorious, indulging, consume" outputs "osrutysge". The last letter of "provoke" is "e". Concatenating: "osrutysge", "e" leads to "osrutysgee". So, "leo, observations, power, thou, dart, emergency, notorious, indulging, consume, provoke" outputs "osrutysgee".

---

**"niece, greedy, anything, retain, moisture, jest, adrian, attic, hugo, accurate, wet, howard"**

**Golden:**
"eygnetncoetd"

**Solution:**
Q: "niece, greedy"
A: The last letter of "niece" is "e". The last letter of "greedy" is "y". Concatenating: "e", "y" leads to "ey". So, "niece, greedy" outputs "ey".

Q: "niece, greedy, anything"
A: "niece, greedy" outputs "ey". The last letter of "anything" is "g". Concatenating: "ey", "g" leads to "eyg". So, "niece, greedy, anything" outputs "eyg".

Q: "niece, greedy, anything, retain"
A: "niece, greedy, anything" outputs "eyg". The last letter of "retain" is "n". Concatenating: "eyg", "n" leads to "eygn". So, "niece, greedy, anything, retain" outputs "eygn".

Q: "niece, greedy, anything, retain, moisture"
A: "niece, greedy, anything, retain" outputs "eygn". The last letter of "moisture" is "e". Concatenating: "eygn", "e" leads to "eygne". So, "niece, greedy, anything, retain, moisture" outputs "eygne".

Q: "niece, greedy, anything, retain, moisture, jest"
A: "niece, greedy, anything, retain, moisture" outputs "eygne". The last letter of "jest" is "t". Concatenating: "eygne", "t" leads to "eygnet". So, "niece, greedy, anything, retain, moisture, jest" outputs "eygnet".

Q: "niece, greedy, anything, retain, moisture, jest, adrian"

A: "niece, greedy, anything, retain, moisture, jest" outputs "eygnet". The last letter of "adrian" is "n". Concatenating: "eygnet", "n" leads to "eygnetn". So, "niece, greedy, anything, retain, moisture, jest, adrian" outputs "eygnetn".

Q: "niece, greedy, anything, retain, moisture, jest, adrian, attic"
A: "niece, greedy, anything, retain, moisture, jest, adrian" outputs "eygnetn". The last letter of "attic" is "c". Concatenating: "eygnetn", "c" leads to "eygnetnc". So, "niece, greedy, anything, retain, moisture, jest, adrian, attic" outputs "eygnetnc".

Q: "niece, greedy, anything, retain, moisture, jest, adrian, attic, hugo"
A: "niece, greedy, anything, retain, moisture, jest, adrian, attic" outputs "eygnetnc". The last letter of "hugo" is "o". Concatenating: "eygnetnc", "o" leads to "eygnetnco". So, "niece, greedy, anything, retain, moisture, jest, adrian, attic, hugo" outputs "eygnetnco".

Q: "niece, greedy, anything, retain, moisture, jest, adrian, attic, hugo, accurate"
A: "niece, greedy, anything, retain, moisture, jest, adrian, attic, hugo" outputs "eygnetnco". The last letter of "accurate" is "e". Concatenating: "eygnetnco", "e" leads to "eygnetncoe". So, "niece, greedy, anything, retain, moisture, jest, adrian, attic, hugo, accurate" outputs "eygnetncoe".

Q: "niece, greedy, anything, retain, moisture, jest, adrian, attic, hugo, accurate, wet"
A: "niece, greedy, anything, retain, moisture, jest, adrian, attic, hugo, accurate" outputs "eygnetncoe". The last letter of "wet" is "t". Concatenating: "eygnetncoe", "t" leads to "eygnetncoet". So, "niece, greedy, anything, retain, moisture, jest, adrian, attic, hugo, accurate, wet" outputs "eygnetncoet".

Q: "niece, greedy, anything, retain, moisture, jest, adrian, attic, hugo, accurate, wet, howard"
A: "niece, greedy, anything, retain, moisture, jest, adrian, attic, hugo, accurate, wet" outputs "eygnetncoet". The last letter of "howard" is "d". Concatenating: "eygnetncoet", "d" leads to "eygnetncoetd". So, "niece, greedy, anything, retain, moisture, jest, adrian, attic, hugo, accurate, wet, howard" outputs "eygnetncoetd".

### 7.5.5 LEAST-TO-MOST PROMPTING: FAILURE

**"supper, procession, region, ruby"**

**Golden:**
"rnny"

**Solution:**
Q: "supper, procession"
A: The last letter of "supper" is "r". The last letter of "procession" is "n". Concatenating: "r", "n" leads to "rn". So, "supper, procession" outputs "rn".

Q: "supper, procession, region"
A: "supper, procession" outputs "rn". The last letter of "region" is "n". Concatenating: "rn", "n" leads to "rnn". So, "supper, procession, region" outputs "rnn".

Q: "supper, procession, region, ruby"
A: "supper, procession, region" outputs "rnn". The last letter of "ruby" is "y". Concatenating: "rnn", "y" leads to "rnnn". So, "supper, procession, region, ruby" outputs "rnnn".

---

**"release, appeared, endure, field, render, monk"**

**Golden:**
"ededrk"

**Solution:**
Q: "release, appeared"
A: The last letter of "release" is "e". The last letter of "appeared" is "d". Concatenating: "e", "d" leads

to "ed". So, "release, appeared" outputs "ed".

Q: "release, appeared, endure"
A: "release, appeared" outputs "ed". The last letter of "endure" is "e". Concatenating: "ed", "e" leads to "ede". So, "release, appeared, endure" outputs "ede".

Q: "release, appeared, endure, field"
A: "release, appeared, endure" outputs "ede". The last letter of "field" is "d". Concatenating: "ede", "d" leads to "eded". So, "release, appeared, endure, field" outputs "eded".

Q: "release, appeared, endure, field, render"
A: "release, appeared, endure, field" outputs "eded". The last letter of "render" is "r". Concatenating: "eded", "r" leads to "ededer". So, "release, appeared, endure, field, render" outputs "ededer".

Q: "release, appeared, endure, field, render, monk"
A: "release, appeared, endure, field, render" outputs "ededer". The last letter of "monk" is "k". Concatenating: "ededer", "k" leads to "edederk". So, "release, appeared, endure, field, render, monk" outputs "edederk".

---

**"mentally, uncomfortable, amount, straight, whatsoever, deference, channel, integrity"**

**Golden:**
"yettrely"

**Solution:**
Q: "mentally, uncomfortable"
A: The last letter of "mentally" is "y". The last letter of "uncomfortable" is "e". Concatenating: "y", "e" leads to "ye". So, "mentally, uncomfortable" outputs "ye".

Q: "mentally, uncomfortable, amount"
A: "mentally, uncomfortable" outputs "ye". The last letter of "amount" is "t". Concatenating: "ye", "t" leads to "yet". So, "mentally, uncomfortable, amount" outputs "yet".

Q: "mentally, uncomfortable, amount, straight"
A: "mentally, uncomfortable, amount" outputs "yet". The last letter of "straight" is "t". Concatenating: "yet", "t" leads to "yett". So, "mentally, uncomfortable, amount, straight" outputs "yett".

Q: "mentally, uncomfortable, amount, straight, whatsoever"
A: "mentally, uncomfortable, amount, straight" outputs "yett". The last letter of "whatsoever" is "r". Concatenating: "yett", "r" leads to "yettar". So, "mentally, uncomfortable, amount, straight, whatsoever" outputs "yettar".

Q: "mentally, uncomfortable, amount, straight, whatsoever, deference"
A: "mentally, uncomfortable, amount, straight, whatsoever" outputs "yettar". The last letter of "deference" is "e". Concatenating: "yettar", "e" leads to "yettare". So, "mentally, uncomfortable, amount, straight, whatsoever, deference" outputs "yettare".

Q: "mentally, uncomfortable, amount, straight, whatsoever, deference, channel"
A: "mentally, uncomfortable, amount, straight, whatsoever, deference" outputs "yettare". The last letter of "channel" is "l". Concatenating: "yettare", "l" leads to "yettarel". So, "mentally, uncomfortable, amount, straight, whatsoever, deference, channel" outputs "yettarel".

Q: "mentally, uncomfortable, amount, straight, whatsoever, deference, channel, integrity"
A: "mentally, uncomfortable, amount, straight, whatsoever, deference, channel" outputs "yettarel". The last letter of "integrity" is "y". Concatenating: "yettarel", "y" leads to "yettarely". So, "mentally, uncomfortable, amount, straight, whatsoever, deference, channel, integrity" outputs "yettarely".

---

**"complexion, criminal, assume, seven, driver, hero, horse, purpose, zeus, joint"**

**Golden:**

"nlenroeest"

**Solution:**
Q: "complexion, criminal"
A: The last letter of "complexion" is "n". The last letter of "criminal" is "l". Concatenating: "n", "l" leads to "nl". So, "complexion, criminal" outputs "nl".

Q: "complexion, criminal, assume"
A: "complexion, criminal" outputs "nl". The last letter of "assume" is "e". Concatenating: "nl", "e" leads to "nle". So, "complexion, criminal, assume" outputs "nle".

Q: "complexion, criminal, assume, seven"
A: "complexion, criminal, assume" outputs "nle". The last letter of "seven" is "n". Concatenating: "nle", "n" leads to "nlen". So, "complexion, criminal, assume, seven" outputs "nlen".

Q: "complexion, criminal, assume, seven, driver"
A: "complexion, criminal, assume, seven" outputs "nlen". The last letter of "driver" is "r". Concatenating: "nlen", "r" leads to "nlenr". So, "complexion, criminal, assume, seven, driver" outputs "nlenr".

Q: "complexion, criminal, assume, seven, driver, hero"
A: "complexion, criminal, assume, seven, driver" outputs "nlenr". The last letter of "hero" is "o". Concatenating: "nlenr", "o" leads to "nlenro". So, "complexion, criminal, assume, seven, driver, hero" outputs "nlenro".

Q: "complexion, criminal, assume, seven, driver, hero, horse"
A: "complexion, criminal, assume, seven, driver, hero" outputs "nlenro". The last letter of "horse" is "e". Concatenating: "nlenro", "e" leads to "nlenroe". So, "complexion, criminal, assume, seven, driver, hero, horse" outputs "nlenroe".

Q: "complexion, criminal, assume, seven, driver, hero, horse, purpose"
A: "complexion, criminal, assume, seven, driver, hero, horse" outputs "nlenroe". The last letter of "purpose" is "e". Concatenating: "nlenroe", "e" leads to "nlenroee". So, "complexion, criminal, assume, seven, driver, hero, horse, purpose" outputs "nlenroee".

Q: "complexion, criminal, assume, seven, driver, hero, horse, purpose, zeus"
A: "complexion, criminal, assume, seven, driver, hero, horse, purpose" outputs "nlenroee". The last letter of "zeus" is "s". Concatenating: "nlenroee", "s" leads to "nlenroees". So, "complexion, criminal, assume, seven, driver, hero, horse, purpose, zeus" outputs "nlenroees".

Q: "complexion, criminal, assume, seven, driver, hero, horse, purpose, zeus, joint"
A: "complexion, criminal, assume, seven, driver, hero, horse, purpose, zeus" outputs "nlenroees". The last letter of "joint" is "t". Concatenating: "nlenroees", "t" leads to "nlenroeast". So, "complexion, criminal, assume, seven, driver, hero, horse, purpose, zeus, joint" outputs "nlenroeast".

---

**"provoked, columbia, sir, principal, seven, extreme, timothy, submitted, considered, spies, dost, treasury"**

**Golden:**
"darlneyddsty"

**Solution:**
Q: "provoked, columbia"
A: The last letter of "provoked" is "d". The last letter of "columbia" is "a". Concatenating: "d", "a" leads to "da". So, "provoked, columbia" outputs "da".

Q: "provoked, columbia, sir"
A: "provoked, columbia" outputs "da". The last letter of "sir" is "r". Concatenating: "da", "r" leads to "dar". So, "provoked, columbia, sir" outputs "dar".

Q: "provoked, columbia, sir, principal"
A: "provoked, columbia, sir" outputs "dar". The last letter of "principal" is "l". Concatenating: "dar", "l"

leads to "darl". So, "provoked, columbia, sir, principal" outputs "darl".

Q: "provoked, columbia, sir, principal, seven"
A: "provoked, columbia, sir, principal" outputs "darl". The last letter of "seven" is "n". Concatenating: "darl", "n" leads to "darln". So, "provoked, columbia, sir, principal, seven" outputs "darln".

Q: "provoked, columbia, sir, principal, seven, extreme"
A: "provoked, columbia, sir, principal, seven" outputs "darln". The last letter of "extreme" is "e". Concatenating: "darln", "e" leads to "darle". So, "provoked, columbia, sir, principal, seven, extreme" outputs "darle".

Q: "provoked, columbia, sir, principal, seven, extreme, timothy"
A: "provoked, columbia, sir, principal, seven, extreme" outputs "darle". The last letter of "timothy" is "y". Concatenating: "darle", "y" leads to "darley". So, "provoked, columbia, sir, principal, seven, extreme, timothy" outputs "darley".

Q: "provoked, columbia, sir, principal, seven, extreme, timothy, submitted"
A: "provoked, columbia, sir, principal, seven, extreme, timothy" outputs "darley". The last letter of "submitted" is "d". Concatenating: "darley", "d" leads to "darleyd". So, "provoked, columbia, sir, principal, seven, extreme, timothy, submitted" outputs "darleyd".

Q: "provoked, columbia, sir, principal, seven, extreme, timothy, submitted, considered"
A: "provoked, columbia, sir, principal, seven, extreme, timothy, submitted" outputs "darleyd". The last letter of "considered" is "d". Concatenating: "darleyd", "d" leads to "darleydd". So, "provoked, columbia, sir, principal, seven, extreme, timothy, submitted, considered" outputs "darleydd".

Q: "provoked, columbia, sir, principal, seven, extreme, timothy, submitted, considered, spies"
A: "provoked, columbia, sir, principal, seven, extreme, timothy, submitted, considered" outputs "darleydd". The last letter of "spies" is "s". Concatenating: "darleydd", "s" leads to "darleydds". So, "provoked, columbia, sir, principal, seven, extreme, timothy, submitted, considered, spies" outputs "darleydds".

Q: "provoked, columbia, sir, principal, seven, extreme, timothy, submitted, considered, spies, dost"
A: "provoked, columbia, sir, principal, seven, extreme, timothy, submitted, considered, spies" outputs "darleydds". The last letter of "dost" is "t". Concatenating: "darleydds", "t" leads to "darleyddst". So, "provoked, columbia, sir, principal, seven, extreme, timothy, submitted, considered, spies, dost" outputs "darleyddst".

Q: "provoked, columbia, sir, principal, seven, extreme, timothy, submitted, considered, spies, dost, treasury"
A: "provoked, columbia, sir, principal, seven, extreme, timothy, submitted, considered, spies, dost" outputs "darleyddst". The last letter of "treasury" is "y". Concatenating: "darleyddst", "y" leads to "darleyddsty". So, "provoked, columbia, sir, principal, seven, extreme, timothy, submitted, considered, spies, dost, treasury" outputs "darleyddsty".

# 8 SCAN

## 8.1 PROMPT CONTEXTS

In this section we present the prompt contexts used for the SCAN benchmark in Section 3.2. It includes one context for each of standard prompting, least-to-most prompting, and chain-of-thought prompting.

### 8.1.1 STANDARD PROMPTING

The context for standard prompting consist of a set of commands together with the corresponding action sequences.

Q: "turn left"
A: "TURN_LEFT"

Q: "turn right"
A: "TURN_RIGHT"

Q: "jump left"
A: "TURN_LEFT" + "JUMP"

Q: "run right"
A: "TURN_RIGHT" + "RUN"

Q: "look twice"
A: "LOOK" * 2

Q: "run and look twice"
A: "RUN" + "LOOK" * 2

Q: "jump right thrice"
A: ("TURN_RIGHT" + "JUMP") * 3

Q: "walk after run"
A: "RUN" + "WALK"

Q: "turn opposite left"
A: "TURN_LEFT" * 2

Q: "turn around left"
A: "TURN_LEFT" * 4

Q: "turn opposite right"
A: "TURN_RIGHT" * 2

Q: "turn around right"
A: "TURN_RIGHT" * 4

Q: "walk opposite left"
A: "TURN_LEFT" * 2 + "WALK"

Q: "walk around left"
A: ("TURN_LEFT" + "WALK") * 4

### 8.1.2 LEAST-TO-MOST PROMPTING

For least-to-most prompting, we need 2 different prompt contexts: one to decompose commands into a sequence of components and one to map commands to actions.

**Command mapping.** The command mapping prompt context consists of a set of commands together with the corresponding action sequences and explanations of how they were obtained.

Q: "turn left"
A: "turn left" outputs "TURN_LEFT".

Q: "turn right"
A: "turn right" outputs "TURN_RIGHT".

Q: "jump left"
A: The output of "jump left" concatenates: the output of "turn left", the output of "jump". "turn left" outputs "TURN_LEFT". "jump" outputs "JUMP". So concatenating the output of "turn left" and the output of "jump" leads to "TURN_LEFT" + "JUMP". So the output of "jump left" is "TURN_LEFT" + "JUMP".

Q: "run right"
A: The output of "run right" concatenates: the output of "turn right", the output of "run". "turn right" outputs "TURN_RIGHT". "run" outputs "RUN". So concatenating the output of "turn right" and the output of "run" leads to "TURN_RIGHT" + "RUN". So the output of "run right" is "TURN_RIGHT" + "RUN".

Q: "look twice"
A: The output of "look twice" concatenates: the output of "look", the output of "look". "look" outputs "LOOK". So repeating the output of "look" two times leads to "LOOK" * 2. So the output of "look twice" is "LOOK" * 2.

Q: "run and look twice"
A: The output of "run and look twice" concatenates: the output of "run", the output of "look twice". "run" outputs "RUN". "look twice" outputs "LOOK" * 2. So concatenating the output of "run" and the output of "look twice" leads to "RUN" + "LOOK" * 2. So the output of "run and look twice" is "RUN" + "LOOK" * 2.

Q: "jump right thrice"
A: The output of "jump right thrice" concatenates: the output of "jump right", the output of "jump right", the output of "jump right". "jump right" outputs "TURN_RIGHT" + "JUMP". So repeating the output of "jump right" three times leads to ("TURN_RIGHT" + "JUMP") * 3. So the output of "jump right thrice" is ("TURN_RIGHT" + "JUMP") * 3.

Q: "walk after run"
A: The output of "walk after run" concatenates: the output of "run", the output of "walk". "run" outputs "RUN". "walk" outputs "WALK". So concatenating the output of "run" and the output of "walk" leads to "RUN" + "WALK". So the output of "walk after run" is "RUN" + "WALK".

Q: "turn opposite left"
A: The output of "turn opposite left" concatenates: the output of "turn left", the output of "turn left". "turn left" outputs "TURN_LEFT". So repeating the output of "turn left" twice leads to "TURN_LEFT" * 2. So the output of "turn opposite left" is "TURN_LEFT" * 2.

Q: "turn around left"
A: The output of "turn around left" concatenates: the output of "turn left", the output of "turn left", the output of "turn left", the output of "turn left". "turn left" outputs "TURN_LEFT". So repeating the output of "turn left" four times leads to "TURN_LEFT" * 4. So the output of "turn around left" is "TURN_LEFT" * 4.

Q: "turn opposite right"
A: The output of "turn opposite right" concatenates: the output of "turn right", the output of "turn right". "turn right" outputs "TURN_RIGHT". So repeating the output of "turn right" twice leads to "TURN_RIGHT" * 2. So the output of "turn opposite right" is "TURN_RIGHT" * 2.

Q: "turn around right"
A: The output of "turn around right" concatenates: the output of "turn right", the output of "turn right", the output of "turn right", the output of "turn right". "turn right" outputs "TURN_RIGHT". So repeating the output of "turn right" four times leads to "TURN_RIGHT" * 4. So the output of "turn around right" is "TURN_RIGHT" * 4.

Q: "walk opposite left"
A: The output of "walk opposite left" concatenates: the output of "turn opposite left", the output of "walk". "turn opposite left" outputs "TURN_LEFT" * 2. "walk" outputs "WALK". So concatenating the output of "turn opposite left" and the output of "walk" leads to "TURN_LEFT" * 2 + "WALK". So the

output of "walk opposite left" is "TURN_LEFT" * 2 + "WALK".

Q: "walk around left"
A: The output of "walk around left" concatenates: the output of "walk left", the output of "walk left", the output of "walk left", the output of "walk left". "walk left" outputs "TURN_LEFT" + "WALK". So repeating the output of "walk around left" four times leads to ("TURN_LEFT" + "WALK") * 4. So the output of "walk around left" is ("TURN_LEFT" + "WALK") * 4.

**Command decomposition.** The command decomposition prompt context consists of a set of commands together with the corresponding sequence of decomposed commands.

Q: "look right after look twice"
A: "look right after look twice" can be solved by: "look right", "look twice".

Q: "jump opposite right thrice and walk"
A: "jump opposite right thrice" can be solved by: "jump opposite right", "jump opposite right thrice". "walk" can be solved by: "walk". So, "jump opposite right thrice and walk" can be solved by: "jump opposite right", "jump opposite right thrice", "walk".

Q: "run left twice and run right"
A: "run left twice" can be solved by: "run left", "run left twice". "run right" can be solved by "run right". So, "run left twice and run right" can.be solved by: "run left", "run left twice", "run right".

Q: "run opposite right"
A: "run opposite right" can be solved by "run opposite right".

Q: "look opposite right thrice after walk"
A: "look opposite right thrice" can be solved by: "look opposite right", "look opposite right thrice". "walk" can be solved by "walk". So, "look opposite right thrice after walk" can be solved by: "look opposite right", "look opposite right thrice", "walk".

Q: "jump around right"
A: "jump around right" can be solved by: "jump right", "jump around right". So, "jump around right" can be solved by: "jump right", "jump around right".

Q: "look around right thrice and walk"
A: "look around right thrice" can be solved by: "look right", "look around right", "look around right thrice". "walk" can be solved by "walk". So, "look around right thrice and walk" can be solved by: "look right", "look around right", "look around right thrice", "walk".

Q: "turn right after run right thrice"
A: "turn right" can be solved by: "turn right". "run right thrice" can be solved by: "run right", "run right thrice". So, "turn right after run right thrice" can be solved by: "turn right", "run right", "run right thrice".

### 8.1.3 CHAIN-OF-THOUGHT PROMPTING

We reuse the command mapping prompt context from least-to-most prompting shown above.

## 8.2 ERROR ANALYSIS: LEAST-TO-MOST PROMPTING

For least-to-most prompting, we analyzed 20 random failures for the models `code-davinci-001` and `text-davinci-002`, and we analyzed all 13 errors for the model `code-davinci-002`. The results are shown in Table 15. Errors may either occur during command decomposition or during command translation. The translation errors are further split into the following types. Incorrect interpretation of "twice" and "thrice" means that the model made an error when applying "twice" and "thrice" to an expression. "After" interpreted as "and" means that the model translated an expression containing "after" as if it instead contained "and". Copy error means that the model made a mistake when copying an intermediate result.

| Error type | code-002 | code-001 | text-002 |
|---|---|---|---|
| Decomposition error | 0 | 7 | 1 |
| Incorrect interpretation of "twice" and "thrice" | 6 | 10 | 16 |
| - Following "around" | 6 | 3 | 15 |
| - Following "opposite" | 0 | 3 | 1 |
| - Other | 0 | 4 | |
| "after" interpreted as "and" | 7 | 4 | 0 |
| Incorrect interpretation of "left" and "right" | 0 | 0 | 4 |
| Copy error | 0 | 4 | 0 |

Table 15: Least-to-most prompting error analysis of 20 random failures for the models `code-davinci-001` and `text-davinci-002` and all 13 errors for the model `code-davinci-002`. Note that for some examples, the model made more than one type of error.

We observe that the best model `text-davinci-002` made only 2 types of mistakes: it sometimes makes a mistake when applying "twice" and "thrice" to an expression containing "around", and it sometimes interprets "after" as "and". For `text-davinci-001`, which is the older version of the same model, the errors are spread across all types. In particular, it's worth noting that `text-davinci-001` makes a significant number of decomposition errors and copy errors that were completely eliminated by its successor.

The model `text-davinci-002` made most of its mistakes when interpreting "twice" and "thrice" following "around". In addition, it sometimes made a mistake when translating "left" and "right", which is something we did not observe with the other models. In some cases, it dropped the command entirely, whereas in other cases it invented a new action such as "LOOK_LEFT" (see examples below).

**Examples of decomposition errors.** In the example "run around left twice after jump around right thrice", the `code-davinci-001` model does not properly decompose the sub-expression "run around left twice". Instead of decomposing it to the sequence ["run left", "run around left", "run around left twice"], it skips "run around left" and decomposes it to ["run left", "run around left twice"]. Consequently, the model translates this sub-expression to ("TURN_LEFT" + "RUN") * 2 instead of ("TURN_LEFT" + "RUN") * 4 * 2.

In the example "look around right twice after jump around left twice", the `code-davinci-001` model does not properly decompose the sub-expression "jump around left twice". Instead of decomposing it to the sequence ["jump left", "jump around left", "jump around left twice"], it skips "jump around left" and decomposes it to ["jump left", "jump around left twice"]. Interestingly, the model is able to recover from this mistake and correctly translates the sub-expression to ("TURN_LEFT" + "JUMP") * 4 * 2, but still produces the wrong final action sequence because it interprets "after" like "and".

**Examples of incorrect interpretation of "twice" and "thrice".** In the example "walk opposite right twice after run around right thrice", the `code-davinci-002` model correctly translates the expression "run around right" to ("TURN_RIGHT" + "RUN") * 4. Then it makes a mistake when applying "thrice" to this expression and produces ("TURN_RIGHT" + "RUN") * 9 instead of ("TURN_RIGHT" + "RUN") * 4 * 3 or ("TURN_RIGHT" + "RUN") * 12.

In the example "jump opposite right twice and jump around right thrice", the `code-davinci-002` model correctly translates the expression "jump around right" to ("TURN_RIGHT" + "JUMP") * 4. Then it makes a mistake when applying "thrice" to this expression and produces ("TURN_RIGHT" + "JUMP") * 8 instead of ("TURN_RIGHT" + "JUMP") * 4 * 3 or ("TURN_RIGHT" + "JUMP") * 12.

In the example "walk around left thrice after run opposite left thrice", the `code-davinci-001` model correctly translates the expression "run opposite left" to "TURN_LEFT" * 2 + "RUN". Then it makes a mistake when applying "thrice" to this expression and produces "TURN_LEFT" * 2 + "RUN" * 3 instead of ("TURN_LEFT" * 2 + "RUN") * 3.

In the example "walk around left thrice after look right twice", the code-davinci-001 model correctly translates the expression "look right" to "TURN_RIGHT" + "LOOK". Then it makes a mistake when applying "twice" to this expression and produces "TURN_RIGHT" + "LOOK" * 2 rather than ("TURN_RIGHT" + "LOOK") * 2.

In the example "walk left and run around right thrice", the code-davinci-001 model interprets "thrice" as 'twice'. This means that it produces "TURN_LEFT" + "WALK" + ("TURN_RIGHT" + "RUN") * 4 * 2 instead of "TURN_LEFT" + "WALK" + ("TURN_RIGHT" + "RUN") * 4 * 3.

In the example "jump right twice and look around left thrice", the text-davinci-002 model correctly translates the sub-expression "look around left" to ("TURN_LEFT" + "LOOK") * 4. But when applying "thrice", it produces the incorrect translation ("TURN_LEFT" + "LOOK") * 3 instead of ("TURN_LEFT" + "LOOK") * 4 * 3.

**Example of interpreting "after" as "and".** In the example "run opposite left thrice after run around left twice", the code-davinci-002 model produces the correct translations for both sub-expressions that are connected by "after", but it combines them as if they were connected by "and". This means that the model produces ("TURN_LEFT" * 2 + "RUN") * 3 + ("TURN_LEFT" + "RUN") * 4 * 2 instead of ("TURN_LEFT" + "RUN") * 4 * 2 + ("TURN_LEFT" * 2 + "RUN") * 3.

In the example "walk around left thrice after walk twice", the code-davinci-002 model produces the correct translations for both sub-expressions that are connected by "after", but it combines them as if they were connected by "and". This means that the model produces ("TURN_LEFT" + "WALK") * 4 * 3 + "WALK" * 2 instead of "WALK" * 2 + ("TURN_LEFT" + "WALK") * 4 * 3.

In the example "look around right twice after jump around left twice", the code-davinci-001 model produces the correct translations for both sub-expressions that are connected by "after", but it combines them as if they were connected by "and". This means that the model produces ("TURN_RIGHT" + "LOOK") * 4 * 2 + ("TURN_LEFT" + "JUMP") * 4 * 2 instead of ("TURN_LEFT" + "JUMP") * 4 * 2 + ("TURN_RIGHT" + "LOOK") * 4 * 2.

**Examples of incorrect interpretation of "left" and "right".** In the example "look opposite right thrice after look around left thrice", the text-davinci-002 model translates the component "look left" to "LOOK" instead of "TURN_LEFT LOOK". As a consequence, the whole command is translated to "LOOK" * 4 * 3 + ("TURN_RIGHT" * 2 + "LOOK") * 3 instead of ("TURN_LEFT" + "LOOK") * 4 * 3 + ("TURN_RIGHT" * 2 + "LOOK") * 3.

In the example "turn around right thrice after look around left twice", the text-davinci-002 model makes up the new action "LOOK_LEFT" as the translation of the component "look left". As a consequence, it translates the whole command to ("LOOK_LEFT" * 4) * 2 + ("TURN_RIGHT" * 4) * 3 instead of ("TURN_LEFT" + "LOOK") * 4 * 2 + ("TURN_RIGHT" * 4) * 3.

**Example of copy error.** In the example "walk opposite right twice after look around left thrice", the code-davinci-001 model produces the correct translations for both sub-expressions that are connected with "after". In particular, it correctly translates "walk opposite right twice" to ("TURN_RIGHT" * 2 + "WALK") * 2. Then it makes two mistakes when combining them: it treats "after" like an "and", and it omits the parenthesis when copying the expression ("TURN_RIGHT" * 2 + "WALK") * 2. This means that the model produces the translation "TURN_RIGHT" * 2 + "WALK" * 2 + ("TURN_LEFT" + "LOOK") * 4 * 3 instead of ("TURN_LEFT" + "LOOK") * 4 * 3 + ("TURN_RIGHT" * 2 + "WALK") * 2.

## 8.3   Example outputs from code-davinci-002

In this section, we present 5 examples of successes and failures of the code-davinci-002 model for chain-of-thought prompting and least-to-most prompting.

### 8.3.1   Chain-of-thought prompting: Success

**"jump around left twice after walk opposite left thrice"**

**Golden:**
TURN_LEFT TURN_LEFT WALK TURN_LEFT TURN_LEFT WALK TURN_LEFT TURN_LEFT WALK

TURN_LEFT JUMP TURN_LEFT JUMP TURN_LEFT JUMP TURN_LEFT JUMP TURN_LEFT JUMP TURN_LEFT JUMP TURN_LEFT JUMP TURN_LEFT JUMP

**Solution:**
Q: "jump around left twice after walk opposite left thrice"
A: The output of "jump around left twice after walk opposite left thrice" concatenates: the output of "walk opposite left thrice", the output of "jump around left twice". "walk opposite left thrice" outputs ("TURN_LEFT" * 2 + "WALK") * 3. "jump around left twice" outputs ("TURN_LEFT" + "JUMP") * 4 * 2. So concatenating the output of "walk opposite left thrice" and the output of "jump around left twice" leads to ("TURN_LEFT" * 2 + "WALK") * 3 + ("TURN_LEFT" + "JUMP") * 4 * 2. So the output of "jump around left twice after walk opposite left thrice" is ("TURN_LEFT" * 2 + "WALK") * 3 + ("TURN_LEFT" + "JUMP") * 4 * 2.

---

**"walk around left twice and turn around right thrice"**

**Golden:**
TURN_LEFT WALK TURN_LEFT WALK TURN_LEFT WALK TURN_LEFT WALK TURN_LEFT WALK TURN_LEFT WALK TURN_LEFT WALK TURN_LEFT WALK TURN_RIGHT TURN_RIGHT TURN_RIGHT TURN_RIGHT TURN_RIGHT TURN_RIGHT TURN_RIGHT TURN_RIGHT TURN_RIGHT TURN_RIGHT TURN_RIGHT TURN_RIGHT

**Solution:**
Q: "walk around left twice and turn around right thrice"
A: The output of "walk around left twice and turn around right thrice" concatenates: the output of "walk around left", the output of "walk around left", the output of "turn around right", the output of "turn around right", the output of "turn around right". "walk around left" outputs ("TURN_LEFT" + "WALK") * 4. "turn around right" outputs "TURN_RIGHT" * 4. So concatenating the output of "walk around left", the output of "walk around left", the output of "turn around right", the output of "turn around right", the output of "turn around right" leads to ("TURN_LEFT" + "WALK") * 4 + ("TURN_LEFT" + "WALK") * 4 + "TURN_RIGHT" * 4 + "TURN_RIGHT" * 4 + "TURN_RIGHT" * 4. So the output of "walk around left twice and turn around right thrice" is ("TURN_LEFT" + "WALK") * 4 + ("TURN_LEFT" + "WALK") * 4 + "TURN_RIGHT" * 4 + "TURN_RIGHT" * 4 + "TURN_RIGHT" * 4.

---

**"jump around right thrice after run left thrice"**

**Golden:**
TURN_LEFT RUN TURN_LEFT RUN TURN_LEFT RUN TURN_RIGHT JUMP TURN_RIGHT JUMP TURN_RIGHT JUMP TURN_RIGHT JUMP TURN_RIGHT JUMP TURN_RIGHT JUMP TURN_RIGHT JUMP TURN_RIGHT JUMP TURN_RIGHT JUMP TURN_RIGHT JUMP TURN_RIGHT JUMP TURN_RIGHT JUMP

**Solution:**
Q: "jump around right thrice after run left thrice"
A: The output of "jump around right thrice after run left thrice" concatenates: the output of "run left thrice", the output of "jump around right thrice". "run left thrice" outputs ("TURN_LEFT" + "RUN") * 3. "jump around right thrice" outputs ("TURN_RIGHT" + "JUMP") * 12. So concatenating the output of "run left thrice" and the output of "jump around right thrice" leads to ("TURN_LEFT" + "RUN") * 3 + ("TURN_RIGHT" + "JUMP") * 12. So the output of "jump around right thrice after run left thrice" is ("TURN_LEFT" + "RUN") * 3 + ("TURN_RIGHT" + "JUMP") * 12.

---

**"run around left twice and walk around left thrice"**

**Golden:**
TURN_LEFT RUN TURN_LEFT RUN TURN_LEFT RUN TURN_LEFT RUN TURN_LEFT RUN TURN_LEFT RUN TURN_LEFT RUN TURN_LEFT RUN TURN_LEFT WALK TURN_LEFT WALK TURN_LEFT WALK TURN_LEFT WALK TURN_LEFT WALK TURN_LEFT WALK TURN_LEFT WALK TURN_LEFT WALK TURN_LEFT WALK TURN_LEFT WALK TURN_LEFT WALK TURN_LEFT WALK TURN_LEFT WALK

**Solution:**
Q: "run around left twice and walk around left thrice"
A: The output of "run around left twice and walk around left thrice" concatenates: the output of "run around left twice", the output of "walk around left thrice". "run around left twice" outputs ("TURN_LEFT" + "RUN") * 4 * 2. "walk around left thrice" outputs ("TURN_LEFT" + "WALK") * 4 * 3. So concatenating the output of "run around left twice" and the output of "walk around left thrice" leads to ("TURN_LEFT" + "RUN") * 4 * 2 + ("TURN_LEFT" + "WALK") * 4 * 3. So the output of "run around left twice and walk around left thrice" is ("TURN_LEFT" + "RUN") * 4 * 2 + ("TURN_LEFT" + "WALK") * 4 * 3.

---

**"jump around left thrice after turn around left thrice"**

**Golden:**
TURN_LEFT TURN_LEFT TURN_LEFT TURN_LEFT TURN_LEFT TURN_LEFT TURN_LEFT TURN_LEFT TURN_LEFT TURN_LEFT TURN_LEFT TURN_LEFT TURN_LEFT JUMP TURN_LEFT JUMP TURN_LEFT JUMP TURN_LEFT JUMP TURN_LEFT JUMP TURN_LEFT JUMP TURN_LEFT JUMP TURN_LEFT JUMP TURN_LEFT JUMP TURN_LEFT JUMP TURN_LEFT JUMP TURN_LEFT JUMP

**Solution:**
Q: "jump around left thrice after turn around left thrice"
A: The output of "jump around left thrice after turn around left thrice" concatenates: the output of "turn around left thrice", the output of "jump around left thrice". "turn around left thrice" outputs "TURN_LEFT" * 4 * 3. "jump around left thrice" outputs ("TURN_LEFT" + "JUMP") * 4 * 3. So concatenating the output of "turn around left thrice" and the output of "jump around left thrice" leads to "TURN_LEFT" * 4 * 3 + ("TURN_LEFT" + "JUMP") * 4 * 3. So the output of "jump around left thrice after turn around left thrice" is "TURN_LEFT" * 4 * 3 + ("TURN_LEFT" + "JUMP") * 4 * 3.

### 8.3.2 CHAIN-OF-THOUGHT PROMPTING: FAILURE

**"look around right and look around right thrice"**

**Golden:**
TURN_RIGHT LOOK TURN_RIGHT LOOK TURN_RIGHT LOOK TURN_RIGHT LOOK TURN_RIGHT LOOK TURN_RIGHT LOOK TURN_RIGHT LOOK TURN_RIGHT LOOK TURN_RIGHT LOOK TURN_RIGHT LOOK TURN_RIGHT LOOK TURN_RIGHT LOOK TURN_RIGHT LOOK TURN_RIGHT LOOK TURN_RIGHT LOOK TURN_RIGHT LOOK

**Solution:**
Q: "look around right and look around right thrice"
A: The output of "look around right and look around right thrice" concatenates: the output of "look around right", the output of "look around right thrice". "look around right" outputs "TURN_RIGHT" * 4 + "LOOK". "look around right thrice" outputs ("TURN_RIGHT" * 4 + "LOOK") * 3. So concatenating the output of "look around right" and the output of "look around right thrice" leads to "TURN_RIGHT" * 4 + "LOOK" + ("TURN_RIGHT" * 4 + "LOOK") * 3. So the output of "look around right and look around right thrice" is "TURN_RIGHT" * 4 + "LOOK" + ("TURN_RIGHT" * 4 + "LOOK") * 3.

---

**"run around left thrice and run left twice"**

**Golden:**
TURN_LEFT RUN TURN_LEFT RUN TURN_LEFT RUN TURN_LEFT RUN TURN_LEFT RUN TURN_LEFT RUN TURN_LEFT RUN TURN_LEFT RUN TURN_LEFT RUN TURN_LEFT RUN TURN_LEFT RUN TURN_LEFT RUN TURN_LEFT RUN TURN_LEFT RUN

**Solution:**
Q: "run around left thrice and run left twice"
A: The output of "run around left thrice and run left twice" concatenates: the output of "run around

left thrice", the output of "run left twice". "run around left thrice" outputs ("TURN_LEFT" + "RUN") * 3. "run left twice" outputs "TURN_LEFT" + "RUN" * 2. So concatenating the output of "run around left thrice" and the output of "run left twice" leads to ("TURN_LEFT" + "RUN") * 3 + "TURN_LEFT" + "RUN" * 2. So the output of "run around left thrice and run left twice" is ("TURN_LEFT" + "RUN") * 3 + "TURN_LEFT" + "RUN" * 2.

---

**"walk and walk around left thrice"**

**Golden:**
WALK TURN_LEFT WALK TURN_LEFT WALK TURN_LEFT WALK TURN_LEFT WALK TURN_LEFT WALK TURN_LEFT WALK TURN_LEFT WALK TURN_LEFT WALK TURN_LEFT WALK TURN_LEFT WALK TURN_LEFT WALK TURN_LEFT WALK

**Solution:**
Q: "walk and walk around left thrice"
A: The output of "walk and walk around left thrice" concatenates: the output of "walk", the output of "walk around left thrice". "walk" outputs "WALK". "walk around left thrice" outputs ("TURN_LEFT" + "WALK") * 4. So concatenating the output of "walk" and the output of "walk around left thrice" leads to "WALK" + ("TURN_LEFT" + "WALK") * 4. So the output of "walk and walk around left thrice" is "WALK" + ("TURN_LEFT" + "WALK") * 4.

---

**"jump around right thrice after turn opposite left twice"**

**Golden:**
TURN_LEFT TURN_LEFT TURN_LEFT TURN_LEFT TURN_RIGHT JUMP TURN_RIGHT JUMP TURN_RIGHT JUMP TURN_RIGHT JUMP TURN_RIGHT JUMP TURN_RIGHT JUMP TURN_RIGHT JUMP TURN_RIGHT JUMP TURN_RIGHT JUMP TURN_RIGHT JUMP TURN_RIGHT JUMP TURN_RIGHT JUMP

**Solution:**
Q: "jump around right thrice after turn opposite left twice"
A: The output of "jump around right thrice after turn opposite left twice" concatenates: the output of "turn opposite left twice", the output of "jump around right thrice". "turn opposite left twice" outputs "TURN_LEFT" * 2. "jump around right thrice" outputs ("TURN_RIGHT" + "JUMP") * 4. So concatenating the output of "turn opposite left twice" and the output of "jump around right thrice" leads to "TURN_LEFT" * 2 + ("TURN_RIGHT" + "JUMP") * 4. So the output of "jump around right thrice after turn opposite left twice" is "TURN_LEFT" * 2 + ("TURN_RIGHT" + "JUMP") * 4.

---

**"run opposite right after walk around right thrice"**

**Golden:**
TURN_RIGHT WALK TURN_RIGHT WALK TURN_RIGHT WALK TURN_RIGHT WALK TURN_RIGHT WALK TURN_RIGHT WALK TURN_RIGHT WALK TURN_RIGHT WALK TURN_RIGHT WALK TURN_RIGHT WALK TURN_RIGHT WALK TURN_RIGHT WALK TURN_RIGHT TURN_RIGHT RUN

**Solution:**
Q: "run opposite right after walk around right thrice"
A: The output of "run opposite right after walk around right thrice" concatenates: the output of "walk around right thrice", the output of "run opposite right". "walk around right thrice" outputs ("TURN_RIGHT" + "WALK") * 3. "run opposite right" outputs "TURN_RIGHT" * 2 + "RUN". So concatenating the output of "walk around right thrice" and the output of "run opposite right" leads to ("TURN_RIGHT" + "WALK") * 3 + "TURN_RIGHT" * 2 + "RUN". So the output of "run opposite right after walk around right thrice" is ("TURN_RIGHT" + "WALK") * 3 + "TURN_RIGHT" * 2 + "RUN".

### 8.3.3 LEAST-TO-MOST PROMPTING: SUCCESS

**"jump around left thrice and walk opposite right"**

**Golden:**
TURN_LEFT JUMP TURN_LEFT JUMP TURN_LEFT JUMP TURN_LEFT JUMP TURN_LEFT JUMP TURN_LEFT JUMP TURN_LEFT JUMP TURN_LEFT JUMP TURN_LEFT JUMP TURN_LEFT JUMP TURN_LEFT JUMP TURN_LEFT JUMP TURN_RIGHT TURN_RIGHT WALK

**Decomposition:**
Q: "jump around left thrice and walk opposite right"
A: "jump around left thrice" can be solved by: "jump left", "jump around left", "jump around left thrice". "walk opposite right" can be solved by: "walk opposite right". So, "jump around left thrice and walk opposite right" can be solved by: "jump left", "jump around left", "jump around left thrice", "walk opposite right".

**Solution:**
Q: "jump left"
A: The output of "jump left" concatenates: the output of "turn left", the output of "jump". "turn left" outputs "TURN_LEFT". "jump" outputs "JUMP". So concatenating the output of "turn left" and the output of "jump" leads to "TURN_LEFT" + "JUMP". So the output of "jump left" is "TURN_LEFT" + "JUMP".

Q: "jump around left"
A: The output of "jump around left" concatenates: the output of "jump left", the output of "jump left", the output of "jump left", the output of "jump left". "jump left" outputs "TURN_LEFT" + "JUMP". So repeating the output of "jump around left" four times leads to ("TURN_LEFT" + "JUMP") * 4. So the output of "jump around left" is ("TURN_LEFT" + "JUMP") * 4.

Q: "jump around left thrice"
A: The output of "jump around left thrice" concatenates: the output of "jump around left", the output of "jump around left", the output of "jump around left". "jump around left" outputs ("TURN_LEFT" + "JUMP") * 4. So repeating the output of "jump around left" three times leads to ("TURN_LEFT" + "JUMP") * 4 * 3. So the output of "jump around left thrice" is ("TURN_LEFT" + "JUMP") * 4 * 3.

Q: "walk opposite right"
A: The output of "walk opposite right" concatenates: the output of "turn opposite right", the output of "walk". "turn opposite right" outputs "TURN_RIGHT" * 2. "walk" outputs "WALK". So concatenating the output of "turn opposite right" and the output of "walk" leads to "TURN_RIGHT" * 2 + "WALK". So the output of "walk opposite right" is "TURN_RIGHT" * 2 + "WALK".

Q: "jump around left thrice and walk opposite right"
A: The output of "jump around left thrice and walk opposite right" concatenates: the output of "jump around left thrice", the output of "walk opposite right". "jump around left thrice" outputs ("TURN_LEFT" + "JUMP") * 4 * 3. "walk opposite right" outputs "TURN_RIGHT" * 2 + "WALK". So concatenating the output of "jump around left thrice" and the output of "walk opposite right" leads to ("TURN_LEFT" + "JUMP") * 4 * 3 + "TURN_RIGHT" * 2 + "WALK". So the output of "jump around left thrice and walk opposite right" is ("TURN_LEFT" + "JUMP") * 4 * 3 + "TURN_RIGHT" * 2 + "WALK".

---

**"turn around left thrice after run around right thrice"**

**Golden:**
TURN_RIGHT RUN TURN_RIGHT RUN TURN_RIGHT RUN TURN_RIGHT RUN TURN_RIGHT RUN TURN_RIGHT RUN TURN_RIGHT RUN TURN_RIGHT RUN TURN_RIGHT RUN TURN_RIGHT RUN TURN_RIGHT RUN TURN_RIGHT RUN TURN_LEFT TURN_LEFT TURN_LEFT TURN_LEFT TURN_LEFT TURN_LEFT TURN_LEFT TURN_LEFT TURN_LEFT TURN_LEFT TURN_LEFT TURN_LEFT

**Decomposition:**
Q: "turn around left thrice after run around right thrice"
A: "turn around left thrice" can be solved by: "turn left", "turn around left", "turn around left thrice". "run around right thrice" can be solved by: "run right", "run around right", "run around right thrice". So, "turn around left thrice after run around right thrice" can be solved by: "turn left", "turn around left", "turn around left thrice", "run right", "run around right", "run around right thrice".

**Solution:**
Q: "turn around left"
A: The output of "turn around left" concatenates: the output of "turn left", the output of "turn left", the output of "turn left", the output of "turn left". "turn left" outputs "TURN_LEFT". So repeating the output of "turn left" four times leads to "TURN_LEFT" * 4. So the output of "turn around left" is "TURN_LEFT" * 4.

Q: "turn around left thrice"
A: The output of "turn around left thrice" concatenates: the output of "turn around left", the output of "turn around left", the output of "turn around left". "turn around left" outputs "TURN_LEFT" * 4. So repeating the output of "turn around left" three times leads to ("TURN_LEFT" * 4) * 3. So the output of "turn around left thrice" is ("TURN_LEFT" * 4) * 3.

Q: "run right"
A: The output of "run right" concatenates: the output of "turn right", the output of "run". "turn right" outputs "TURN_RIGHT". "run" outputs "RUN". So concatenating the output of "turn right" and the output of "run" leads to "TURN_RIGHT" + "RUN". So the output of "run right" is "TURN_RIGHT" + "RUN".

Q: "run around right"
A: The output of "run around right" concatenates: the output of "run right", the output of "run right", the output of "run right", the output of "run right". "run right" outputs "TURN_RIGHT" + "RUN". So repeating the output of "run right" four times leads to ("TURN_RIGHT" + "RUN") * 4. So the output of "run around right" is ("TURN_RIGHT" + "RUN") * 4.

Q: "run around right thrice"
A: The output of "run around right thrice" concatenates: the output of "run around right", the output of "run around right", the output of "run around right". "run around right" outputs ("TURN_RIGHT" + "RUN") * 4. So repeating the output of "run around right" three times leads to (("TURN_RIGHT" + "RUN") * 4) * 3. So the output of "run around right thrice" is (("TURN_RIGHT" + "RUN") * 4) * 3.

Q: "turn around left thrice after run around right thrice"
A: The output of "turn around left thrice after run around right thrice" concatenates: the output of "run around right thrice", the output of "turn around left thrice". "run around right thrice" outputs (("TURN_RIGHT" + "RUN") * 4) * 3. "turn around left thrice" outputs ("TURN_LEFT" * 4) * 3. So concatenating the output of "run around right thrice" and the output of "turn around left thrice" leads to (("TURN_RIGHT" + "RUN") * 4) * 3 + ("TURN_LEFT" * 4) * 3. So the output of "turn around left thrice after run around right thrice" is (("TURN_RIGHT" + "RUN") * 4) * 3 + ("TURN_LEFT" * 4) * 3.

---

**"turn left after look around left thrice"**

**Golden:**
TURN_LEFT LOOK TURN_LEFT LOOK TURN_LEFT LOOK TURN_LEFT LOOK TURN_LEFT LOOK TURN_LEFT LOOK TURN_LEFT LOOK TURN_LEFT LOOK TURN_LEFT LOOK TURN_LEFT LOOK TURN_LEFT LOOK TURN_LEFT LOOK TURN_LEFT

**Decomposition:**
Q: "turn left after look around left thrice"
A: "turn left" can be solved by: "turn left". "look around left thrice" can be solved by: "look left", "look around left", "look around left thrice". So, "turn left after look around left thrice" can be solved by: "turn left", "look left", "look around left", "look around left thrice".

**Solution:**
Q: "look left"
A: The output of "look left" concatenates: the output of "turn left", the output of "look". "turn left" outputs "TURN_LEFT". "look" outputs "LOOK". So concatenating the output of "turn left" and the output of "look" leads to "TURN_LEFT" + "LOOK". So the output of "look left" is "TURN_LEFT" + "LOOK".

Q: "look around left"
A: The output of "look around left" concatenates: the output of "look left", the output of "look left", the

output of "look left", the output of "look left". "look left" outputs "TURN_LEFT" + "LOOK". So repeating the output of "look left" four times leads to ("TURN_LEFT" + "LOOK") * 4. So the output of "look around left" is ("TURN_LEFT" + "LOOK") * 4.

Q: "look around left thrice"
A: The output of "look around left thrice" concatenates: the output of "look around left", the output of "look around left", the output of "look around left". "look around left" outputs ("TURN_LEFT" + "LOOK") * 4. So repeating the output of "look around left" three times leads to (("TURN_LEFT" + "LOOK") * 4) * 3. So the output of "look around left thrice" is (("TURN_LEFT" + "LOOK") * 4) * 3.

Q: "turn left after look around left thrice"
A: The output of "turn left after look around left thrice" concatenates: the output of "look around left thrice", the output of "turn left". "look around left thrice" outputs (("TURN_LEFT" + "LOOK") * 4) * 3. "turn left" outputs "TURN_LEFT". So concatenating the output of "look around left thrice" and the output of "turn left" leads to (("TURN_LEFT" + "LOOK") * 4) * 3 + "TURN_LEFT". So the output of "turn left after look around left thrice" is (("TURN_LEFT" + "LOOK") * 4) * 3 + "TURN_LEFT".

---

**"turn around left thrice and jump around left thrice"**

**Golden:**
TURN_LEFT TURN_LEFT TURN_LEFT TURN_LEFT TURN_LEFT TURN_LEFT TURN_LEFT TURN_LEFT TURN_LEFT TURN_LEFT TURN_LEFT TURN_LEFT TURN_LEFT JUMP TURN_LEFT JUMP TURN_LEFT JUMP TURN_LEFT JUMP TURN_LEFT JUMP TURN_LEFT JUMP TURN_LEFT JUMP TURN_LEFT JUMP TURN_LEFT JUMP TURN_LEFT JUMP TURN_LEFT JUMP TURN_LEFT JUMP TURN_LEFT JUMP

**Decomposition:**
Q: "turn around left thrice and jump around left thrice"
A: "turn around left thrice" can be solved by: "turn left", "turn around left", "turn around left thrice". "jump around left thrice" can be solved by: "jump left", "jump around left", "jump around left thrice". So, "turn around left thrice and jump around left thrice" can be solved by: "turn left", "turn around left", "turn around left thrice", "jump left", "jump around left", "jump around left thrice".

**Solution:**
Q: "turn around left"
A: The output of "turn around left" concatenates: the output of "turn left", the output of "turn left", the output of "turn left", the output of "turn left". "turn left" outputs "TURN_LEFT". So repeating the output of "turn left" four times leads to "TURN_LEFT" * 4. So the output of "turn around left" is "TURN_LEFT" * 4.

Q: "turn around left thrice"
A: The output of "turn around left thrice" concatenates: the output of "turn around left", the output of "turn around left", the output of "turn around left". "turn around left" outputs "TURN_LEFT" * 4. So repeating the output of "turn around left" three times leads to ("TURN_LEFT" * 4) * 3. So the output of "turn around left thrice" is ("TURN_LEFT" * 4) * 3.

Q: "jump left"
A: The output of "jump left" concatenates: the output of "turn left", the output of "jump". "turn left" outputs "TURN_LEFT". "jump" outputs "JUMP". So concatenating the output of "turn left" and the output of "jump" leads to "TURN_LEFT" + "JUMP". So the output of "jump left" is "TURN_LEFT" + "JUMP".

Q: "jump around left"
A: The output of "jump around left" concatenates: the output of "jump left", the output of "jump left", the output of "jump left", the output of "jump left". "jump left" outputs "TURN_LEFT" + "JUMP". So repeating the output of "jump left" four times leads to ("TURN_LEFT" + "JUMP") * 4. So the output of "jump around left" is ("TURN_LEFT" + "JUMP") * 4.

Q: "jump around left thrice"
A: The output of "jump around left thrice" concatenates: the output of "jump around left", the output of "jump around left", the output of "jump around left". "jump around left" outputs ("TURN_LEFT" + "JUMP") * 4. So repeating the output of "jump around left" three times leads to (("TURN_LEFT" + "JUMP") * 4) * 3. So the output of "jump around left thrice" is (("TURN_LEFT" + "JUMP") * 4) * 3.

Q: "turn around left thrice and jump around left thrice"
A: The output of "turn around left thrice and jump around left thrice" concatenates: the output of "turn around left thrice", the output of "jump around left thrice". "turn around left thrice" outputs ("TURN_LEFT" * 4) * 3. "jump around left thrice" outputs (("TURN_LEFT" + "JUMP") * 4) * 3. So concatenating the output of "turn around left thrice" and the output of "jump around left thrice" leads to ("TURN_LEFT" * 4) * 3 + (("TURN_LEFT" + "JUMP") * 4) * 3. So the output of "turn around left thrice and jump around left thrice" is ("TURN_LEFT" * 4) * 3 + (("TURN_LEFT" + "JUMP") * 4) * 3.

---

**"walk around right twice after walk around right twice"**

**Golden:**
TURN_RIGHT WALK TURN_RIGHT WALK TURN_RIGHT WALK TURN_RIGHT WALK TURN_RIGHT WALK TURN_RIGHT WALK TURN_RIGHT WALK TURN_RIGHT WALK TURN_RIGHT WALK TURN_RIGHT WALK TURN_RIGHT WALK TURN_RIGHT WALK TURN_RIGHT WALK TURN_RIGHT WALK TURN_RIGHT WALK TURN_RIGHT WALK

**Decomposition:**
Q: "walk around right twice after walk around right twice"
A: "walk around right twice" can be solved by: "walk right", "walk around right", "walk around right twice". "walk around right twice" can be solved by: "walk right", "walk around right", "walk around right twice". So, "walk around right twice after walk around right twice" can be solved by: "walk right", "walk around right", "walk around right twice", "walk right", "walk around right", "walk around right twice".

**Solution:**
Q: "walk right"
A: The output of "walk right" concatenates: the output of "turn right", the output of "walk". "turn right" outputs "TURN_RIGHT". "walk" outputs "WALK". So concatenating the output of "turn right" and the output of "walk" leads to "TURN_RIGHT" + "WALK". So the output of "walk right" is "TURN_RIGHT" + "WALK".

Q: "walk around right"
A: The output of "walk around right" concatenates: the output of "walk right", the output of "walk right", the output of "walk right", the output of "walk right". "walk right" outputs "TURN_RIGHT" + "WALK". So repeating the output of "walk around right" four times leads to ("TURN_RIGHT" + "WALK") * 4. So the output of "walk around right" is ("TURN_RIGHT" + "WALK") * 4.

Q: "walk around right twice"
A: The output of "walk around right twice" concatenates: the output of "walk around right", the output of "walk around right". "walk around right" outputs ("TURN_RIGHT" + "WALK") * 4. So repeating the output of "walk around right" twice leads to ("TURN_RIGHT" + "WALK") * 4 * 2. So the output of "walk around right twice" is ("TURN_RIGHT" + "WALK") * 4 * 2.

Q: "walk around right twice after walk around right twice"
A: The output of "walk around right twice after walk around right twice" concatenates: the output of "walk around right twice", the output of "walk around right twice". "walk around right twice" outputs ("TURN_RIGHT" + "WALK") * 4 * 2. So repeating the output of "walk around right twice" twice leads to ("TURN_RIGHT" + "WALK") * 4 * 2 * 2. So the output of "walk around right twice after walk around right twice" is ("TURN_RIGHT" + "WALK") * 4 * 2 * 2.

### 8.3.4 LEAST-TO-MOST PROMPTING: FAILURE

**"jump opposite right twice and jump around right thrice"**

**Golden:**
TURN_RIGHT TURN_RIGHT JUMP TURN_RIGHT TURN_RIGHT JUMP TURN_RIGHT JUMP

TURN_RIGHT JUMP TURN_RIGHT JUMP TURN_RIGHT JUMP TURN_RIGHT JUMP TURN_RIGHT JUMP TURN_RIGHT JUMP TURN_RIGHT JUMP TURN_RIGHT JUMP TURN_RIGHT JUMP TURN_RIGHT JUMP TURN_RIGHT JUMP

**Decomposition:**
Q: "jump opposite right twice and jump around right thrice"
A: "jump opposite right twice" can be solved by: "jump opposite right", "jump opposite right twice". "jump around right thrice" can be solved by: "jump right", "jump around right", "jump around right thrice". So, "jump opposite right twice and jump around right thrice" can be solved by: "jump opposite right", "jump opposite right twice", "jump right", "jump around right", "jump around right thrice".

**Solution:**
Q: "jump opposite right"
A: The output of "jump opposite right" concatenates: the output of "turn opposite right", the output of "jump". "turn opposite right" outputs "TURN_RIGHT" * 2. "jump" outputs "JUMP". So concatenating the output of "turn opposite right" and the output of "jump" leads to "TURN_RIGHT" * 2 + "JUMP". So the output of "jump opposite right" is "TURN_RIGHT" * 2 + "JUMP".

Q: "jump opposite right twice"
A: The output of "jump opposite right twice" concatenates: the output of "jump opposite right", the output of "jump opposite right". "jump opposite right" outputs "TURN_RIGHT" * 2 + "JUMP". So repeating the output of "jump opposite right" twice leads to ("TURN_RIGHT" * 2 + "JUMP") * 2. So the output of "jump opposite right twice" is ("TURN_RIGHT" * 2 + "JUMP") * 2.

Q: "jump right"
A: The output of "jump right" concatenates: the output of "turn right", the output of "jump". "turn right" outputs "TURN_RIGHT". "jump" outputs "JUMP". So concatenating the output of "turn right" and the output of "jump" leads to "TURN_RIGHT" + "JUMP". So the output of "jump right" is "TURN_RIGHT" + "JUMP".

Q: "jump around right"
A: The output of "jump around right" concatenates: the output of "jump right", the output of "jump right", the output of "jump right", the output of "jump right". "jump right" outputs "TURN_RIGHT" + "JUMP". So repeating the output of "jump around right" four times leads to ("TURN_RIGHT" + "JUMP") * 4. So the output of "jump around right" is ("TURN_RIGHT" + "JUMP") * 4.

Q: "jump around right thrice"
A: The output of "jump around right thrice" concatenates: the output of "jump right", the output of "jump right", the output of "jump right", the output of "jump right", the output of "jump right", the output of "jump right", the output of "jump right", the output of "jump right". "jump right" outputs "TURN_RIGHT" + "JUMP". So repeating the output of "jump around right" eight times leads to ("TURN_RIGHT" + "JUMP") * 8. So the output of "jump around right thrice" is ("TURN_RIGHT" + "JUMP") * 8.

Q: "jump opposite right twice and jump around right thrice"
A: The output of "jump opposite right twice and jump around right thrice" concatenates: the output of "jump opposite right twice", the output of "jump around right thrice". "jump opposite right twice" outputs ("TURN_RIGHT" * 2 + "JUMP") * 2. "jump around right thrice" outputs ("TURN_RIGHT" + "JUMP") * 8. So concatenating the output of "jump opposite right twice" and the output of "jump around right thrice" leads to ("TURN_RIGHT" * 2 + "JUMP") * 2 + ("TURN_RIGHT" + "JUMP") * 8. So the output of "jump opposite right twice and jump around right thrice" is ("TURN_RIGHT" * 2 + "JUMP") * 2 + ("TURN_RIGHT" + "JUMP") * 8.

---

**"walk around left thrice after walk around right twice"**

**Golden:**
TURN_RIGHT WALK TURN_RIGHT WALK TURN_RIGHT WALK TURN_RIGHT WALK TURN_RIGHT WALK TURN_RIGHT WALK TURN_RIGHT WALK TURN_RIGHT WALK TURN_LEFT WALK TURN_LEFT WALK TURN_LEFT WALK TURN_LEFT WALK TURN_LEFT WALK TURN_LEFT WALK TURN_LEFT WALK TURN_LEFT WALK TURN_LEFT WALK TURN_LEFT WALK TURN_LEFT WALK TURN_LEFT WALK

**Decomposition:**
Q: "walk around left thrice after walk around right twice"
A: "walk around left thrice" can be solved by: "walk left", "walk around left", "walk around left thrice". "walk around right twice" can be solved by: "walk right", "walk around right", "walk around right twice". So, "walk around left thrice after walk around right twice" can be solved by: "walk left", "walk around left", "walk around left thrice", "walk right", "walk around right", "walk around right twice".

**Solution:**
Q: "walk left"
A: The output of "walk left" concatenates: the output of "turn left", the output of "walk". "turn left" outputs "TURN_LEFT". "walk" outputs "WALK". So concatenating the output of "turn left" and the output of "walk" leads to "TURN_LEFT" + "WALK". So the output of "walk left" is "TURN_LEFT" + "WALK".

Q: "walk around left"
A: The output of "walk around left" concatenates: the output of "walk left", the output of "walk left", the output of "walk left", the output of "walk left". "walk left" outputs "TURN_LEFT" + "WALK". So repeating the output of "walk around left" four times leads to ("TURN_LEFT" + "WALK") * 4. So the output of "walk around left" is ("TURN_LEFT" + "WALK") * 4.

Q: "walk around left thrice"
A: The output of "walk around left thrice" concatenates: the output of "walk around left", the output of "walk around left", the output of "walk around left". "walk around left" outputs ("TURN_LEFT" + "WALK") * 4. So repeating the output of "walk around left" three times leads to ("TURN_LEFT" + "WALK") * 4 * 3. So the output of "walk around left thrice" is ("TURN_LEFT" + "WALK") * 4 * 3.

Q: "walk right"
A: The output of "walk right" concatenates: the output of "turn right", the output of "walk". "turn right" outputs "TURN_RIGHT". "walk" outputs "WALK". So concatenating the output of "turn right" and the output of "walk" leads to "TURN_RIGHT" + "WALK". So the output of "walk right" is "TURN_RIGHT" + "WALK".

Q: "walk around right"
A: The output of "walk around right" concatenates: the output of "walk right", the output of "walk right", the output of "walk right", the output of "walk right". "walk right" outputs "TURN_RIGHT" + "WALK". So repeating the output of "walk around right" four times leads to ("TURN_RIGHT" + "WALK") * 4. So the output of "walk around right" is ("TURN_RIGHT" + "WALK") * 4.

Q: "walk around right twice"
A: The output of "walk around right twice" concatenates: the output of "walk around right", the output of "walk around right". "walk around right" outputs ("TURN_RIGHT" + "WALK") * 4. So repeating the output of "walk around right" twice leads to ("TURN_RIGHT" + "WALK") * 4 * 2. So the output of "walk around right twice" is ("TURN_RIGHT" + "WALK") * 4 * 2.

Q: "walk around left thrice after walk around right twice"
A: The output of "walk around left thrice after walk around right twice" concatenates: the output of "walk around left thrice", the output of "walk around right twice". "walk around left thrice" outputs ("TURN_LEFT" + "WALK") * 4 * 3. "walk around right twice" outputs ("TURN_RIGHT" + "WALK") * 4 * 2. So concatenating the output of "walk around left thrice" and the output of "walk around right twice" leads to ("TURN_LEFT" + "WALK") * 4 * 3 + ("TURN_RIGHT" + "WALK") * 4 * 2. So the output of "walk around left thrice after walk around right twice" is ("TURN_LEFT" + "WALK") * 4 * 3 + ("TURN_RIGHT" + "WALK") * 4 * 2.

---

**"run around left thrice after run around right twice"**

**Golden:**
TURN_RIGHT RUN TURN_RIGHT RUN TURN_RIGHT RUN TURN_RIGHT RUN TURN_RIGHT RUN TURN_RIGHT RUN TURN_RIGHT RUN TURN_RIGHT RUN TURN_LEFT RUN TURN_LEFT RUN TURN_LEFT RUN TURN_LEFT RUN TURN_LEFT RUN TURN_LEFT RUN TURN_LEFT RUN TURN_LEFT RUN TURN_LEFT RUN TURN_LEFT RUN TURN_LEFT RUN TURN_LEFT RUN TURN_LEFT RUN

**Decomposition:**
Q: "run around left thrice after run around right twice"

A: "run around left thrice" can be solved by: "run left", "run around left", "run around left thrice". "run around right twice" can be solved by: "run right", "run around right", "run around right twice". So, "run around left thrice after run around right twice" can be solved by: "run left", "run around left", "run around left thrice", "run right", "run around right", "run around right twice".

**Solution:**
Q: "run left"
A: The output of "run left" concatenates: the output of "turn left", the output of "run". "turn left" outputs "TURN_LEFT". "run" outputs "RUN". So concatenating the output of "turn left" and the output of "run" leads to "TURN_LEFT" + "RUN". So the output of "run left" is "TURN_LEFT" + "RUN".

Q: "run around left"
A: The output of "run around left" concatenates: the output of "run left", the output of "run left", the output of "run left", the output of "run left". "run left" outputs "TURN_LEFT" + "RUN". So repeating the output of "run around left" four times leads to ("TURN_LEFT" + "RUN") * 4. So the output of "run around left" is ("TURN_LEFT" + "RUN") * 4.

Q: "run around left thrice"
A: The output of "run around left thrice" concatenates: the output of "run around left", the output of "run around left", the output of "run around left". "run around left" outputs ("TURN_LEFT" + "RUN") * 4. So repeating the output of "run around left" three times leads to ("TURN_LEFT" + "RUN") * 4 * 3. So the output of "run around left thrice" is ("TURN_LEFT" + "RUN") * 4 * 3.

Q: "run right"
A: The output of "run right" concatenates: the output of "turn right", the output of "run". "turn right" outputs "TURN_RIGHT". "run" outputs "RUN". So concatenating the output of "turn right" and the output of "run" leads to "TURN_RIGHT" + "RUN". So the output of "run right" is "TURN_RIGHT" + "RUN".

Q: "run around right"
A: The output of "run around right" concatenates: the output of "run right", the output of "run right", the output of "run right", the output of "run right". "run right" outputs "TURN_RIGHT" + "RUN". So repeating the output of "run around right" four times leads to ("TURN_RIGHT" + "RUN") * 4. So the output of "run around right" is ("TURN_RIGHT" + "RUN") * 4.

Q: "run around right twice"
A: The output of "run around right twice" concatenates: the output of "run around right", the output of "run around right". "run around right" outputs ("TURN_RIGHT" + "RUN") * 4. So repeating the output of "run around right" twice leads to ("TURN_RIGHT" + "RUN") * 4 * 2. So the output of "run around right twice" is ("TURN_RIGHT" + "RUN") * 4 * 2.

Q: "run around left thrice after run around right twice"
A: The output of "run around left thrice after run around right twice" concatenates: the output of "run around left thrice", the output of "run around right twice". "run around left thrice" outputs ("TURN_LEFT" + "RUN") * 4 * 3. "run around right twice" outputs ("TURN_RIGHT" + "RUN") * 4 * 2. So concatenating the output of "run around left thrice" and the output of "run around right twice" leads to ("TURN_LEFT" + "RUN") * 4 * 3 + ("TURN_RIGHT" + "RUN") * 4 * 2. So the output of "run around left thrice after run around right twice" is ("TURN_LEFT" + "RUN") * 4 * 3 + ("TURN_RIGHT" + "RUN") * 4 * 2.

---

**"jump opposite right twice after jump around right thrice"**

**Golden:**
TURN_RIGHT JUMP TURN_RIGHT JUMP TURN_RIGHT JUMP TURN_RIGHT JUMP TURN_RIGHT JUMP TURN_RIGHT JUMP TURN_RIGHT JUMP TURN_RIGHT JUMP TURN_RIGHT JUMP TURN_RIGHT JUMP TURN_RIGHT JUMP TURN_RIGHT JUMP TURN_RIGHT TURN_RIGHT JUMP TURN_RIGHT TURN_RIGHT JUMP

**Decomposition:**
Q: "jump opposite right twice after jump around right thrice"
A: "jump opposite right twice" can be solved by: "jump opposite right", "jump opposite right twice". "jump around right thrice" can be solved by: "jump right", "jump around right", "jump around right

thrice". So, "jump opposite right twice after jump around right thrice" can be solved by: "jump opposite right", "jump opposite right twice", "jump right", "jump around right", "jump around right thrice".

**Solution:**
Q: "jump opposite right"
A: The output of "jump opposite right" concatenates: the output of "turn opposite right", the output of "jump". "turn opposite right" outputs "TURN_RIGHT" * 2. "jump" outputs "JUMP". So concatenating the output of "turn opposite right" and the output of "jump" leads to "TURN_RIGHT" * 2 + "JUMP". So the output of "jump opposite right" is "TURN_RIGHT" * 2 + "JUMP".

Q: "jump opposite right twice"
A: The output of "jump opposite right twice" concatenates: the output of "jump opposite right", the output of "jump opposite right". "jump opposite right" outputs "TURN_RIGHT" * 2 + "JUMP". So repeating the output of "jump opposite right" twice leads to ("TURN_RIGHT" * 2 + "JUMP") * 2. So the output of "jump opposite right twice" is ("TURN_RIGHT" * 2 + "JUMP") * 2.

Q: "jump right"
A: The output of "jump right" concatenates: the output of "turn right", the output of "jump". "turn right" outputs "TURN_RIGHT". "jump" outputs "JUMP". So concatenating the output of "turn right" and the output of "jump" leads to "TURN_RIGHT" + "JUMP". So the output of "jump right" is "TURN_RIGHT" + "JUMP".

Q: "jump around right"
A: The output of "jump around right" concatenates: the output of "jump right", the output of "jump right", the output of "jump right", the output of "jump right". "jump right" outputs "TURN_RIGHT" + "JUMP". So repeating the output of "jump around right" four times leads to ("TURN_RIGHT" + "JUMP") * 4. So the output of "jump around right" is ("TURN_RIGHT" + "JUMP") * 4.

Q: "jump around right thrice"
A: The output of "jump around right thrice" concatenates: the output of "jump right", the output of "jump right", the output of "jump right", the output of "jump right", the output of "jump right", the output of "jump right", the output of "jump right", the output of "jump right". "jump right" outputs "TURN_RIGHT" + "JUMP". So repeating the output of "jump around right" eight times leads to ("TURN_RIGHT" + "JUMP") * 8. So the output of "jump around right thrice" is ("TURN_RIGHT" + "JUMP") * 8.

Q: "jump opposite right twice after jump around right thrice"
A: The output of "jump opposite right twice after jump around right thrice" concatenates: the output of "jump around right thrice", the output of "jump opposite right twice". "jump around right thrice" outputs ("TURN_RIGHT" + "JUMP") * 8. "jump opposite right twice" outputs ("TURN_RIGHT" * 2 + "JUMP") * 2. So concatenating the output of "jump around right thrice" and the output of "jump opposite right twice" leads to ("TURN_RIGHT" + "JUMP") * 8 + ("TURN_RIGHT" * 2 + "JUMP") * 2. So the output of "jump opposite right twice after jump around right thrice" is ("TURN_RIGHT" + "JUMP") * 8 + ("TURN_RIGHT" * 2 + "JUMP") * 2.

---

**"walk around left thrice after walk right twice"**

**Golden:**
TURN_RIGHT WALK TURN_RIGHT WALK TURN_LEFT WALK TURN_LEFT WALK TURN_LEFT WALK TURN_LEFT WALK TURN_LEFT WALK TURN_LEFT WALK TURN_LEFT WALK TURN_LEFT WALK TURN_LEFT WALK TURN_LEFT WALK TURN_LEFT WALK TURN_LEFT WALK

**Decomposition:**
Q: "walk around left thrice after walk right twice"
A: "walk around left thrice" can be solved by: "walk left", "walk around left", "walk around left thrice". "walk right twice" can be solved by: "walk right", "walk right twice". So, "walk around left thrice after walk right twice" can be solved by: "walk left", "walk around left", "walk around left thrice", "walk right", "walk right twice".

**Solution:**

Q: "walk left"
A: The output of "walk left" concatenates: the output of "turn left", the output of "walk". "turn left" outputs "TURN_LEFT". "walk" outputs "WALK". So concatenating the output of "turn left" and the output of "walk" leads to "TURN_LEFT" + "WALK". So the output of "walk left" is "TURN_LEFT" + "WALK".

Q: "walk around left"
A: The output of "walk around left" concatenates: the output of "walk left", the output of "walk left", the output of "walk left", the output of "walk left". "walk left" outputs "TURN_LEFT" + "WALK". So repeating the output of "walk around left" four times leads to ("TURN_LEFT" + "WALK") * 4. So the output of "walk around left" is ("TURN_LEFT" + "WALK") * 4.

Q: "walk around left thrice"
A: The output of "walk around left thrice" concatenates: the output of "walk around left", the output of "walk around left", the output of "walk around left". "walk around left" outputs ("TURN_LEFT" + "WALK") * 4. So repeating the output of "walk around left" three times leads to ("TURN_LEFT" + "WALK") * 4 * 3. So the output of "walk around left thrice" is ("TURN_LEFT" + "WALK") * 4 * 3.

Q: "walk right"
A: The output of "walk right" concatenates: the output of "turn right", the output of "walk". "turn right" outputs "TURN_RIGHT". "walk" outputs "WALK". So concatenating the output of "turn right" and the output of "walk" leads to "TURN_RIGHT" + "WALK". So the output of "walk right" is "TURN_RIGHT" + "WALK".

Q: "walk right twice"
A: The output of "walk right twice" concatenates: the output of "walk right", the output of "walk right". "walk right" outputs "TURN_RIGHT" + "WALK". So repeating the output of "walk right" twice leads to ("TURN_RIGHT" + "WALK") * 2. So the output of "walk right twice" is ("TURN_RIGHT" + "WALK") * 2.

Q: "walk around left thrice after walk right twice"
A: The output of "walk around left thrice after walk right twice" concatenates: the output of "walk around left thrice", the output of "walk right twice". "walk around left thrice" outputs ("TURN_LEFT" + "WALK") * 4 * 3. "walk right twice" outputs ("TURN_RIGHT" + "WALK") * 2. So concatenating the output of "walk around left thrice" and the output of "walk right twice" leads to ("TURN_LEFT" + "WALK") * 4 * 3 + ("TURN_RIGHT" + "WALK") * 2. So the output of "walk around left thrice after walk right twice" is ("TURN_LEFT" + "WALK") * 4 * 3 + ("TURN_RIGHT" + "WALK") * 2.

## 8.4 EXPANDING PYTHON EXPRESSIONS USING PROMPTING

In Section 3.2, we mention that expanding the Python expressions that we use as an intermediate representation can be done either with a simple postprocessing script or by prompting a language model. In the following, we present a prompt that achieves 99.7% accuracy on a random sample of 1000 Python expressions that are outputted by our solution (using the code-davinci-002 model). This demonstrates that the L2M method can solve SCAN with a combined accuracy more than 99% (99.7% accuracy for generating the intermediate Python expressions and 99.7% for expanding these expressions), even if we do not use the Python executor and instead perform the expansion of the intermediate representation via prompting.

Q: "JUMP" * 3
Rewrite: "JUMP" * 3
A: 1 JUMP 2 JUMP 3 JUMP

Q: "RUN" * 4 * 2
Rewrite: "RUN" * 8
A: 1 RUN 2 RUN 3 RUN 4 RUN 5 RUN 6 RUN 7 RUN 8 RUN

Q: "TURN_RIGHT" + "WALK"
Rewrite: "TURN_RIGHT" + "WALK"
A: TURN_RIGHT WALK

Q: ("TURN_LEFT" + "LOOK") * 2 + "TURN_LEFT" + "LOOK"
Rewrite: ("TURN_LEFT" + "LOOK") * 2 + "TURN_LEFT" + "LOOK"
A: 1 (TURN_LEFT LOOK) 2 (TURN_LEFT LOOK) TURN_LEFT LOOK

Q: ("TURN_RIGHT" * 2 + "JUMP") * 4
Rewrite: ("TURN_RIGHT" * 2 + "JUMP") * 4
A: 1 (1 TURN_RIGHT 2 TURN_RIGHT JUMP) 2 (1 TURN_RIGHT 2 TURN_RIGHT JUMP) 3 (1 TURN_RIGHT 2 TURN_RIGHT JUMP) 4 (1 TURN_RIGHT 2 TURN_RIGHT JUMP)

Q: "TURN_LEFT" * 2 + ("TURN_RIGHT" + "WALK") * 4 * 2
Rewrite: "TURN_LEFT" * 2 + ("TURN_RIGHT" + "WALK") * 8
A: 1 TURN_LEFT 2 TURN_LEFT 1 (TURN_RIGHT WALK) 2 (TURN_RIGHT WALK) 3 (TURN_RIGHT WALK) 4 (TURN_RIGHT WALK) 5 (TURN_RIGHT WALK) 6 (TURN_RIGHT WALK) 7 (TURN_RIGHT WALK) 8 (TURN_RIGHT WALK)

**Discussion.** The prompt consists of 6 examples, each of which illustrates part of the knowledge needed for this task. Note that we add numbers and parentheses when we unfold multiplication to make it easier for the model to keep track of the repetitions.

1. Multiplication
2. Sequential multiplication
3. Addition
4. Avoid commutativity / associativity in addition
5. Nested multiplication
6. Addition of two multiplications

## 9 DROP

### 9.1 RESULTS WITH TEXT-DAVINCI-002 AND LM-540B

We reported the results using `code-davinci-002`. Here, we report results using the `text-davinci-002` model and a language model with 540 billion parameters (`LM-540B`).

| text-davinci-002 | | |
|---|---|---|
| Prompting method | Non-football (500 cases) | Football (500 cases) |
| Zero-Shot | 27.00 | 31.60 |
| Standard prompting | 49.40 | 54.40 |
| Chain-of-Thought | 60.80 | 57.40 |
| Least-to-Most | **74.20** | **63.40** |

Table 16: Accuracies (%) of zero-shot and prompting methods with the GPT-3 `text-davinci-002` model on the numerical reasoning subset of DROP. We evaluate on 500 randomly sampled non-football/football examples. Compared to Table **??**, we observe that `text-davinci-002` is consistently worse than `code-davinci-002`.

### 9.2 NON-FOOTBALL SUBSET

#### 9.2.1 ZERO-SHOT PROMPTING

For zero-shot, the prompt format is as follows:

Q: {question}
A: The answer is

Notice that we add "The answer is" at the beginning of the answer section.

| lm-540b | | |
|---|---|---|
| Prompting method | Non-football (3988 cases) | Football (1862 cases) |
| Zero-Shot | 48.42 | 44.95 |
| Standard prompting | 56.54 | 60.47 |
| Chain-of-Thought | 63.84 | 67.35 |
| Least-to-Most | **79.24** | **69.98** |

Table 17: Accuracies (%) of zero-shot and prompting methods with a pretrained language model with 540 billion parameters (lm-540).

### 9.2.2 STANDARD PROMPTING WITH 3 EXAMPLES

Q: Since the 1970s, U.S. governments have negotiated managed-trade agreements, such as the North American Free Trade Agreement in the 1990s, the Dominican Republic-Central America Free Trade Agreement in 2006, and a number of bilateral agreements. In Europe, six countries formed the European Coal and Steel Community in 1951 which became the European Economic Community in 1958. Two core objectives of the EEC were the development of a common market, subsequently renamed the single market, and establishing a customs union between its member states. How many years did the European Coal and Steel Community exist?
A: The answer is 7.

Q: In the county, the population was spread out with 23.50% under the age of 18, 8.70% from 18 to 24, 29.70% from 25 to 44, 24.70% from 45 to 64, and 13.30% who were 65 years of age or older. How many more percent are under the age of 18 compared to the 18 to 24 group?
A: The answer is 14.8.

Q: Playing in their second straight Thanksgiving game, the Eagles struggled especially on defense, where they were unable to stop the much-hyped Lions offense. The worst of it all was how unproven rookie Eric Rowe was tasked with covering wide receiver Calvin Johnson, leading to Johnson catching 3 touchdowns. Stafford's five passing touchdowns, including three of them to Johnson was too much for the Eagles to overcome and for the second consecutive time this season, the Eagles gave up 45 points in a game. With the loss, the Eagles drop to 4-7 on the season and 6-1 when playing on Thanksgiving. How many TD passes did Stafford throw other than to Johnson?
A: The answer is 2.

### 9.2.3 CHAIN-OF-THOUGHT PROMPTING WITH 3 EXAMPLES

Q: Since the 1970s, U.S. governments have negotiated managed-trade agreements, such as the North American Free Trade Agreement in the 1990s, the Dominican Republic-Central America Free Trade Agreement in 2006, and a number of bilateral agreements. In Europe, six countries formed the European Coal and Steel Community in 1951 which became the European Economic Community in 1958. Two core objectives of the EEC were the development of a common market, subsequently renamed the single market, and establishing a customs union between its member states. How many years did the European Coal and Steel Community exist?
A: According to the passage, the European Coal and Steel Community was established in 1951 and became the EEC in 1958. 1958 - 1951 = 7. So the answer is 7.

Q: In the county, the population was spread out with 23.50% under the age of 18, 8.70% from 18 to 24, 29.70% from 25 to 44, 24.70% from 45 to 64, and 13.30% who were 65 years of age or older. How many more percent are under the age of 18 compared to the 18 to 24 group?
A: According to the passage, 23.5% are under the age of 18, and 8.7% are from ages 18 to 24. 23.5% - 8.7% = 14.8%. So the answer is 14.8.

Q: Playing in their second straight Thanksgiving game, the Eagles struggled especially on defense, where they were unable to stop the much-hyped Lions offense. The worst of it all was how unproven rookie Eric Rowe was tasked with covering wide receiver Calvin Johnson, leading to Johnson catching 3 touchdowns. Stafford's five passing touchdowns, including three of them to Johnson was too much for the Eagles to overcome and for the second consecutive time this season, the Eagles gave up 45 points in a game. With the loss, the Eagles drop to 4-7 on the season and 6-1 when playing on Thanksgiving. How many TD passes did Stafford throw other than to Jhonson?

A: According to the passage, Stafford threw 5 TD passes, 3 of which were to Johnson. 5 - 3 = 2. So the answer is 2.

### 9.2.4 LEAST-TO-MOST PROMPTING I: PROBLEM DECOMPOSITION (5 EXAMPLES)

Q: The median age in the city was 22.1 years. 10.1% of residents were under the age of 18; 56.2% were between the ages of 18 and 24; 16.1% were from 25 to 44; 10.5% were from 45 to 64; and 7% were 65 years of age or older. Which age group is larger: under the age of 18 or 18 and 24?
A: To answer the question "Which age group is larger: under the age of 18 or 18 and 24?", we need to know: "How many percent were under the age of 18?", "How many percent were between the ages of 18 and 24?".

Q: Old age pensions were raised by 300 francs per month to 1,700 francs for a single person and to 3,700 francs for a couple, while health insurance benefits were made more widely available to unemployed persons and part-time employees. How many francs were the old age pensions for a single person before they were raised?
A: To answer the question "How many francs were the old age pensions for a single person before they were raised?", we need to know: "How many francs were the old age pensions for a single person?", "How many francs were old age pensions raised for a single person?".

Q: In April 2011, the ECB raised interest rates for the first time since 2008 from 1% to 1.25%, with a further increase to 1.50% in July 2011. However, in 2012-2013 the ECB lowered interest rates to encourage economic growth, reaching the historically low 0.25% in November 2013. Soon after the rates were cut to 0.15%, then on 4 September 2014 the central bank reduced the rates from 0.15% to 0.05%, the lowest rates on record. How many percentage points did interest rates drop between April 2011 and September 2014?
A: To answer the question "How many percentage points did interest rates drop between April 2011 and September 2014?", we need to know: "What was the interest rate in April 2011?", "What was the interest rate in September 2014?".

Q: Non-nationals make up more than half of the population of Bahrain. According to government statistics dated between 2005-2009 roughly 290,000 Indians, 125,000 Bangladeshis, 45,000 Pakistanis, 45,000 Filipinos, and 8,000 Indonesians. How many Pakistanis and Indonesians are in Bahrain?
A: To answer the question "How many Pakistanis and Indonesians are in Bahrain?", we need to know: "How many Pakistanis are in Bahrain?", "How many Indonesians are in Bahrain?".

Q: The Census Bureaus 2006-2010 American Community Survey showed that (in 2010 inflation adjustment dollars) median household income was $52,056 and the median family income was $58,942. How many years did the Census Bureaus American Community Survey last?
A: To answer the question "How many years did the Census Bureaus American Community Survey last?", we need to know: "When did the Census Bureaus American Community Survey start?", "When did the Census Bureaus American Community Survey end?".

### 9.2.5 LEAST-TO-MOST PROMPTING II: PROBLEM SOLVING (3 EXAMPLES)

The city of Bangkok has a population of 8,280,925. The census showed that it is home to 81,570 Japanese and 55,893 Chinese nationals, as well as 117,071 expatriates from other Asian countries, 48,341 from Europe, 23,418 from the Americas.
Q: How many Chinese nationals are there?
A: "The census showed that it is home to 81,570 Japanese and 55,893 Chinese nationals". So the answer is 55,893 Chinese nationals.

Q: How many European nationals are there?
A: "48,341 from Europe". So the answer is 48,341.

Q: How many more Chinese nationals are there than European nationals?
A: We know that there are 55,893 Chinese nationals. We also know that there are 48,341 European nationals. So there are 55,893 - 48,341 = 7552 more Chinese nationals than European nationals. So the answer is 7552.

The gender distribution of the population was 50.2% male and 49.8% female. Of the adult population, 29 people or 14.6% of the population are between 20 and 29 years old. 28 people or 14.1% are

30 to 39, 36 people or 18.2% are 40 to 49, and 31 people or 15.7% are 50 to 59.
Q: What is the percentage of people between 40 to 49?.
A: "36 people or 18.2% are 40 to 49". So the answer is 18.2

Q: How many percent of people were not 40 to 49?
A: We know that 18.2% are 40 to 49. So 100% - 18.2% = 81.8% are not 40 to 49. So the answer is 81.8%.

During the 1970s the oil shocks (1973 oil crisis and 1979 energy crisis) hit Iceland hard. Inflation rose to 43% in 1974 and 59% in 1980, falling to 15% in 1987 but rising to 30% in 1988. Iceland experienced moderately strong GDP growth (3% on average) from 1995 to 2004. Growth slowed between 2000 and 2002, but the economy expanded by 4.3% in 2003 and grew by 6.2% in 2004. Growth in 2005 exceeded 6%.
Q: When did the oil crisis occur?
A: "1973 oil crisis and 1979 energy crisis". So the answer is 1973.

Q: When did the energy crisis occur?
A: "1979 energy crisis". So the answer is 1979.

Q: How many years passed between the oil and energy crises?
A: We know that the oil crisis occurred in 1973. We also know that the energy crisis occurred in 1979. So 1979 - 1973 = 6 years passed between the oil and energy crises. So the answer is 6.

### 9.3    FOOTBALL SUBSET

#### 9.3.1    ZERO-SHOT PROMPTING

For zero-shot, the prompt format is as follows:

Q: {question}
A: The answer is

Notice that we add "The answer is" at the beginning of the answer section.

#### 9.3.2    STANDARD PROMPTING WITH 3 EXAMPLES

Q: The Seahawks played the San Francisco 49ers. In the first quarter, the Hawks RB Julius Jones got a 27-yard TD run, along with DT Craig Terrill returning a fumble 9 yards for a touchdown. In the third quarter, the 49ers almost rallied as RB H. J. Torres made a 12-yard TD pass to Lucas Nelly, along with Mare kicking a 32-yard field goal. In the final quarter, Julius Jones got another 11-yard TD. How many yards do the shortest touchdown run and the longest touchdown pass combine for?
A: The answer is 21.

Q: The Steelers went home for a duel with the Baltimore Ravens. Pittsburgh would deliver the opening punch in the first quarter with a 1-yard touchdown from running back Rashard Mendenhall. The Ravens would make it even as running back Willis McGahee got a 9-yard TD. The Ravens kicker Billy Cundiff got a 45-yard field goal in the second quarter, concluding the first half with a 10-7 lead. The Steelers brought the game into overtime with a 38-yard field goal by Andrew Foster. The Ravens Billy Cundiff pulled off a winning 33-yard field goal in overtime. How many points did the Ravens have at halftime?
A: The answer is 10.

Q: The Vikings flew to Bank of America Stadium to face the Carolina Panthers. After a scoreless first quarter, Carolina got on the board with quarterback Matt Moore finding fullback Brad Hoover on a 1-yard TD pass. After yet another scoreless quarter, Carolina sealed the game as Matt Moore completed a 42-yard touchdown pass to wide receiver Steve Smith. How many scoreless quarters were there?
A: The answer is 2.

#### 9.3.3    CHAIN-OF-THOUGHT PROMPTING WITH 3 EXAMPLES

Q: The Seahawks played the San Francisco 49ers. In the first quarter, the Hawks RB Julius Jones got a 27-yard TD run, along with DT Craig Terrill returning a fumble 9 yards for a touchdown. In

the third quarter, the 49ers almost rallied as RB H. J. Torres made a 12-yard TD pass to Lucas Nelly, along with Mare kicking a 32-yard field goal. In the final quarter, Julius Jones got another 11-yard TD. How many yards do the shortest touchdown run and the longest touchdown pass combine for?
A: All the touchdown runs are: a 27-yard touchdown run, a 9-yard touchdown run, a 11-yard touchdown run. The smallest number among 27, 9, 11 is 9. So the shortest touchdown run was 9 yards. All the touchdown passes are: a 12-yard touchdown pass. So the longest touchdown pass was 12 yards. So the shortest touchdown run and the longest touchdown pass combine for 9 + 12 = 21 yards. So the answer is 21 yards.

Q: The Steelers went home for a duel with the Baltimore Ravens. Pittsburgh would deliver the opening punch in the first quarter with a 1-yard touchdown from running back Rashard Mendenhall. The Ravens would make it even as running back Willis McGahee got a 9-yard TD. The Ravens kicker Billy Cundiff got a 45-yard field goal in the second quarter, concluding the first half with a 10-7 lead. The Steelers brought the game into overtime with a 38-yard field goal by Andrew Foster. The Ravens Billy Cundiff pulled off a winning 33-yard field goal in overtime. How many points did the Ravens have at halftime?
A: The Ravens kicker Billy Cundiff got a 45-yard field goal in the second quarter, concluding the first half with a 10-7 lead. So the Ravens had 10 points at halftime. So the answer is 10 points.

Q: The Vikings flew to Bank of America Stadium to face the Carolina Panthers. After a scoreless first quarter, Carolina got on the board with quarterback Matt Moore finding fullback Brad Hoover on a 1-yard TD pass. After yet another scoreless quarter, Carolina sealed the game as Matt Moore completed a 42-yard touchdown pass to wide receiver Steve Smith. How many scoreless quarters were there?
A: The first and third quarters were the scoreless quarters. So there are 2 scoreless quarters. So the answer is 2.

### 9.3.4 LEAST-TO-MOST PROMPTING I: PROBLEM DECOMPOSITION (6 EXAMPLES)

Q: The Seahawks played the San Francisco 49ers. In the first quarter, the 'Hawks RB Julius Jones got a 27-yard TD run, along with DT Craig Terrill returning a fumble 9 yards for a touchdown. In the third quarter, the 49ers almost rallied as RB T. J. Duckett made a 12-yard TD pass to Lucas Nelly, along with Mare kicking a 32-yard field goal. How many yards do the shortest touchdown run and the longest touchdown pass combine for?
A: To answer the question "How many yards do the shortest touchdown run and the longest touchdown pass combine for?", we need to know: "How many yards was the shortest touchdown run?", "How many yards was the longest touchdown pass?".

Q: The Steelers went home for an AFC North duel with the Baltimore Ravens. Pittsburgh would deliver the opening punch in the first quarter with a 1-yard touchdown run from running back Rashard Mendenhall. The Ravens would make it even in the second quarter as running back Willis McGahee got a 9-yard touchdown run. The Ravens kicker Billy Cundiff got a 45-yard field goal in the second quarter and a 33-yard field goal in the third quarter. Game ended with a scoreless fourth quarter. How many points did the Ravens have at halftime?
A: To answer the question "How many points did the Ravens have at halftime?", we need to know: "What were all the scores the Ravens had at halftime?".

Q: In 1995, the Kings overcame a 3-4 start to win eight of their final nine games and finished with a record, the second-best in the AFC. Quarterback Neil ODonnell, who completed 246 out of 416 passes for 2,970 yards and 17 touchdowns, with only seven interceptions led their offense. The Kings finished their 1995 season having lost how many games difference to the number of games they had won?
A: To answer the question "The Kings finished their 1995 season having lost how many games difference to the number of games they had won?", we need to know: "How many games the Kings had lost in their 1995 season?", "How many games the Kings had won in their 1995 season?".

Q: The Broncos traveled to Sun Life Stadium to face the Miami Dolphins. The Dolphins grabbed the lead in the second quarter, with field goals of 35 and 45 yards by kicker Dan Carpenter. In the final quarter, the Dolphins quarterback Matt Moore threw a 16-yard touchdown pass to tight end Anthony Fasano (with a failed two-point conversion attempt), followed by a 35-yard field goal by Carpenter, giving Miami a 15-0 lead. Finally, the Broncos answered with Alex Jake kicking a 48-yard field

goal, ending the game with 15-3. How many yards was Dan Carpenter's longest field goal?
A: To answer the question "How many yards was Dan Carpenter's longest field goal?", we need to know: "What were all Dan Carpenter's field goals?".

Q: The Cardinals flew to Gillette Stadium for a duel with the New England Patriots. In the fourth quarter, New England ended the boring game with Gostkowski's 30-yard field goal. Original starting quarterback Kurt Warner (6/18 for 30 yards) was pulled in the third quarter via coach's decision. How many incomplete passes did Kurt Warner have?
A: To answer the question "How many incomplete passes did Kurt Warner have?", we need to know: "How many passes did Kurt Warner attempt?", "How many passes did Kurt Warner complete?".

Q: The Vikings flew to Bank of America Stadium to face the Carolina Panthers. After a scoreless first quarter, Carolina got on the board with quarterback Matt Moore finding fullback Brad Hoover on a 1-yard touchdown pass. After yet another scoreless quarter, Carolina sealed the game as Matt Moore completed a 42-yard TD pass to wide receiver Steve Smith. How many scoreless quarters were there?
A: To answer the question "How many scoreless quarters were there?", we need to know: "What were all the scoreless quarters?".

### 9.3.5 LEAST-TO-MOST PROMPTING II: PROBLEM SOLVING (3 EXAMPLES)

The Seahawks played the San Francisco 49ers. In the first quarter, the 'Hawks RB Julius Jones got a 27-yard TD run, along with DT Craig Terrill returning a fumble 9 yards for a touchdown. In the third quarter, the 49ers almost rallied as RB H. J. Torres made a 12-yard TD pass to Lucas Nelly, along with Mare kicking a 32-yard field goal. In the final quarter, Julius Jones got another 11-yard TD.
Q: How many yards was the shortest touchdown run?
A: All the touchdown runs are: a 27-yard touchdown run, a 9-yard touchdown run, a 11-yard touchdown run. The smallest number between 27, 9, 11 is 9. The shortest touchdown run was 9 yards. The answer is 9.

Q: How many yards was the longest touchdown pass?
A: All the touchdown passes are: a 12-yard touchdown pass. The longest touchdown pass was 12 yards. The answer is 12.

Q: How many yards do the shortest touchdown run and the longest touchdown pass combine for?
A: We know that the shortest touchdown run was 9-yard. We also know that the longest touchdown pass was 12-yard. 9 + 12 = 21. The answer is 21.

The Steelers went home for a duel with the Baltimore Ravens. Pittsburgh would deliver the opening punch in the first quarter with a 1-yard touchdown from running back Rashard Mendenhall. The Ravens would make it even as running back Willis McGahee got a 9-yard TD. The Ravens kicker Billy Cundiff got a 45-yard field goal in the second quarter, concluding the first half. The Steelers brought the game into overtime with a 38-yard field goal by Andrew Foster. The Ravens Billy Cundiff pulled off a winning 33-yard field goal in overtime.
Q: What were all the scores the Ravens had at halftime?
A: The answer is: a 9-yard touchdown run, a 45-yard field goal.

Q: How many points did the Ravens have at halftime?
A: We know that the Ravens had a touchdown and a field goal at halftime. We also know that a touchdown is worth 7 points and a field goal is worth 3 points. 7 + 3 = 10. The answer is 10.

The Vikings flew to Bank of America Stadium to face the Carolina Panthers. After a scoreless first quarter, Carolina got on the board with quarterback Matt Moore finding fullback Brad Hoover on a 1-yard TD pass. After yet another scoreless quarter, Carolina sealed the game as Matt Moore completed a 42-yard touchdown pass to wide receiver Steve Smith.
Q: What were all the scoreless quarters?
A: The answer is: the first quarter and the third quarter.

Q: How many scoreless quarters were there?
A: We know that the first and third quarters were all the scoreless quarters. The answer is 2.

### 9.4 EXAMPLES WHERE LEAST-TO-MOST SUCCEEDED BUT CHAIN-OF-THOUGHT FAILED

#### 9.4.1 CASE 1

Q: The first two Buick automobiles were made in 1899 and 1900 at the "Buick Auto-Vim and Power Company" by chief-engineer Walter Lorenzo Marr, but company owner David Dunbar Buick was reluctant to begin making automobiles, being satisfied with stationary and marine engine production, so Marr left Buick in 1901 to found Marr (automobile) under his own name. His replacement was Eugene Richard, who applied for a patent in 1902 for Marrs valve-in-head (Overhead valve engine) engine, which patent, number 771,095, was awarded to Richard in the name of Buick in 1904. In 1903, the third Buick automobile was made, this time by Richard, but in 1904 Buick, whose company was now called "Buick Motor Company," moved to Flint, Michigan, and Richard stayed behind. Marr was rehired in Flint as chief engineer, to begin making automobiles in production. That year, 37 Buick automobiles were made, production increasing to 750 in 1905, 1,400 in 1906, 4,641 in 1907, and 8,800 in 1908, taking the number one spot away from close competitors Oldsmobile, Ford Motor Company, and Maxwell automobile. How many Buick automobiles were built between 1904 and 1905?

**Chain-of-thought by `code-davinci-002`**   A: According to the passage, 37 Buick automobiles were made in 1904, and 750 were made in 1905. 750 - 37 = 713. So the answer is 713.

**Least-to-most by `code-davinci-002`**   Q: How many Buick automobiles were built in 1904?
A: "In 1903, the third Buick automobile was made, this time by Richard, but in 1904 Buick, whose company was now called "Buick Motor Company," moved to Flint, Michigan, and Richard stayed behind. Marr was rehired in Flint as chief engineer, to begin making automobiles in production. That year, 37 Buick automobiles were made". So the answer is 37.

Q: How many Buick automobiles were built in 1905?
A: "production increasing to 750 in 1905". So the answer is 750.

Q: How many Buick automobiles were built between 1904 and 1905?
A: We know that 37 Buick automobiles were built in 1904. We also know that 750 Buick automobiles were built in 1905. So 37 + 750 = 787 Buick automobiles were built between 1904 and 1905. So the answer is 787.

**Analysis**   Chain-of-thought failed because it computed "750 - 37 = 713", which is wrong. The correct answer is "37 + 750 = 787".

#### 9.4.2 CASE 2

Q: The Guatemalan authorities were unaware of ongoing contacts between the colonial authorities in Yucatán and the Itza. In the second half of 1695 President Barrios began planning another expedition against the Itza from Verapaz, following harsh criticism of Juan Díaz de Velasco's retreat from near Lake Petén Itzá in April of that year. Barrios died in November, and oidor José de Escals became acting head of the Guatemalan colonial administration. He appointed oidor Bartolomé de Amésqueta to lead the next expedition against the Itza. Amésqueta marched his men through torrential rain from Cahabón to Mopán, arriving on 25 February 1696; the Guatemalans were still unaware of the clashes between García's forces and the Itza. Amésqueta's expedition was suffering from shortages of native labour and supplies. Twenty-five of his 150 men were sick, and his advance stalled as the force rested up at Mopán. Captain Díaz de Velasco volunteered to lead a party of 25 men ahead to the lake; he was accompanied by Dominican friars Cristóbal de Prada and Jacinto de Vargas. He was also accompanied by AjK'ixaw, the Itza nobleman who had been taken prisoner on Díaz's previous expedition, who served as a trusted guide, scout and translator. They set out from Mopán on 7 March and left some sick soldiers with some supplies after five days, they then caught up with an advance party of roadbuilders and their military escort at Río Chakal. The two forces combined, giving Díaz a force of 49 soldiers and 34 porters and archers from Verapaz. Once they arrived at IxB'ol, near the shore of Lake Petén Itzá, AjK'ixaw was sent ahead as an emissary to Nojpetén. How many months after Juan Díaz de Velasco's retreat did President Barrios die?

**Chain-of-thought by `code-davinci-002`**   A: According to the passage, President Barrios died in November, and the expedition against the Itza was led by Bartolomé de Amésqueta in February. So the answer is 3.

**Least-to-most by `code-davinci-002`**   Q: When did Juan Díaz de Velasco's retreat?
A: "April of that year". So the answer is April.

Q: When did President Barrios die?
A: "November". So the answer is November.

Q: How many months after Juan Díaz de Velasco's retreat did President Barrios die?
A: We know that Juan Díaz de Velasco's retreat was in April. We also know that President Barrios died in November. So November - April = 7 months after Juan Díaz de Velasco's retreat did President Barrios die. So the answer is 7.

**Analysis**   Chain-of-thought generated the wrong final answer because one fact it used, "the expedition against the Itza was led by Bartolomé de Amésqueta in February", is unrelated.

### 9.4.3   CASE 3

Q: Peele was christened on 25 July 1556 at St James Garlickhythe in the City of London. His father, James Peele (d. Dec 30, 1585), who appears to have belonged to a Devonshire family, was clerk of Christs Hospital, a school which was then situated in central London, and wrote two treatises on bookkeeping, The Maner and Fourme How to Kepe a Perfecte Reconyng (1553) and The Pathe Waye to Perfectnes (1569). The latter depicts James Peele in a woodcut on the title page. No contemporary likeness of George is known, although he was said to be short-legged, dark complected, red haired, and squinting, although not necessarily from reliable sources. Georges mother, Anne, died July 1, 1580, and his father married Christian Widers (d. 1597 in St. Christopher le Stocks) on November 3, 1580. She became a nurse on the Hospital payroll, where she remained five years after James Peeles death, when she married Ralph Boswell. His siblings included Anne (d. Jan 10, 1568/9), Isabel, Judith (d. Apr. 16, 1582), and James (b. Jan 3, 1563/4). Anne married John Alford on May 14, 1565, and had one son, Robert (October 9, 1567- c. March 12, 1654/5). Judith married John Jackman on June 19, 1575, and had three children, Susan (b. June 3, 1576), William (April 30, 1577-July 1, 1577), and Sarah (d. May 24, 1578). On Feb. 5, 1568/9, Isabel married Mathew Shakespeare, with whom she had eight children. Many scholars believe that this was a cousin of William Shakespeare, but this has not been verified. James Peele also wrote the Ironmongers Pageants of 1566 and 1569, which may have led to Georges writing of two Lord Mayors pageants. How many years after her marriage to James Peele did Christian Widers die?

**Chain-of-thought by `code-davinci-002`**   A: According to the passage, Christian Widers died in 1597. James Peele died in 1585. 1597 - 1585 = 12. So the answer is 12.

**Least-to-most by `code-davinci-002`**   Q: When did Christian Widers marry James Peele?
A: "She became a nurse on the Hospital payroll, where she remained five years after James Peeles death, when she married Ralph Boswell". So the answer is 1580.

Q: When did Christian Widers die?
A: "She became a nurse on the Hospital payroll, where she remained five years after James Peeles death, when she married Ralph Boswell". So the answer is 1597.

Q: How many years after her marriage to James Peele did Christian Widers die?
A: We know that Christian Widers married James Peele in 1580. We also know that Christian Widers died in 1597. So 1597 - 1580 = 17 years after her marriage to James Peele did Christian Widers die. So the answer is 17.

**Analysis**   Chain-of-thought generated the wrong final answer because one fact it used, "James Peele died in 1585", is unrelated.

### 9.4.4    CASE 4

Q: Among the other denominations there are approximately 93,000 Mormons in 253 congregations, 25,000 Jewish adherents in 21 synagogues, 12,000 Muslims in 39 masjids, 7,000 Buddhists in 34 temples, 7,000 Hindus in 17 temples, 2,500 Unitarian Universalism in 9 congregations, 2,000 Bahai in 17 temples, 5 Sikh temples, a Zoroastrian temple, a Jain temple and an uncounted number of neopagans. How many more Mormons were there compared to Muslims?

**Chain-of-thought by `code-davinci-002`**   A: According to the passage, there are 93,000 Mormons and 25,000 Muslims. 93,000 - 25,000 = 68,000. So the answer is 68,000.

**Least-to-most by `code-davinci-002`**   Q: How many Mormons were there?
A: "93,000 Mormons". So the answer is 93,000.

Q: How many Muslims were there?
A: "12,000 Muslims". So the answer is 12,000.

Q: How many more Mormons were there compared to Muslims?
A: We know that there were 93,000 Mormons. We also know that there were 12,000 Muslims. So there were 93,000 - 12,000 = 81,000 more Mormons than Muslims. So the answer is 81,000.

**Analysis**   Chain-of-thought failed because it got one intermediate fact (the number of Muslims) wrong.

### 9.4.5    CASE 5

Q: Killigrew lived in Hanworth in Middlesex and Falmouth in Cornwall. On 4 November 1566 Killigrew married in the church of St Peter Le Poer, London, Catherine, fourth daughter of Sir Anthony Cooke. He thus became Cecil's brother-in-law. His wife died in 1583, and on 7 November 1590 he was married in the same church to Jaél de Peigne, a French Hugenot. She was naturalised in June 1601. After Henry's death she remarried on 19 April 1617 George Downham, Bishop of Derry, and died c.1632. By his first wife, Killigrew had four daughters: How many years after here marriage to Killigrew did Catherine die?

**Chain-of-thought by `code-davinci-002`**   A: According to the passage, Catherine died in 1583, and Killigrew married Jaél de Peigne in 1590. 1590 - 1583 = 7. So the answer is 7.

**Least-to-most by `code-davinci-002`**   Q: When did Catherine marry Killigrew?
A: "On 4 November 1566 Killigrew married in the church of St Peter Le Poer, London, Catherine". So the answer is 4 November 1566.

Q: When did Catherine die?
A: "His wife died in 1583". So the answer is 1583.

Q: By his first wife, Killigrew had four daughters: How many years after here marriage to Killigrew did Catherine die?
A: We know that Catherine married Killigrew in 1566. We also know that Catherine died in 1583. So 1583 - 1566 = 17 years after here marriage to Killigrew did Catherine die. So the answer is 17.

**Analysis**   Chain-of-thought generated the wrong final answer because one fact it used, "Killigrew married Jaél de Peigne in 1590", is unrelated.

### 9.5    ERROR ANALYSIS: LEAST-TO-MOST PROMPTING

We randomly picked 20 failure cases, and found that out of those 20 failure cases:

- 4 are due to wrong problem decomposition – the decomposed problems do not make sense, or there is no decomposition at all.

- 13 are due to wrong problem solving – it gave the wrong answer for a decomposed problem.

- 3 are due to wrong "ground truth" – the given label is wrong regardless whether the prediction is correct or not (although, the model predictions for the sampled 4 cases are all correct).

### 9.5.1 EXAMPLE OF WRONG PROBLEM DECOMPOSITION

In the following example, the decomposed question is just a rephrase of the original question.

Q: Then, in 1544, five French galleys under Polin, including the superb Réale, accompanied Barbarossa's fleet, on a diplomatic mission to Suleiman. The French fleet accompanied Barbarossa during his attacks on the west coast of Italy on the way to Constantinople, as he laid waste to the cities of Porto Ercole, Giglio, Talamona, Lipari and took about 6,000 captives, but separated in Sicily from Barbarossa's fleet to continue alone to the Ottoman capital. Jerôme Maurand, a priest of Antibes who accompanied Polin and the Ottoman fleet in 1544, wrote a detailed account in Itinéraire d'Antibes à Constantinople. They arrived in Constantinople on 10 August 1544 to meet with Suleiman and give him an account of the campaign. Polin was back to Toulon on 2 October 1544. On his way to Constantinople, how many cities did Polin laid waste to?

A: To answer the question "On his way to Constantinople, how many cities did Polin laid waste to?", we need to know: "How many cities did Polin laid waste to on his way to Constantinople?".

### 9.5.2 EXAMPLE OF WRONG PROBLEM SOLVING

In the following example, the answer to the decomposed question, "How many percent of people were from 2 or more races in 2000?" is wrong.

As of 2011 the 88,112 residents of Orem had a racial and ethnic composition of 89.3% White American, 0.9% black or African American, 0.9% Native Americans of the United States, 1.6% Asian American, 0.5% Pacific Islanders, 4% non-Hispanics reporting some other race, 2.9% two or more races reported and 14.8% Hispanic, as Orem has a large Mexican American community with other Latinos residing in the city. This contrasts with the census of 2000, which showed a racial makeup of 90.80% White (U.S. Census), 0.33% African American (U.S. Census), 0.73% Native American (U.S. Census), 1.45% Asian (U.S. Census), 0.86% Pacific Islander (U.S. Census), 3.64% from Race (United States Census), and 2.18% from two or more races. Hispanic (U.S. Census) or Latino (U.S. Census) of any race were 8.56% of the population. The 2000 Census counted 84,324 people, 23,382 households, and 19,079 families. The population density at that time was 4,572.6 people per square mile (1,765.6/km$^2$). There were 24,166 housing units at an average density of 1,310.4 per square mile (506.0/km$^2$). There were 23,382 households out of which 48.8% had children under the age of 18 living with them, 69.0% were Marriage living together, 9.5% had a female householder with no husband present, and 18.4% were non-families. 12.4% of all households were made up of individuals and 5.1% had someone living alone who was 65 years of age or older. The average household size was 3.57 and the average family size was 3.93.

Q: How many percent of people were from 2 or more races in 2000?
A: "3.64% from Race (United States Census), and 2.18% from two or more races". So the answer is 3.64% + 2.18% = 5.82%.

Q: How many percent of people were not from 2 or more races in 2000?
A: We know that 5.82% were from 2 or more races. So 100% - 5.82% = 94.18% were not from 2 or more races. So the answer is 94.18%.

### 9.5.3 EXAMPLE OF WRONG GIVEN LABEL

In the example below, the given label is 417, which is incorrect. The correct ground truth should be 418. The model gives the correct answer via least-to-most prompting.

There were 20,928 births in 2006. Of these, 19,757 (94.40% of the births, 95.19% of the population) were to Non-Hispanic Whites. There were 22 births to American Indians (0.11% of the births and 0.54% of the population), 177 births to Asians (0.85% of the births and 0.68% of the population), 219 births to Hispanics (1.05% of the births and 0.88% of the population) and 753 births to Blacks and others (3.60% of the births and 3.56% of the population).

Q: How many births were there for Asians?
A: "177 births to Asians". So the answer is 177.

Q: How many births were there for Native Americans?
A: "22 births to American Indians". So the answer is 22.

Q: How many births were there for Hispanics?
A: "219 births to Hispanics". So the answer is 219.

Q: How many total births were there for Asians, Native Americans and Hispanics?
A: We know that there were 177 births for Asians. We also know that there were 22 births for Native Americans. We also know that there were 219 births for Hispanics. So 177 + 22 + 219 = 418 total births for Asians, Native Americans and Hispanics. So the answer is 418.

# 10 GSM8K

## 10.1 EXPERIMENT RESULTS: ONE-SHOT PROMPTS

We compare here the effectiveness on compositional generalization of least-to-most prompting vs. chain-of-thought prompting by constructing for each prompting method a simple prompt context that contains a single example that is solvable with just 2 reasoning steps. We then evaluate accuracy on examples that may involve larger numbers of reasoning steps. The same example is used for both prompting methods. For the least-to-most prompting prompt, we adopt a simplified approach in which the problem decomposition and solution stages are merged into a single pass, with just one follow-up request being made to the language model to solicit the final answer.

The accuracy (%) of the two prompting methods with the GPT-3 `code-davinci-002` model, with breakdown by number of steps in the expected solution, are listed in Table 18.

Accuracy numbers for all prompting methods are calculated after applying the same post-processing as described in Section 3.3 for DROP.

While the least-to-most prompting accuracy is overall only moderately higher than that of chain-of-thought prompting, the accuracy breakdown by number of steps shows that least-to-most prompting significantly outperforms chain-of-thought prompting as the number of reasoning steps increases beyond what was illustrated in the prompt.

| Accuracy by Steps | All | 2 | 3 | 4 | 5+ |
|---|---|---|---|---|---|
| Least-to-Most (1-shot): $a_L$ | 62.39 | 74.53 | 68.91 | 59.73 | 45.23 |
| Chain-of-Thought (1-shot): $a_C$ | 60.87 | 76.68 | 67.29 | 59.39 | 39.07 |
| Accuracy change: $(a_L/a_C) - 1$ | +2.49 | -2.80 | +2.40 | +0.58 | +15.77 |

Table 18: Accuracy (%) of a simple 1-shot least-to-most prompt with the GPT-3 `code-davinci-002` model on GSM8K, compared to that of a corresponding chain-of-thought prompt, broken down by number of reasoning steps required in the expected solution. Examples with 3 or more reasoning steps would require generalizing to more steps than were shown in the demonstration example in the prompt (which contains just 2 steps).

## 10.2 EXPERIMENT RESULTS: ENGINEERED PROMPTS

We compare here the overall accuracy of the above-reported "Chain-of-Thought (1-shot)" and "Least-to-Most (1-shot)" methods with alternative existing prompting methods, as well as with variants of chain-of-thought and least-to-most prompting in which the prompts were engineered using multiple in-domain examples taken from the GSM8K train set.

The evaluated prompting methods are as follows (see Appendices 10.3 and 10.4 for the exact prompt contexts):

- **Zero-Shot**: Simple zero-shot prompting.
- **Standard prompting**: Standard few-shot prompting, using the same 4 examples as in the "problem solving" prompt context of "Least-to-Most (best)".

- **Chain-of-Thought (original)**: Chain-of-thought prompting, using the original 8-shot prompt context described in Wei et al. (2022).

- **Chain-of-Thought (1-shot)**: The simple 1-shot chain-of-thought prompting method described in Appendix 10.1 above.

- **Least-to-Most (1-shot)**: The simple 1-shot least-to-most prompting method described in Appendix 10.1 above.

- **Chain-of-Thought (best)**: Chain-of-thought prompting, using the same 4 examples as in the "problem solving" prompt context of "Least-to-Most (best)", with the solutions adjusted to chain-of-thought format.

- **Least-to-Most (best)**: Least-to-most prompting using separate prompts for the "problem decomposition" and "problem solution" steps and with multiple examples selected from the GSM8K train set. The "problem decomposition" prompt context contains 7 examples, with hand-crafted problem decompositions. The "problem solution" prompt contains 4 examples, with hand-crafted solutions for each step.

The accuracies (%) of these prompting methods with the GPT-3 `code-davinci-002` model are listed in Table 19.

It can be noted first that, although the "Chain-of-Thought (1-shot)" prompt context is considerably simpler than the 8-shot prompt context proposed in the original chain-of-thought paper, we find the overall accuracy achieved to be quite close (60.87% for the 1-shot prompt, compared to 61.18% for the original 8-shot prompt). This suggests that "Chain-of-Thought (1-shot)" is indeed a reasonable chain-of-thought baseline to analyze in comparison to "Least-to-Most (1-shot)".

Further, while we find the proposed 1-shot prompt attractive due to its simplicity and lack of dataset-specific content, we do find that further improvements in overall accuracy of both chain-of-thought and least-to-most prompting can be achieved if additional prompt engineering is applied, using multiple in-domain examples of arbitrary complexity from the GSM8K train set, as seen in the accuracies of "Chain-of-Thought (best)" and "Least-to-Most (best)". We do not observe improvement in overall accuracy from least-to-most prompting compared to chain-of-thought prompting in this setting, where most of the test questions do not require more steps more steps to solve than the demonstration examples.

| Prompting method | Accuracy |
|---|---|
| Zero-Shot | 16.38 |
| Standard prompting | 17.06[3] |
| Chain-of-Thought (original) | 61.18 |
| Chain-of-Thought (1-shot) | 60.88 |
| Least-to-Most (1-shot) | 62.39 |
| Chain-of-Thought (best) | **68.61**[3] |
| Least-to-Most (best) | 68.01 |

Table 19: Accuracies (%) of various prompting methods with the GPT-3 `code-davinci-002` model on GSM8K.

## 10.3 PROMPT CONTEXTS: ONE-SHOT PROMPTS

We include here the prompt contexts used in the experiments reported in Appendix 10.1.

In this section and in the following one, the placeholder "{question}" indicates the place where the original question is to be inserted, in cases where the format would not be obvious (e.g., where

---

[3]Note that in two of the prompt contexts used in an earlier pre-print of this paper, one of the examples had contained a mistake, which has been corrected in this version of the paper. Specifically, in the "Chain-of-Thought (best)" context, the last example had mistakenly omitted the final step, such that it ended incorrectly with "The answer is 17." rather than "which means that sandy will get $20 - $17 = $3 as change. The answer is 3." Similarly, the "Standard prompting" context incorrectly stated the answer as "17" rather than "3" for this example. The earlier versions of the prompts yielded the following accuracies: Standard prompting = 18.65, Chain-of-Thought (best) = 62.77.

instead of simply ending the prompt with "A:", we include some additional prompt text like "A: The answer is").

In the case of "Least-to-Most (1-shot)", the prompt prefix for the initial request ends in "Let's break down this problem:". We then append to that prompt the initial reply that was received from the language model, followed by a newline and the string "The answer is:", which we then use as the prompt in a second request, whose reply we treat as the final answer.

### 10.3.1 CHAIN-OF-THOUGHT (1-SHOT)

Q: Elsa has 5 apples. Anna has 2 more apples than Elsa. How many apples do they have together?
A: Anna has 2 more apples than Elsa, so Anna has 2 + 5 = 7 apples. Elsa and Anna have 5 + 7 = 12 apples together. The answer is 12.

### 10.3.2 LEAST-TO-MOST (1-SHOT)

Q: Elsa has 5 apples. Anna has 2 more apples than Elsa. How many apples do they have together?
A: Let's break down this problem: 1. How many apples does Anna have? 2. How many apples do Elsa and Anna have together?
1. Anna has 2 more apples than Elsa. So Anna has 2 + 5 = 7 apples.
2. Elsa and Anna have 5 + 7 = 12 apples together.

Q: {question}
A: Let's break down this problem:
——
The answer is:

## 10.4 PROMPT CONTEXTS: ENGINEERED PROMPTS

We include here the additional prompt templates used in the experiments reported in Appendix 10.2, with the exception of "Chain-of-Thought (original)", which is the same as the "Prompt for Math Word Problems" reported in Appendix G of Wei et al. (2022).

### 10.4.1 ZERO-SHOT

Q: {question}
A: The answer is

### 10.4.2 STANDARD PROMPTING: 4 EXAMPLES

Q: Four years ago, Kody was only half as old as Mohamed. If Mohamed is currently twice 30 years old, how old is Kody?
A: The answer is 32.

Q: Carla bought 2 bags of mini peanut butter cups on clearance. Each bag was $6.00 but was 75% off. How much did she spend on 2 bags of candy?
A: The answer is 3.

Q: If Pam is currently twice as young as Rena is, and in 10 years Rena will be 5 years older than her, how old is Pam now?
A: The answer is 5.

Q: Cappuccinos cost $2, iced teas cost $3, cafe lattes cost $1.5 and espressos cost $1 each. Sandy orders some drinks for herself and some friends. She orders three cappuccinos, two iced teas, two cafe lattes, and two espressos. How much change does she receive back for a twenty-dollar bill?
A: The answer is 3.

### 10.4.3 CHAIN-OF-THOUGHT (BEST): 4 EXAMPLES

Q: Four years ago, Kody was only half as old as Mohamed. If Mohamed is currently twice 30 years old, how old is Kody?
A: We were told that Mohamed is currently twice 30 years old, so he is currently 30 * 2 = 60 years old. That means that four years ago he must have been 60 - 4 = 56 years old. Four years ago, Kody was half as old as Mohamed, so Kody must have been 56 / 2 = 28 years old then. Since Kody was 28 years old four years ago, she must now be 28 + 4 = 32 years old. The answer is 32.

Q: Carla bought 2 bags of mini peanut butter cups on clearance. Each bag was $6.00 but was 75% off. How much did she spend on 2 bags of candy?
A: Each bag was $6.00 but was 75% off. So each bag cost $6.00 * (1 - 0.75) = $6.00 * 0.25 = $1.50. Carla bought 2 bags. So she spent $1.50 * 2 = $3.00. The answer is 3.

Q: If Pam is currently twice as young as Rena is, and in 10 years Rena will be 5 years older than her, how old is Pam now?
A: Since Rena will be 5 years older than Pam in 10 years, she must be 5 years older than Pam now as well. If Pam is currently twice as young as Rena, that means that Rena is currently twice as old as Pam is. So if P stands for Pam's age now and R stands for Rena's age now, then we know that R = 2 * P And since Rena is 5 years older than Pam now, we know that R = P + 5. By substitution, we have P + 5 = 2 * P, which means that P = 5. The answer is 5.

Q: Cappuccinos cost $2, iced teas cost $3, cafe lattes cost $1.5 and espressos cost $1 each. Sandy orders some drinks for herself and some friends. She orders three cappuccinos, two iced teas, two cafe lattes, and two espressos. How much change does she receive back for a twenty-dollar bill?
A: Sandy ordered three cappuccinos, which cost $2 each, so she spent $2 * 3 = $6 on cappuccinos. She ordered two iced teas, which cost $3 each, so she spent $3 * 2 = $6 dollars on ice teas. She ordered two cafe lattes, which cost $1.5 each, so she spent $1.5 * 2 = $3 on cafe lattes. She ordered two espressos, which cost $1 each, so she spent $1 * 2 = $2 on espressos. So altogether, Sandy spent $6 + $6 + $3 + $2 = $17 on drinks, which means that sandy will get $20 - $17 = $3 as change. The answer is 3.

### 10.4.4 LEAST-TO-MOST (BEST) I - PROBLEM DECOMPOSITION: 7 EXAMPLES

Q: Four years ago, Kody was only half as old as Mohamed. If Mohamed is currently twice 30 years old, how old is Kody?
A: To answer the question "How old is Kody?", we need to know: "How old is Mohamed?", "How old was Mohamed four years ago?", "How old was Kody four years ago?".

Q: If Pam is currently twice as young as Rena is, and in 10 years Rena will be 5 years older than her, how old is Pam now?
A: To answer the question "How old is Pam now?", we need to know: "How much older is Rena than Pam currently?".

Q: As a freelancer, Baylor is paid for every finished work of a client he does on a freelance marketplace. Currently, he has $4000 on his dashboard from previous work done. He is currently working for three clients, with the first client paying him half the amount of money he currently has on his dashboard once the job is done. The second client will pay him 2/5 times more money than the first client once Baylor finishes his work. The third client will pay him twice the amount of money the first and second clients pay him together once he finishes the job. How much money will Baylor have in his dashboard after all the clients pay him for his work?
A: To answer the question "How much money will Baylor have in his dashboard after all the clients pay him for his work?", we need to know: "How much will Baylor's first client pay him for his work?", "How much more will Baylor's second client pay him for his work compared to the first client?", "How much will Baylor's second client pay him for his work?", "How much will the first and second clients pay him together once he finishes the job?", "How much will Baylor's third client pay him for his work?", "How much money will all the clients pay Baylor for his work?".

Q: Cappuccinos cost $2, iced teas cost $3, cafe lattes cost $1.5 and espressos cost $1 each. Sandy orders some drinks for herself and some friends. She orders three cappuccinos, two iced teas, two cafe lattes, and two espressos. How much change does she receive back for a twenty-dollar bill?
A: To answer the question "How much change does she receive back for a twenty-dollar bill?", we need to know: "How much did the cappuccinos cost in total?", "How much did the iced teas cost in total?", "How much did the cafe lattes cost in total?", "How much did the espressos cost in total?", "How much did the drinks cost in total?".

Q: Betty & Paige are raising money for their kids' little league team by hosting a bake sale. Betty has baked 4 dozen chocolate chip cookies, 6 dozen oatmeal raisin cookies and 2 dozen regular brownies. Paige baked 6 dozen sugar cookies, 3 dozen blondies and 5 dozen cream cheese swirled brownies. If they sell the cookies for $1.00 apiece and the blondies/brownies at $2.00 apiece, how much money will they raise?
A: To answer the question "How much money will they raise?", we need to know: "How many dozen cookies did they bake (not including blondies/brownies)?", "How many cookies did they bake (not including blondies/brownies)?", "How many dozen blondies/brownies did they bake (not including cookies)?", "How many blondies/brownies did they bake (not including cookies)?", "How much money will they raise from the cookies (not including blondies/brownies)?", "How much money will they raise from the blondies/brownies (not including cookies)?".

Q: On a moonless night, three fireflies danced in the evening breeze. They were joined by four less than a dozen more fireflies, before two of the fireflies flew away. How many fireflies remained?
A: To answer the question "How many fireflies remained?", we need to know: "How many fireflies joined?".

Q: Sam, Sid, and Steve brought popsicle sticks for their group activity in their Art class. Sam has thrice as many as Sid, and Sid has twice as many as Steve. If Steve has 12 popsicle sticks, how many popsicle sticks can they use for their Art class activity?
A: To answer the question "How many popsicle sticks can they use for their Art class activity?", we need to know: "How many popsicle sticks does Sid have?", "How many popsicle sticks does Sam have?".

### 10.4.5 LEAST-TO-MOST (BEST) II - PROBLEM SOLVING: 4 EXAMPLES

Four years ago, Kody was only half as old as Mohamed. If Mohamed is currently twice 30 years old, how old is Kody?

Q: How old was Mohamed four years ago?
A: We were told that Mohamed is currently twice 30 years old, so he is currently 30 * 2 = 60 years old. That means that four years ago he must have been 60 - 4 = 56 years old. The answer is 56.

Q: How old is Kody?
A: Four years ago, Kody was half as old as Mohamed, so Kody must have been 56 / 2 = 28 years old then. Since Kody was 28 years old four years ago, she must now be 28 + 4 = 32 years old. The answer is 32.

Carla bought 2 bags of mini peanut butter cups on clearance. Each bag was $6.00 but was 75% off. How much did she spend on 2 bags of candy?

Q: How much did she spend on 2 bags of candy?
A: Each bag was $6.00 but was 75% off. So each bag cost $6.00 * (1 - 0.75) = $6.00 * 0.25 = $1.50. Carla bought 2 bags. So she spent $1.50 * 2 = $3.00. The answer is 3.

If Pam is currently twice as young as Rena is, and in 10 years Rena will be 5 years older than her, how old is Pam now?

Q: How much older is Rena than Pam currently?
A: Since Rena will be 5 years older than Pam in 10 years, she must be 5 years older than Pam now as well. The answer is 5.

Q: How old is Pam now?
A: If Pam is currently twice as young as Rena, that means that Rena is currently twice as old as Pam is. So if P stands for Pam's age now and R stands for Rena's age now, then we know that R = 2 * P And since Rena is 5 years older than Pam now, we know that R = P + 5. By substitution, we have P + 5 = 2 * P, which means that P = 5. The answer is 5.

Cappuccinos cost $2, iced teas cost $3, cafe lattes cost $1.5 and espressos cost $1 each. Sandy orders some drinks for herself and some friends. She orders three cappuccinos, two iced teas, two cafe lattes, and two espressos. How much change does she receive back for a twenty-dollar bill?

Q: How much did Sandy spend on drinks?
A: Sandy ordered three cappuccinos, which cost $2 each, so she spent $2 * 3 = $6 on cappuccinos. She ordered two iced teas, which cost $3 each, so she spent $3 * 2 = $6 dollars on ice teas. She ordered two cafe lattes, which cost $1.5 each, so she spent $1.5 * 2 = $3 on cafe lattes. She ordered two espressos, which cost $1 each, so she spent $1 * 2 = $2 on espressos. So altogether, Sandy spent $6 + $6 + $3 + $2 = $17 on drinks. The answer is 17.

