# OpenReview forum: "Least-to-Most Prompting Enables Complex Reasoning in Large Language Models"
_ICLR.cc/2023/Conference — ICLR 2023 poster_

### Official Review · Reviewer_Vyt2 · 2022-10-14

**Confidence:** 4
**Correctness:** 3
**Technical Novelty And Significance:** 3
**Empirical Novelty And Significance:** 3
**Recommendation:** 6

**Clarity, Quality, Novelty And Reproducibility:**

This work is interesting and original.

In terms of clarity, it can be improved further:
- It's not easy to understand how the full pipeline would work for a complex problem (which needs to be broken down recursively many times). Does stage 1 consists of a single pass through the model, to directly output how to break down the problem into the easy pieces?
- In stage 2, when solving each subproblem, is the statement of the subproblem copied from the output of stage 1 or is it generated by the model (in which case probably the output of stage 1 is part of the prompt for stage 2)?

It would help to have one full example, where each forward pass through the model is clear, and in each forward pass it is clear what is the prompt and what is the model's continuation.

**Strength And Weaknesses:**

The proposed technique is new and interesting. It is a nice addition to the literature on prompting large language models to solve complex tasks.

I see a major limitation with this approach: it looks like the prompt examples need to be designed specifically for each dataset. Additionally, the prompt examples add significant supervision on how the task should be solved, and this supervision is not present when using other methods (such as standard prompting).

For instance, in the symbolic manipulation task, the examples for least-to-most prompting are constructed by exploiting the property "concatenating the last letters of $n$ words is the same as concatenating the last letters of the first $n-1$ words, and then concatenating the result with the last letter of the $n$-th word". In my opinion, learning this property is an important part of learning to solve the task algorithmically.

**Summary Of The Paper:**

This paper introduces a new prompting techniques for large language models, called least-to-most prompting. It consists in two stages: (1) prompt the model to ask it to break down the given problem into easier subproblems; (2) prompt the model to solve each subproblem. Experimental results are shown for datasets of symbolic manipulation, compositional generalization, and math reasoning, with GPT-3 models.

**Summary Of The Review:**

The proposed approach is interesting, but it also appear to have a significant limitation (you need to provide supervision that provides information on how the task needs to be solved). This limitation seems to make it difficult to apply the approach more broadly (dataset-specific supervision is required) and makes comparisons with other methods not completely fair (because this approach has effectively more supervision about the task).

---

> ### Author Response · Authors · 2022-11-17
> **Authors’ rebuttal [2/2]: how the full pipeline would work for a complex problem**
>
>
> **Concern: It's not easy to understand how the full pipeline would work for a complex problem**
>
> We can think of least-to-most prompting as a conversation between two LLMs where one guides the other to solve a complex problem step by step.
>
> Let us go through a full example to understand the pipeline. Assume we solve the last-latter task.
>
> We first manually design two prompts respectively for problem decomposition and solving.
>
> **Prompt for decomposition:**
>
> ---
>
> Q: "machine, learning"
>
> A: creating sequential sublists of the list "machine, learning":
> "machine"
> "machine, learning"
>
>
> Q: "machine, learning, artificial"
>
> A: creating sequential sublists of the list "machine, learning, artificial":
> "machine"
> "machine, learning"
> "machine, learning, artificial"
>
> ---
>
> **Prompt for solving:**
>
> ---
> Q: “think, machine”
>
> A: The last letter of “think” is “k”. The last letter of “machine” is “e”. Concatenating “k”, “e” leads to “ke”. So, “think, machine” outputs “ke”.
>
>
>
> Q: “think, machine, learning”
>
> A: “think, machine” outputs “ke”. The last letter of “learning” is “g”. Concatenating “ke”, “g” leads to “keg”. So, “think, machine, learning” outputs “keg”.
>
> ---
>
> Now let use the above prompts to solve “mind, emerge, from, neuron”.
>
> **Step 1.**  Pass the following prompt to the LLM to decompose “mind, emerge, from, neuron”
>
> ---
>
> Q: "machine, learning"
>
> A: creating sequential sublists of the list "machine, learning":
> "machine"
> "machine, learning"
>
> Q: "machine, learning, artificial"
>
> A: creating sequential sublists of the list "machine, learning, artificial":
> "machine"
> "machine, learning"
> "machine, learning, artificial"
>
> Q: “mind, emerge, from, neuron”
>
> ---
>
> We will see the following output:
>
> ---
>
> A: creating sequential sublists of the list “mind, emerge, from, neuron”:
> “mind”
> “mind, emerge”
> “mind, emerge, from”
> “mind, emerge, from, neuron”
>
> ---
>
>
> **Step 2.**  We pass the following prompt to the LLM to solve the first sublist  “mind, emerge”:
>
> ---
>
> Q: “think, machine”
>
> A: The last letter of “think” is “k”. The last letter of “machine” is “e”. Concatenating “k”, “e” leads to “ke”. So, “think, machine” outputs “ke”.
>
> Q: “think, machine, learning”
>
> A: “think, machine” outputs “ke”. The last letter of “learning” is “g”. Concatenating “ke”, “g” leads to “keg”. So, “think, machine, learning” outputs “keg”.
>
> Q:  “mind, emerge
>
> ---
>
> We will see the following output:
>
> ---
>
> A: The last letter of “mind” is “d”. The last letter of “emerge” is ”e”. Concatenating “d”, “e” leads to “de”. So, “mind, emerge” outputs “de”.
>
> ---
>
> **Step 3.**  We  pass the following prompt to the LLM to solve the second sublist “mind, emerge, from”
>
> ---
>
> Q: “think, machine”
>
> A: The last letter of “think” is “k”. The last letter of “machine” is “e”. Concatenating “k”, “e” leads to “ke”. So, “think, machine” outputs “ke”.
>
> Q: “think, machine, learning”
>
> A: “think, machine” outputs “ke”. The last letter of “learning” is “g”. Concatenating “ke”, “g” leads to “keg”. So, “think, machine, learning” outputs “keg”.
>
> Q:  “mind, emerge”
>
> A: The last letter of “mind” is “d”. The last letter of “emerge” is ”e”. Concatenating “d”, “e” leads to “de”. So, “mind, emerge” outputs “de”.
>
> Q: “mind, emerge, from”
>
> ---
>
> We will see the following output:
>
> ---
>
> A: “mind, emerge” outputs “de”. The last letter of “from” is “m”. Concatenating “de”, “m” leads to “dem”. So, “mind, emerge, from” outputs “dem”.
>
> ---
>
> **Step 4.**  We pass the following prompt to the LLM to solve the last sublist “mind, emerge, from, neuron”:
>
> ---
>
> Q: “think, machine”
> A: The last letter of “think” is “k”. The last letter of “machine” is “e”. Concatenating “k”, “e” leads to “ke”. So, “think, machine” outputs “ke”.
>
> Q: “think, machine, learning”
> A: “think, machine” outputs “ke”. The last letter of “learning” is “g”. Concatenating “ke”, “g” leads to “keg”. So, “think, machine, learning” outputs “keg”.
>
> Q:  “mind, emerge”
> A: The last letter of “mind” is “d”. The last letter of “emerge” is ”e”. Concatenating “d”, “e” leads to “de”. So, “mind, emerge” outputs “de”.
>
> Q: “mind, emerge, from”
> A: “mind, emerge” outputs “de”. The last letter of “from” is “m”. Concatenating “de”, “m” leads to “dem”. So, “mind, emerge, from” outputs “dem”.
>
> Q: “mind, emerge, from, neuron”
>
> ---
>
> We will see the following output:
>
> ---
>
> A: “mind, emerge, from” outputs “dem”. The last letter of “neuron” is “n”. Concatenating “dem”, “n” leads to “demn”. So, “mind, emerge, from, neuron” outputs “demn”.
>
> ---
>
> Till here the problem is solved.

---

> ### Author Response · Authors · 2022-11-17
> **Authors’ rebuttal [1/2]: major limitations --- the prompt examples need to be designed specifically for each dataset**
>
> Thank you for your time reviewing our submission and constructive comments. Below is our rebuttal for addressing your concerns. We would be more than happy to follow up if you have any further issues.
>
> **Concern: “I see a major limitation with this approach: it looks like the prompt examples need to be designed specifically for each dataset.”  “have a significant limitation (you need to provide supervision that provides information on how the task needs to be solved). This limitation seems to make it difficult to apply the approach more broadly (dataset-specific supervision is required)”.**
>
> We agree this is a major limitation of our work. But we would like to point out that this is also a major limitation for the whole field of machine learning, like supervised learning, weakly surprised learning, RL, and few-shot prompting. All of them need task-specific examples, for training, finetuning or prompting.
>
> **Our work here is not aimed at eliminating this limitation but at easing this limitation by dramatically reducing the number of task specific examples that are required**. For example, in the SCAN task, we only need to show 14 examples to demonstrate how to map natural language commands to action sequences while the sota methods in the literature need tens of thousands of labeled examples.
>
>
> **Concern: “Additionally, the prompt examples add significant supervision on how the task should be solved, and this supervision is not present when using other methods (such as standard prompting).”  “makes comparisons with other methods not completely fair (because this approach has effectively more supervision about the task).”**
>
> For a comparison among different prompting methods with a similar amount of information , please check here: https://openreview.net/forum?id=WZH7099tgfM&noteId=OBa3FNMvrh
>
> The key message is that **we need to look at the ratio of performance gain over provided information, i.e. data efficiency, the most fundamental issue in machine learning**, instead of a single factor only. Let’s use the last-letter task as an example to explain this point.
>
> - It would be best if standard prompting could work since it needs least information in each example. However, as shown in Table 3, few-shot standard prompting’s accuracy is zero in the last-letter task.  To improve its performance, one would consider adding many more examples. Actually, in this task, no matter how many examples are used, standard prompting’s accuracy remains zero. However, few-shot chain-of-thought prompting, which just modifies the prompting examples by adding a chain-of-thought before the final answer, achieves much better performance.  So the overall data efficiency is greatly improved.
>
> - Again, to improve further the accuracy, one would consider adding more chain-of-thought examples. However, it turns out that it is much more data efficient to use least-to-most prompting which modifies chain-of-thought examples by introducing decomposition. As shown in the results (see the link above),  the accuracy after doubling the chain-of-thought examples is still worse than the accuracy of least-to-most prompting with a large margin!
>
> **Concern: For instance, in the symbolic manipulation task, the examples for least-to-most prompting are constructed by exploiting the property "concatenating the last letters of n words is the same as concatenating the last letters of the first  n-1 words, and then concatenating the result with the last letter of the n-th word". In my opinion, learning this property is an important part of learning to solve the task algorithmically.**
>
> This is again a data efficiency issue. You may need thousands of examples to train a model to learn this property. Here we show an alternative that we can directly teach LLMs to understand the recursion by only 2 examples. Thus, giving these two options: (1) label thousands of examples; or (2) write several lines of instructions to teach LLMs. Which one would you choose in practice? :)

---

### Official Review · Reviewer_d1S7 · 2022-10-25

**Confidence:** 4
**Correctness:** 3
**Technical Novelty And Significance:** 3
**Empirical Novelty And Significance:** 4
**Recommendation:** 8

**Clarity, Quality, Novelty And Reproducibility:**

**Clarity**

The reply of the authors was greatly appreciated and helped me understand the paper better. It points however to the intrinsic difficulty of explaining the experimental setup when this one involves a large amount of text and successive calls to a LM that modifies some parts of the prompt. I encourage the authors to experiment with different and simple visualization techniques that could convey this better. Related to this - and a detail -: unifying the colours used to highlight the re-use of answers in Fig 1 and the various tables would improve clarity as as well.

**Novelty**
Very good

**Reproducibility**
The authors made sure to show results not only on their proprietary model, but also on an API - whose access and underlying model however can change abruptly. The authors have limited influence on that however

**Quality**
The evaluated datasets are limited, and the impact of prompt engineering is not assessed. Having said that, and considering the fast-moving pace of research into the capabilities of LLM, I consider that this is a worthwhile addition to the ICLR roster, and constitutes a paper which will be analyzed and cited in the immediate future


**Strength And Weaknesses:**

STRENGHTS
+ As in CoT the proposed method does not require any training or fine-tuning. The experiments reported here can be done solely by interacting with the API of a large language model provider
+ The performance jump is significant on problems considered difficult

WEAKNESSES
- The prompt are designed task-specific. It is not clear what was the validation set to create those prompts. A robustness study on using different prompts would be welcome and strengthen the message that the results hold across a variety of lexical variations of the same idea
- Clarity of the paper (see below)

**Summary Of The Paper:**

This paper follows-up on recent work around "chain-of-thought" (CoT) prompting, where the expected output of the model is not only the desired output, but also rationales that would reflect reasoning steps that a human would use to solve that same problem.

The additional insight of this paper is that this process can be made hierarchical, by first requesting the model to output a series of subproblem such that solving those subproblems would naturally lead to the solution of the initial problem. After this, each subproblem is solved sequentially, with the previous answer appended to the prompt before requesting to solve the next sub-problem.


========= OLD REVIEW - PRE ANSWER BY AUTHORS =========

I would like to hold back with my review until the authors get an opportunity to reply (long live review discussion!). So please disregard the rest of the fields, including final assessment. Apologies if my question is somehow obvious, but I could not find an answer to this through repetitive reading of the paper:

Could the authors please clarify which parth is given as prompt to the LM, and which part is generated? From the colored highlight it seems to me that:
 - in Fig 1, the subquestion 2 is provided by the human, and not generated by the LLM.
 - my understanding is that in Table 2 and 5, everything that comes before "So," is given as a prompt (in addition of the few-shot examples of course) in the different calls to the API. This would mean for example that in Table 2:
     * you call OpenAI's API a first time, with the prompt before the right-side of Table 1 + everything before the first blue highlighted text (up to 'leads to "de". So,').
     * you call OpenAI's API a second time, with everything up to the first red highlighted text: up to 'leads to "dem", So,'

If that is the case, then this result seems underwhelming to me - and not corresponding to the claims. Because in that case the exercise seems rather trivial as all the heavy-lifting is done by the prompts which - again if my understanding is correct - are manually written and not machine-generated.

thanks a lot!

**Summary Of The Review:**

Another surprising discovery into capabilities of LLM, showing great improvement on existing tasks without the need to finetune.

---

> ### Author Response · Authors · 2022-11-13
> **Rebuttal [2/2]: Illustration of L2M prompting**
>
> As a practical illustration, let us go over a simplified version of least-to-most prompting, which combines decomposition (reduction) and solving in a single pass (see Table 11).
>
> For the given task of math word problems (GSM8K), let us use the following manually written prompt context that consists of a single exemplar:
>
> ---
>
> Q: Elsa has 5 apples. Anna has 2 more apples than Elsa. How many apples do they have together?
>
> A: Let’s break down this problem: 1. How many apples does Anna have? 2. How many apples do Elsa and Anna have together?
>
> 1. Anna has 2 more apples than Elsa. So Anna has 2 + 5 = 7 apples.
> 2. Elsa and Anna have 5 + 7 = 12 apples together.
>
> ---
>
> Now assume that we need to solve the following test example:
>
> Eliza's rate per hour for the first 40 hours she works each week is $10. She also receives an overtime pay of 1.2 times her regular hourly rate. If Eliza worked for 45 hours this week, how much are her earnings for this week?
>
> To solve this problem, we take the manually written prompt context and append to it the test example:
>
> ---
>
> Q: Elsa has 5 apples. Anna has 2 more apples than Elsa. How many apples do they have together?
>
> A: Let’s break down this problem: 1. How many apples does Anna have? 2. How many apples do Elsa and Anna have together?
>
> 1. Anna has 2 more apples than Elsa. So Anna has 2 + 5 = 7 apples.
> 2. Elsa and Anna have 5 + 7 = 12 apples together.
>
> Q: Eliza's rate per hour for the first 40 hours she works each week is \$10. She also receives an overtime pay of 1.2 times her regular hourly rate. If Eliza worked for 45 hours this week, how much are her earnings for this week?
>
> ---
>
> We then pass this prompt to the language model (in this case code-davinci-002), which yields the following automatically generated output:
>
> ---
>
> Let's break down this problem: 1. How much does Eliza earn for the first 40 hours she works each week? 2. How much does Eliza earn in overtime pay? 3. How much does Eliza earn in total for the week?
>
> 1. Eliza earns \\$10 per hour for the first 40 hours she works each week, so she earns \\$10 x 40 = \\$400 for those hours.
> 2. Eliza earns 1.2 times her regular hourly rate for overtime hours, so she earns 1.2 x \\$10 = \\$12 per overtime hour. Eliza worked 5 overtime hours this week, so she earned\ \$12 x 5 = \\$60 in overtime pay.
> 3. Eliza's total earnings for the week are \\$400 + \\$60 = \\$460.
>
> ---
>
> The same process is applied for the other test examples of this task. In particular, we use the exact same prompt context for all of them!
>
> Note that the decomposition (reduction) and solving can be separated into 2 prompts to make L2M more flexible, e.g. in the last-letter and SCAN task in the paper. In this case, we use a single manually written prompt context to teach the model to decompose and then solve subquestions. But again, the prompt context remain independent of the test examples (i.e., the same context is used for all test examples).

---

> ### Author Response · Authors · 2022-11-13
> **Rebuttal [1/2]: Address your specific questions in Fig 1, Table 2 and Table 5**
>
> Yes, long live review discussion! We appreciate that you held back your review until we had a chance to reply to your questions. Indeed, there seems to be a misunderstanding here, so let us start by addressing your questions and then give an example of least-to-most (L2M) prompting.
>
> **Addressing your questions**
>
> The most important thing to understand is that **none of the manually written parts of the prompt are specific to the test examples** (in fact, there is absolutely no need for humans to take into account the test examples in any way).
>
> Instead, the only thing that is manually written is a small prompt context consisting of a few exemplars that provide the model with a general illustration of the expected behavior for a given task. This means that **the prompt context is specific to a given task but not to a given test example** (i.e., the exact same prompt context is used for all test examples of the task).
>
> With that, let’s address your specific questions.
>
> $$\textbf{Figure 1}$$
> In stage 1, we pass to the language model the original question. The language model returns the decomposition (reduction) into two subquestions. The first subquestion is “How long does each trip take?” and the second subquestion happens to be the original question “How many times can she ride the slide before it closes?”. The essential thing here is that **the intermediate subquestions are generated by the model**. The last subquestion is the original question that we need to solve. Depending on the prompt context (omitted in the fig), it can be reproduced by the model or manually appended since it has been given/known.
>
> In stage 2, we start by passing to the model the first subquestion (“How long does each trip take?”). The model provides a reply (“It takes Amy 4 minutes to climb and 1 minute to slide down. 4 + 1 = 5. So, each trip takes 5 minute”), which we append to the second subquestion (“How many times can she ride the slide before it closes?”). We further pass them to the model,and the model provides the reply (“The water slide closes in 15 minutes. Each trip takes 5 minutes. So Amy can slide 15 ÷ 5 = 3 times before it closes”).
>
> $$\textbf{Table 2}$$
> As stated in the caption, this table shows a test case for the sequence “mind, emerge, from, neuron”. All the subproblems and their solutions shown in this table are automatically generated by the language model. The highlights are used to communicate to the reader how LLMs “smartly” build their solution to a new subquestion on the solutions of the previously solved subquestions.
>
> $$\textbf{Table 5}$$
> As indicated by the caption, this table shows manually written “demonstration” exemplars from the prompt context for this task. These exemplars teach the model how to solve a given task. For instance, the second exemplar in this table teaches the model how to deal with “around right”. Note again that we use the same prompt context for all test examples of this task!

---

> ### Author Response · Authors · 2022-11-15
> **Thank you for your update**
>
> "The additional insight of this paper is that this process can be made hierarchical, by first requesting the model to output a series of subproblem such that solving those subproblems would naturally lead to the solution of the initial problem. After this, each subproblem is solved sequentially, with the previous answer appended to the prompt before requesting to solve the next sub-problem."
>
> This is a really precise summary. It is exactly the whole point of this work. Decomposition/planning is supposed to be very natural in humans' problem solving. For example, when we write a program, we would first have a high-level sketch/plan, and then write code line by line. The surprising thing is that somehow LLMs can be taught to solve problems in such a way too. So far we haven't noticed any theory could be applied to explain such an incredible coincidence.
>
> We will absolutely follow your helpful suggestions to further polish this paper to make it more clear.

---

### Official Review · Reviewer_YZYL · 2022-10-25

**Confidence:** 4
**Correctness:** 3
**Technical Novelty And Significance:** 3
**Empirical Novelty And Significance:** 3
**Recommendation:** 8

**Clarity, Quality, Novelty And Reproducibility:**

The paper is mostly clear. However, examples in the main paper do not have few-shot demos which made it a bit hard to understand.

The least-to-most prompting idea seems to be novel, somewhat straightforward. There are more principled and better ideas now, e.g.
Maieutic Prompting: Logically Consistent Reasoning with Recursive Explanations
https://arxiv.org/pdf/2205.11822

I believe maieutic prompting is considered concurrent work.

Even though I consider the paper has substantial originality, the paper lacks depth and deep insight on whether this method represents genuine progress in improving big model's reasoning ability or is very dataset dependent.

**Strength And Weaknesses:**

Strength
1. The paper proposes a new prompting strategy for big language models to solve multi-step reasoning tasks. It breaks the problems into sub-problems by querying the language model, and then solves each sub-problem sequentially.

2. The paper shows that the proposed least-to-most prompting significantly improves over the prior work, chain-of-thought prompting on three tasks, symbolic manipulation, compositional generalization and math reasoning.

Weaknesses
1. A major limitation of the paper is the problem reduction step. Since the three datasets are relatively constrained, the few demonstration examples may cover most of the cases on how to break into sub-problems. In other words, the model may only need to copy patterns from the few-shot demos without any generalization.

2. The paper lacks insights on key questions. Does the model have ability to decompose questions very different from demo examples into correct sub-problems? Even the model solves each sub-problem correctly, what makes the model compose the final solution correctly?

3. The paper needs to evaluate on more reasoning problems (e.g. StrategyQA, high school reading comprehension) or clearly state the type of reasoning problem the method works well. It needs a better limitation section that what is stated in the conclusion and discussion section.

**Summary Of The Paper:**

Recently chain-of-thought prompting ("lets think step by step") has demonstrated big language models can perform several reasoning tasks well, e.g. match word problem. However, it is known to not perform well when the problem to solve is harder than the few-shot examples in the prompt. To address the problem, this paper proposes a new prompting method called least-to-most prompting. It consists of two steps. The first step is to reduce a complex problem into a list of subproblems while the second one is to sequentially solve each sub-problem. Answers to previous subproblems are available to solve the next one. The paper evaluates on three types of reasoning tasks, symbolic manipulation, compositional generalization and math reasoning. The paper shows least-to-most prompting outperforms chain-of-thought prompting by a large margin. It is worth noting that GPT- 3 code-davinci-002 model with least-to-most-prompting solves the SCAN benchmark with an accuracy of 99.7% using 14 examples while chain-of-thought prompting only achieves an accuracy of 16.2%.

**Summary Of The Review:**

Big language models have achieved impressive results in NLP. It is natural to think they can solve simple-step problems. To solve hard problems, one just decompose them into sub-problems. This paper presents experimental results on this straight-forward idea. Although the evaluation is very limited, it does have one very impressive results, GPT- 3 code-davinci-002 model with least-to-most-prompting solves the SCAN benchmark with an accuracy of 99.7% using 14 examples while chain-of-thought prompting only achieves an accuracy of 16.2% and neural-symbolic models in the literature specialized for solving SCAN are trained with the full training set of more than 15,000 examples.

To build confidence and understand its limitations, I would encourage the authors to evaluate on other types of reasoning tasks such as StrategyQA, TimeDial, high school reading comprehension, and many others in Bigbench (Beyond the Imitation Game: Quantifying and extrapolating the capabilities of language models https://arxiv.org/pdf/2206.04615.pdf).

I would also encourage the authors to provide deeper insights and analysis as stated in the weakness section.

== Post authors' response

The response provided more examples as evidences of generalization. It is still not clear to me why L2M is able to generalize. The paper only offered a tenuous connection to educational psychology. I think the results are intriguing and L2M worth further research. Therefore, I upgrade my rating from marginal accept to accept.

---

> ### Author Response · Authors · 2022-11-15
> **Authors’ rebuttal [3/3]: other issues**
>
> **Concern: Even the model solves each sub-problem correctly, what makes the model compose the final solution correctly?**
>
> L2M does not have a stage called composing. As shown in the examples of solving math word problems, the original problem is reduced into a list of dependent subproblems with increasing complexities. This is where the name “least-to-most” comes from. The subproblem list can be regarded as a high-level solving sketch or plan. Then, the model sequentially solves these subproblems while the solution to a new subproblem is built over the solutions of previously solved subproblems. Consequently, each subproblem is easily solved within one step. This is how L2M can solve problems harder than the demonstrated examples to achieve a better performance than CoT.
>
> **Concern: examples in the main paper do not have few-shot demos which made it a bit hard to understand.**
>
> All tasks in this paper are few-shot. We have put their full prompts and sampled results generated from LLMs in the Appenedix (52 pages). Moreover, in the main text, we have a full demo for the last-letter task (see the prompt in table 1 ands a test case in table 2).
>
> **Concern: There are more principled and better ideas now, e.g. Maieutic Prompting: Logically Consistent Reasoning with Recursive Explanations.**
>
> Maieutic prompting is a very nice work but we are afraid it is orthogonal to the work here:
> - As stated in their paper, maieutic prompting first generates diverse hypotheses (restricted to using true/false as answers!), and then collectively eliminates the contradicting candidates to obtain consistent answers (using MAX-SAT). Fig 2 in that paper clearly illustrates its key idea.
> - However, L2M aims at solving easy-to-hard generalization by decomposing a complex complex problem into a list of subproblems with increasing complexities. So, in the L2M experiment designs, the demonstration problems usually can be solved within 2 or 3 steps, while the test problems may need many steps to solve (see the GSM8k examples above).
>
> Also note that the maieutic prompting paper does not have results on any task evaluated in this submission. It also does not make any claim on easy-to-hard generalization. If there are papers which have applied maieutic prompting to solve GSM8K/SCAN with great results, or you have constructed maieutic prompts to successfully solve GSM8K/SCAN, we are happy to know.

---

> ### Author Response · Authors · 2022-11-15
> **Authors’ rebuttal [2/3]: StrategyQA**
>
> **Concern: The paper needs to evaluate on more reasoning problems (e.g. StrategyQA, high school reading comprehension).**
>
> We did a quick try on StrategyQA with a simple L2M prompt (similar to the prompt in Table 11) as follows:
>
> ---
>
> Q: Can sunlight travel to the deepest part of the Black Sea?
> A: Let's breakdown this problem: "What is the maximum depth of the Black Sea?", "How deep can sunlight penetrate a sea?", "Can sunlight travel to the deepest part of the Black Sea?"
>
> 1. The Black Sea has a maximum depth of 2,212 meters.
> 2. Sunlight does not penetrate water below 1000 meters.
> 3. 1000 meters is less than 2,212 meters. So sunlight cannot travel to the deepest part of the Black Sea
>
> Then answer is: false.
>
> ---
>
> **Here is a test case** (text-davinci-002 with its default setup):
>
> ---
>
> Q: Did Aristotle Use a Laptop?
> A: Let's breakdown this problem: "When was the laptop invented?", "When did Aristotle live?", "Did Aristotle Use a Laptop?"
>
> 1. The laptop was invented in 1981.
> 2. Aristotle lived from 384 BC to 322 BC.
> 3. 1981 is greater than 322 BC. So Aristotle did not use a laptop.
>
> The answer is: false.
>
> ---
>
> **Here is another test case:**
>
> ---
>
> Q: Are chinchillas cold-blooded?
> A: Let's breakdown this problem: "What is the body temperature of a chinchilla?", "What is the definition of cold-blooded?", "Are chinchillas cold-blooded?"
>
> 1. The body temperature of a chinchilla is 37°C.
> 2. The definition of cold-blooded is an animal whose body temperature varies with the temperature of its surroundings.
> 3. 37°C is not the same as the temperature of its surroundings. So chinchillas are not cold-blooded.
>
> The answer is: false.
>
> ---
>
> We hope the above decomposition generalizations look fun to you. We did a full empirical study by rewriting the examples in the CoT prompt (from the CoT paper) into the L2M style by adding decomposition into each example, as we have shown in the above examples. We obtained an accuracy of 75.4%. This is kind of the same performance as CoT rather than significant outperforming one would expect.
>
> We did error analysis for understanding the performance of L2M/CoT and their comparison:
> - The performance bottleneck for both CoT and L2M is that LLMs lack factual knowledge rather than reasoning or decomposition issues.
>    However, compared to CoT, when using L2M, the grounding issue could be more easily fixed by grounding those simple-fact subquestions, e.g.
>    using Google to answer those subquestions;
> - All problems in StrategyQA is at the same level of difficulty: solved within at most 3 steps. So this task is fundamentally different from the tasks in this submission where we aim at easy-to-hard generalization: solving problems harder than the examples in the prompt, e.g. longer list in the last-letter task, longer action sequences in SCAN, more hops in GSM8K.  **L2M shows big advantages in easy-to-hard generalization tasks since every subproblem in a complex problem is just as easy as those example problems in the prompt.**

---

> ### Author Response · Authors · 2022-11-15
> **Authors’ rebuttal [1/3]: decomposition does generalize**
>
> **Concern: The model may only need to copy patterns from the few-shot demos without any generalization. Does the model have the ability to decompose questions very different from demo examples into correct sub-problems?**
>
> Let’s look at the GSM8K task as an example. This dataset contains thousands of grade school math word problems which require many different decomposition patterns that cannot be covered by the demonstration examples in the prompt context. Instead, the model needs to perform a significant amount of generalization to obtain the decompositions.
>
> Let’s use the simple prompt context shown in Table 11 to illustrate how the decomposition generalizes. We repeat this context here for the convenience of comparison:
>
> ---
>
> Q: Elsa has 5 apples. Anna has 2 more apples than Elsa. How many apples do they have together?
>
> A: Let’s break down this problem: 1. How many apples does Anna have? 2. How many apples do Elsa and Anna have together?
>
> 1. Anna has 2 more apples than Elsa. So Anna has 2 + 5 = 7 apples.
>
> 2. Elsa and Anna have 5 + 7 = 12 apples together.
>
> ---
>
> Now we use it to solve the following problem. Note that the decomposition is significantly different. Among others, it uses 3 rather than just 2 steps.
>
> Q: Eliza's rate per hour for the first 40 hours she works each week is \\$10. She also receives an overtime pay of 1.2 times her regular hourly rate. If Eliza worked for 45 hours this week, how much are her earnings for this week?
>
> Here is the output  (using code-davinci-002, with its default setup):
>
> ---
>
> A: Let's break down this problem: 1. How much does Eliza earn for the first 40 hours she works each week? 2. How much does Eliza earn in overtime pay? 3. How much does Eliza earn in total for the week?
>
> 1. Eliza earns \\$10 per hour for the first 40 hours she works each week, so she earns \\$10 x 40 = \\$400 for those hours.
> 2. Eliza earns 1.2 times her regular hourly rate for overtime hours, so she earns 1.2 x \\$10 = \\$12 per overtime hour. Eliza worked 5 overtime hours this week, so she earned \\$12 x 5 = \\$60 in overtime pay.
> 3. Eliza's total earnings for the week are \\$400 + \\$60 = \\$460.
>
> ---
>
> Here is another example. Again, the decomposition is significantly different and requires a total of 8 steps!
>
> Q: Kim has started his own housekeeping business and is calculating how much profit he will make from his clients. He already has 3 clients, but is talking to another 5 potential clients and feels confident enough to include them in his calculations. Each client’s home will need 2 bottles of bleach and a pack of cloths to clean. Bottles of bleach will cost \\$2 each and packs of cloths will cost \\$5 each. These are his only expenses. He calculates that his total income each week will be \\$92. Profit is the difference between total income and total expenses, so how much profit, in dollars, will Lucas make each week?
>
> ---
>
> A: Let's break down this problem: 1. How many clients does Kim have? 2. How many bottles of bleach does Kim need? 3. How many packs of cloths does Kim need? 4. How much does Kim spend on bleach? 5. How much does Kim spend on cloths? 6. How much does Kim spend in total? 7. How much does Kim make in total? 8. How much profit does Kim make?
>
> 1. Kim has 3 clients and is talking to another 5 potential clients. So Kim has 3 + 5 = 8 clients.
> 2. Each client’s home will need 2 bottles of bleach. So Kim needs 2 * 8 = 16 bottles of bleach.
> 3. Each client’s home will need a pack of cloths. So Kim needs 8 packs of cloths.
> 4. Bottles of bleach will cost \\$2 each. So Kim spends 2 * 16 = \\$32 on bleach.
> 5. Packs of cloths will cost \\$5 each. So Kim spends 5 * 8 = \\$40 on cloths.
> 6. Kim spends \\$32 + \\$40 = \\$72 in total.
> 7. Kim makes \\$92 in total.
> 8. Profit is the difference between total income and total expenses. So Kim makes \\$92 - \\$72 = \\$20 profit.
>
> ---
>
>
> We believe that the above examples illustrate non-trivial decomposition generalization that go far beyond copying of patterns.
>
> We would like to highlight that we are talking about “patterns” from a machine learning perspective not from humans (for a mathematican, all the above math word problems have the same pattern: elementary math problems).  If we agree with this argument point, even for the simple task of the last-letter, although the pattern is clear for humans, that does not mean that pattern is clear for machines. Otherwise, CoT would perform as well as L2M by simply following the solution pattern. The generalizaton in the last lettern task is usually called length-generalization in the literature, and the generalization in SCAN is compositional generalization. Both generalizations/patterns are obvious for humans but have proven to be really challenging for machine learning. Training seq-2-seq models for those tasks with tons of labeled data can only achieve accuracies of about 10%.

---

### Official Review · Reviewer_JgLG · 2022-10-25

**Confidence:** 4
**Correctness:** 2
**Technical Novelty And Significance:** 3
**Empirical Novelty And Significance:** 3
**Recommendation:** 6

**Clarity, Quality, Novelty And Reproducibility:**


** Clarity **

The paper is generally clearly written and is a pleasure to read. There are some minor issues (which I list below), but they did not strongly affect my overall assessment of the paper.

First, the phrase at the end of page 5 ("We find that the ...) is continued amid a number of tables on page 6 ("... solving rate remains the same"), and then almost immediately followed by another table. It'd be better to move text or tables around so that it's a bit easier to read.

Appendices are extremely thorough (which is a plus), but they also feature a lot of material that is difficult to navigate. It may, potentially, be better to move some of the additional results and demonstrations into supplementary materials.

The abundance of model input/output printouts in the main body of the paper is helpful, but I think that some of them may be condensed and explained verbally in text.

Overall, the clarity is up to the standards of the ICLR conference.

** Quality **

The experiments cover a range of substantially different tasks, both simulated and realistic. There are a few issues with experiment design, however, some of which I find substantial.

Experiment 1 (symbolic manipulation):
- The prompts differ not only in the recursive part ('So, “think, machine” outputs
“ke”'), but also in the examples themselves. The chain-of-thought prompts in table 5 use completely different sets of words for length 2 and length 3 prompts. Because of that, it's impossible to tell whether it's the selection of words or the inclusion of the "reduction"/"recursive" phrase that is key for the method to work.
- "Such a trivial reduction can be done straightforwardly by a manually written script or prompting language models with several
demonstrations." regarding breaking a multi-word problem in to smaller constituents. However, if done with a script, the comparison with the regular "chain-of-thought" method is unfair. At the same time, if done using prompting, while straightforward, it can still break down on longer prompts. As far as I can tell, no detail is given in the paper on the "straightforward" prompting, and so the generalization performance of such prompting on longer sequences can not be assessed. Therefore, it may be that the improvements we see are due to better decomposition that may be delegated to an external script in the least-to-most prompting case and that has to be done internally by the language model in the case of "chain-of-thought" prompting.

Because of these concerns, the results from the first experiment can not be fully interpreted.

Experiment 2 (SCAN):
The description of the experimental procedure is, again, partially unclear:
"To generate the final results, we either run a postprocessing script or prompt language models to expand the Python expressions. It is straightforward to teach language models to expand Python expressions via a few demonstrations."
It's unclear which method was used and whether there was a difference in performance. Again, while prompting may be straightforward,
language models are known to fail in unexpected ways on longer sequences. If scripts were used as part of Python expression expansion,
it makes the comparison between the models unfair.

What makes me especially worried is that in the SCAN experiment, it's not clear, why having Python notation such as " * 2" would be better than having verbal notation (such as "twice"), so it makes me worried that the benefit may be in reducing the errors during the decoding stage.

Experiment 3:
This issue applies to different experiments too, but is most evident in experiment 3.
I believe that for a fair comparison between prompting methods it is reasonable to normalize by the amount of information given. Least-to-most prompting essentially gives more prompts than the "chain of thought" one. They are partially redundant, but, nevertheless, in a way, the least-to-most prompting can be benefitting from the sheer number of examples (a solution to each subproblem may be seen as an example).

From the practical standpoint, it is also reasonable to normalize by the amount of instruction provided to a language model, and I don't think it is 100% fair to say that "least-to-most prompting" and "chain-of-thought" prompting receive equivalent amount of instruction because both are two-shot, when each "least-to-most" prompting "shot" takes twice or more space in terms of volume of text.

While not all of these issues are crucial, I believe that overall, they create an inconsistent impression, and because of that, I feel that the quality of the contribution is marginally below the requirements of the ICLR conference.

** Novelty **

The work is substantially novel and original, and expand upon the "chain-of-thought" prompting in a meaningful way.

** Reproducibility **

I have mentioned some of the reproducibility issues in the "quality" section. The authors also do not provide the code, which could have helped to mitigate the issue.

** Other suggestions **

Regarding the last sentence on page 4 -- "These look somewhat reminiscent of careless mistakes by humans". I find that the paper will benefit if this sentence is removed.

First, I disagree that even a careless human would end up with "wsgss" when concatenating "wsg" and "s". Second, the subtle implication that the model is somehow humanlike in its behaviour is very ambitious. When it's based on such loose grounds, it gives a very speculative impression, which overall damages the perceived credibility and rigour of the paper.

This is a minor suggestion that did not affect my overall assessment.

**Details Of Ethics Concerns:**

The authors reference additional experiments on a 540 billion parameter language model in the appendix. It's not central to the contribution, and the added value of this comparison is dubious.

The reason I mention it is that I believe that it might place additional pressure on reviewers, since it is well known that there is only one such model (PaLM) and only one research group with access to it (Google Research).

**Strength And Weaknesses:**

Strengths:
- The problem of teaching language models new skills "on the fly" is extremely relevant
- The paper has numerous illustrations, which helps to understand the approach and model behavior
- The approach is novel and original

Weaknesses:
- There are a number of issues with experimental design. I detail those in the "quality" section of the review.
- In terms of reproducibility, some elements of the approach are not very clearly reported, specifically, in the first two experiments, the authors say that either a script or simple prompting can be used (for decomposition in experiment 1, and for expanding python expressions), and it's not specified what was actually used.

Here is one of the relevant quotes:
"To generate the final results, we either run a postprocessing script or prompt language models to expand the Python expressions. It is straightforward to teach language models to expand Python expressions via a few demonstrations.
However, I can not find a more detailed description nor the results on the performance of these straightforward expansions. While straightforward from the prompting standpoint, it might still result in less than perfect accuracy & providing exact prompts is crucial for replication.

- While thorough, the abundance of tables and model input/output printouts is, at times, getting in the way of readability.

**Summary Of The Paper:**

The authors propose a new prompting method, called "Least To Most" prompting. Building upon the success of the "chain-of-thought" prompting method, the authors propose to split the prompting process into two stages: problem reduction, and problem solving. The former focusing on decomposing the task at hand into its constituent parts, the latter corresponding to solving each subproblem and combining the results into the final answer. The authors show that their model outperforms "chain of thought" prompting on a number of tasks, including both simulated tasks like  last letter concatenation and the SCAN dataset and on more realistic mathematical reasoning problems (Drop and GSM8k).

**Summary Of The Review:**

Overall, I enjoyed reading the paper, and I believe that it has a great potential. I am very torn when it comes to its assessment, since while many of the issues I've mentioned are minor, they accumulate to give the paper an inconsistent impression. The experimental section issues are the most important. Experiment 3 is, overall, the most convincing, but given that it also has some methodological concerns, and, given the issues with experiment 1 and 2, I lean towards a weak rejection. I will, however, stay open to revise my assessment based on other reviews and authors' rebuttal.

** UPD **
Many of my concerns were properly addressed, therefore I increase the score, pushing it above the acceptance threshold. I would have given it a 7, but the interface does not allow it.

That being said, I believe that additional results and clarifications dramatically improve the quality of the paper, and that it is now above the acceptance threshold and is of interest to a large portion of the ICLR community. In fact, I hope that the paper is accepted.

---

> ### Author Response · Authors · 2022-11-13
> **Rebuttal for experimental design [3/3]: All experiments, the amount of information is not normalized**
>
> **Concern: The amount of information provided to various methods is not normalized.**
>
> Because L2M prompting consists of a decomposition prompt as well as a solution prompt, it is an interesting question to what degree the performance gain is caused by the additional information / instruction volume rather than by the fundamental design of the method.
>
> While the data presented in the paper provides a fair amount of evidence that the information volume plays a minor role, we acknowledge that the paper would benefit from an experiment that investigates this question more explicitly.
>
> We therefore constructed a variation of the last-letter concatenation experiment where we compare the L2M and CoT prompting methods with different amounts of total instruction, and we will add this experiment to the paper.
>
> For L2M prompting, we use 2 exemplars for decomposing the sublists (see the above) and 2 exemplars for the solution. We compare this to a CoT prompt context where we use the same number (4) of exemplars and a CoT prompt context where we use twice as many exemplars,  that is, 8 exemplars.
>
> The results are presented below. (Note that all results except the CoT prompt context with 8 exemplars are already provided in Table 12 of the Appendix). They show that L2M substantially outperforms CoT even if CoT uses twice the number of exemplars. In fact, going from 4 to 8 exemplars for CoT mostly improves the accuracy for short sequences but does not help much for longer sequences. Note that in addition to the number of exemplars, we also provide the number of tokens (as parsed by code-davinci-002) as a more precise measure of the provided information.
>
> || #Tok |  L = 4 | L = 6 | L = 8 | L = 10 | L = 12 |
> |-|--------|------|--------|-------|-------|--------|
> |CoT (4) | 287 | 88.6 | 77.0 | 53.4 | 44.0 | 37.4 |
> | CoT (8) |	573 | 91.0 | 79.8 | 56.8 | 46.8 | 38.4 |
> | L2M (2+2)  | 203 | $\textbf{94.0}$ | $\textbf{88.4}$| $\textbf{83.0}$ | $\textbf{76.4}$ | $\textbf{74.0}$ |
>
> **This result shows that L2M is substantially more data efficient than CoT**, which is in turn substantially more data efficient than standard prompting. (The accuracy of standard prompting remains at 0% no matter how many exemplars we use.)

---

> > ### Comment · Reviewer_JgLG · 2022-11-23
> > **Thank you for your response!**
> >
> > I deeply appreciate your thorough and thoughtful response to my concerns. I believe that the additional results make the paper much better.
> >
> > I increased the score to make it above the acceptance threshold. I would have given the paper a "7", but it's not possible. I can not quite give it an 8, but it is close. That being said, with additional results, I am confident that the paper is good enough to be accepted to ICLR and to be of interest to a large part of the community. I hope it is accepted.

---

> > > ### Author Response · Authors · 2022-11-28
> > > **"Since I can not see the final, improved version of the paper, I can't 100% ascertain that all the changes are incorporated optimally and clearly, hence I can not give an "8" at present".**
> > >
> > > Thank you for your response to our rebuttal! We are happy to know that our rebuttal resolved your concerns.
> > >
> > > "Since I can not see the final, improved version of the paper, I can't 100% ascertain that all the changes are incorporated optimally and clearly, hence I can not give an "8" at present".
> > >
> > > As you might have seen, we have already accordingly updated our submission. We would love to know if you have any other concern on our work.
> > >
> > > ==========
> > >
> > > You updated your response to "I can not quite give it an 8, but it is close".  Again, we would love to know the further concern that you might have.
> > >
> > > We appreciate you said "I hope it is accepted".

---

> ### Author Response · Authors · 2022-11-13
> **Rebuttal for experimental design [2/3]: SCAN**
>
> **Concern: Python expression expansion makes the comparison between the methods unfair.**
>
> To clarify, for results in Table 8, all 3 methods (standard prompting, CoT, L2M) generate Python expressions, and then we use the Python interpreter to expand those expressions into action sequences. Thus, the comparison in Table 8 is fair.
>
> **Concern: Why having Python notation such as " * 2" would be better than having verbal notation (such as "twice".**
>
> **We utilize the Python notation as an intermediate representation (IR) of the final result.** In prior works on solving SCAN, it is common to design an IR to achieve better performance, then the IR will be further translated to action sequences via an executor. For example, in Chen et al., 2020, the neural model generates a trace of symbolic operators, and then a symbolic machine executes the trace to produce the output sequences. Liu et al., 2020 designed a form of analytical expressions as their IR. Another line of work trained models to predict grammar rules (Nye et al., 2020; Shaw et al., 2021), so that input sentences are parsed into the action sequences using the predicted grammar rules, instead of directly generating the output sequences from the input. We followed this line of work to design our approach.
>
> **Concern: Performance difference between running a Python executor and prompting.**
>
> **Prompting language models to expand the Python expressions remains >99% accuracy.** We present a prompt to expand the Python expressions below. Using the code-davinci-002, this prompt achieves 99.7% accuracy on a random sample of 1000 Python expressions that are outputted by the L2M solution. This demonstrates that the L2M method can solve SCAN with an accuracy >99% even if we do not use the Python executor, and instead perform each step via prompting.
>
> The prompt we constructed includes 6 examples to cover:
> - Exemplar 1: multiplication
> - Exemplar 2: sequential multiplication
> - Exemplar 3: addition
> - Exemplar 4: no commutative/associative in addition
> - Exemplar 5: nested multiplication
> - Exemplar 6: addition of two multiplications
>
> We add numbers and parentheses in multiplication to make it easier for the model to keep track of the repetitions. With this prompt, the postprocessing step only requires the removal of these numbers and parenthesis, without the Python executor. Please let us know if you feel that this still does not address your concern in a satisfactory manner.
>
> ---
>
> Q: ``JUMP'' * 3
>
> Rewrite: "JUMP" * 3
>
> A: 1 JUMP 2 JUMP 3 JUMP
>
> ---
>
> Q: "RUN" * 4 * 2
>
> Rewrite: "RUN" * 8
>
> A: 1 RUN 2 RUN 3 RUN 4 RUN 5 RUN 6 RUN 7 RUN 8 RUN
>
> ---
>
> Q: "TURN_RIGHT" + "WALK"
>
> Rewrite: "TURN_RIGHT" + "WALK"
>
> A: TURN_RIGHT WALK
>
> ---
>
> Q: ("TURN_LEFT" + "LOOK") * 2 + "TURN_LEFT" + "LOOK"
>
> Rewrite: ("TURN_LEFT" + "LOOK") * 2 + "TURN_LEFT" + "LOOK"
>
> A: 1 (TURN_LEFT LOOK) 2 (TURN_LEFT LOOK) TURN_LEFT LOOK
>
> ---
>
> Q: ("TURN_RIGHT" * 2 + "JUMP") * 4
>
> Rewrite: ("TURN_RIGHT" * 2 + "JUMP") * 4
>
> A: 1 (1 TURN_RIGHT 2 TURN_RIGHT JUMP) 2 (1 TURN_RIGHT 2 TURN_RIGHT JUMP) 3 (1 TURN_RIGHT 2 TURN_RIGHT JUMP) 4 (1 TURN_RIGHT 2 TURN_RIGHT JUMP)
>
> ---
>
> Q: "TURN_LEFT" * 2 + ("TURN_RIGHT" + "WALK") * 4 * 2
>
> Rewrite: "TURN_LEFT" * 2 + ("TURN_RIGHT" + "WALK") * 8
>
> A: 1 TURN_LEFT 2 TURN_LEFT 1 (TURN_RIGHT WALK) 2 (TURN_RIGHT WALK) 3 (TURN_RIGHT WALK) 4 (TURN_RIGHT WALK) 5 (TURN_RIGHT WALK) 6 (TURN_RIGHT WALK) 7 (TURN_RIGHT WALK) 8 (TURN_RIGHT WALK)
>
> ---

---

> ### Author Response · Authors · 2022-11-13
> **Rebuttal for experimental design [1/3]: symbolic manipulation**
>
> We really appreciate your detailed and constructive feedback. Hope you will see this reply thread is helpful in addressing your concerns about quality / experimental design. We will update our paper accordingly to make it more clear and also take into account your other suggestions.
>
> ***  Quality ***
>
> Issues with experimental design.
>
> **Experiment 1 (symbolic manipulation)**
>
> **Concern: The prompts differ not only in the recursive part, but also in the chosen exemplars themselves.**
>
> For the CoT prompt context, we deliberately chose two different exemplars since CoT performs better if the exemplars are independent. (The independence helps maximize the information.) We recognize that this is not obvious from the submitted version of the paper, which makes it harder to compare CoT and L2M in a controlled fashion.
>
> To address this concern, we performed additional experiments where we use for CoT the same exemplars as in L2M. Then, the CoT prompt context becomes:
>
> ---
> Q: “think, machine”
>
> A: The last letter of “think” is “k”. The last letter of “machine” is “e”. Concatenating “k”, “e” leads to “ke”. So, “think, machine” outputs “ke”.
>
> Q: “think, machine, learning”
>
> A: The last letter of “think” is “k”. The last letter of “machine” is “e”. The last letter of “learning” is “g”. Concatenating “k”, “e”, “g” leads to “keg”. So, “think, machine, learning” outputs “keg”.
>
> ---
>
> Here are the results when using this prompt context on the code-davinci-002 model (new = using the prompt above, old = using the prompt in the paper):
>
> L=4:	new=84.2%, old= $\mathbf{89.4\\%}$
>
> L=6:	new=69.2%, old=$\mathbf{75.0\\%}$
>
> L=8:	new=50.2%, old=$\mathbf{51.8\\%}$
>
> L=10: new=$\mathbf{39.8\\%}$, old=$\mathbf{39.8\\%}$
>
> L=12: new=31.8%, old=$\mathbf{33.6\\%}$
>
> The numbers show that on average the new prompt is worse than the old one. This matches our expectation as the new prompt context contains less information than the old one because of redundancy in its exemplars. Most importantly, this allows a controlled comparison between L2M and CoT, and it shows that the difference in performance is not caused by the choice of exemplars but is fundamental to the recursive design of L2M.
>
> **Concern: Details about breaking down a list of words into a sequence of sublists.**
>
> As you correctly pointed out, the submitted version of the paper does not provide details on how we obtained the sequence of sublists from the initial list of words. We recognize that this prevents a fully controlled comparison between CoT and L2M prompting.
>
> To address this concern, we provide the following prompt for breaking down a list of words into a sequence of sublists (and we will add it to the paper together with the discussion below).
>
> ---
>
> Q: "machine, learning"
>
> A: creating sequential sublists of the list "machine, learning":
>
> "machine"
>
> "machine, learning"
>
> Q: "machine, learning, artificial"
>
> A: creating sequential sublists of the list "machine, learning, artificial":
>
> "machine"
>
> "machine, learning"
>
> "machine, learning, artificial"
>
> ---
>
> Even though this prompt only contains 2 exemplars for lists of sizes 2 and 3, **it achieves 100% accuracy** on lists up to size 12 (which is the maximum that we need for our experiment) using the code-davinci-002 model. This demonstrates that it is easy for the model to generate the sequence of sublists. Hence, the difference in performance between L2M and CoT is not due to better decomposition performed by an external script but is fundamental to the recursive design of L2M.
>
> As an illustration of the above prompt, we provide the following example (using code-davinci-002 with its default setup):
>
> Q: "Least, to, Most, Prompting, Enables, Complex, Reasoning, in, Large, Language, Models"
>
> A: creating sequential sublists of the list "Least, to, Most, Prompting, Enables, Complex, Reasoning, in, Large, Language, Models":
>
> "Least"
>
> "Least, to"
>
> "Least, to, Most"
>
> "Least, to, Most, Prompting"
>
> "Least, to, Most, Prompting, Enables"
>
> "Least, to, Most, Prompting, Enables, Complex"
>
> "Least, to, Most, Prompting, Enables, Complex, Reasoning"
>
> "Least, to, Most, Prompting, Enables, Complex, Reasoning, in"
>
> "Least, to, Most, Prompting, Enables, Complex, Reasoning, in, Large"
>
> "Least, to, Most, Prompting, Enables, Complex, Reasoning, in, Large, Language"
>
> "Least, to, Most, Prompting, Enables, Complex, Reasoning, in, Large, Language, Models"

---

### Decision · Program_Chairs · 2023-01-20

**Decision:**

Accept: poster

**Justification For Why Not Higher Score:**

The paper comes with strong empirical results and demonstrates the effectiveness of the proposed approach. However, as one of the reviewers highlighted, it still needs deep insights into why it works (and why it's not just performing prompt engineering on some specific datasets). It will be an excellent paper if the authors consider adding further insights. Yet, the authors did not directly respond to this comment in their response.

I have noted the authors' complaints to the first reviewer when making this recommendation.


**Justification For Why Not Lower Score:**

This is a good paper that reports interesting findings. It can be accepted.

**Metareview: Summary, Strengths And Weaknesses:**

This paper presents a new approach to using prompts for solving reasoning-related problems using large language models. The idea is to use prompts to decompose a complex problem into smaller subproblems. The authors show that doing this would encourage the large language models to focus on smaller problems before answering the original problem. Experiments show improved results using such an approach on some datasets.

The reviewers agree that this work conveys original ideas worth sharing with the community. The strong empirical results reported are another piece of evidence added to the community that shows the significance of developing clever ways to interact with large language models using prompts. The authors also addressed many of the questions raised by the reviewers in the rebuttals.

There are some limitations of the work. As highlighted by one of the reviewers, a major one is that the current draft needs to offer deeper insights into such results. While understanding the behaviors of large language models can be challenging, including some insights (based on case studies, or other types of empirical analyses, if theoretical insights are difficult to establish) would make the current work much stronger. Conducting such investigations would also alleviate the concern about whether we are making solid progress rather than performing prompt engineering over some specific datasets, as the current approach relies on manually designed prompts for each dataset.

**Note From Pc:**

if the above contains the word "oral" or "spotlight" please see: "oral" presentation means -> notable-top-5% and "spotlight" means -> notable-top-25%. As stated in our emails, we are disassociating presentation type from AC recommendations